# Streaming Linear System Identification with Reverse Experience Replay

**Prateek Jain**
Google AI Research Lab,
Bengaluru, India 560016
prajain@google.com

**Suhas S Kowshik**
Department of EECS
MIT,
Cambridge, MA 02139
suhask@mit.edu

**Dheeraj Nagaraj**
Department of EECS
MIT,
Cambridge, MA 02139
dheeraj@mit.edu

**Praneeth Netrapalli**
Google AI Research Lab,
Bengaluru, India 560016
pnetrapalli@google.com

## Abstract

We consider the problem of estimating a linear time-invariant (LTI) dynamical system from a single trajectory via streaming algorithms, which is encountered in several applications including reinforcement learning (RL) and time-series analysis. While the LTI system estimation problem is well-studied in the *offline* setting, the practically important streaming/online setting has received little attention. Standard streaming methods like stochastic gradient descent (SGD) are unlikely to work since streaming points can be highly correlated. In this work, we propose a novel streaming algorithm, SGD with Reverse Experience Replay (SGD − RER), that is inspired by the experience replay (ER) technique popular in the RL literature. SGD − RER divides data into small buffers and runs SGD backwards on the data stored in the individual buffers. We show that this algorithm exactly deconstructs the dependency structure and obtains information theoretically optimal guarantees for both parameter error and prediction error. Thus, we provide the first – to the best of our knowledge – optimal SGD-style algorithm for the classical problem of linear system identification with a first order oracle. Furthermore, SGD − RER can be applied to more general settings like sparse LTI identification with known sparsity pattern, and non-linear dynamical systems. Our work demonstrates that the knowledge of data dependency structure can aid us in designing statistically and computationally efficient algorithms which can "decorrelate" streaming samples.

## 1 Introduction

In this paper, we study the problem of learning linear-time invariant (LTI) systems, where the goal is to estimate the matrix $A^* \in \mathbb{R}^{d \times d}$ from the given samples $(X_0, \ldots, X_T)$ that obey:

$$X_{\tau+1} = A^* X_\tau + \eta_\tau, \quad X_\tau \in \mathbb{R}^d, \quad \eta_\tau \overset{i.i.d.}{\sim} \mu, \tag{1}$$

where $\mu$ is an unbiased noise distribution. The problem is central in control theory and reinforcement learning (RL) literature [1, 2]. It is also equivalent to estimating Vector Autoregressive (VAR) model popular in the time-series analysis literature [3], where it has been used in several applications like finding gene regulatory information network [4].

Despite a long line of classical literature for the problem, most of the existing results focus on the *offline* setting, where all the samples $(X_0, \ldots, X_T)$ are available apriori. In this setting, ordinary

35th Conference on Neural Information Processing Systems (NeurIPS 2021).

least squares (OLS) method that estimates $A$ as, $\hat{A} = \arg\min_A \sum_{\tau=0}^{T-1} \|X_{\tau+1} - AX_\tau\|^2$ is known to be nearly optimal [5, 6]. However, such offline solutions do not apply to the streaming setting – where $A^*$ needs to be estimated online – that has applications in several domains like RL, large-scale forecasting systems, recommendation systems [7, 8].

In this paper, we study the above mentioned problem of learning LTI systems via first order gradient oracle with streaming data. The goal is to design an estimator that provides accurate estimation while ensuring nearly optimal time complexity and space complexity that is nearly *independent* of $T$. Note that due to specific form arising in linear regression, the optimal solution to OLS can be estimated in online fashion using Sherman-Morrison-Woodbury formula. But such a solution is limited and does not apply to practically important settings like *generalized non-linear dynamical* system or when $A^*$ is high-dimensional and has special structure like low-rank or sparsity [9, 10].

So, in this work, we focus on designing Stochastic Gradient Descent (SGD) style methods that can work directly with first order gradient oracle, and hence is more widely applicable to the settings mentioned above. In fact, after the first appearance of this manuscript, the algorithm (SGD − RER) and the techniques introduced in this paper were used to obtain near-optimal guarantees for learning certain classes of *non-linear dynamical systems* [11] as well as in Q-learning tabular MDPs in RL [12]. We note that prior to [11], even optimal *offline* algorithms were unknown for such non-linear systems.

SGD is a popular method for general streaming settings, and has been shown to be *optimal* for problems like streaming linear regression [13]. However, when the data has temporal dependencies, as in the estimation of linear dynamical systems, such a naive implementation of SGD may not perform well as observed in [14, 15]. In fact, for linear system identification, our experiments suggest that SGD suffers from a non-zero bias (Section 6). In order to address temporal dependencies in data, practitioners use a heuristic called *experience replay*, which maintains a *buffer* of points, and samples points *randomly* from the buffer. However, for linear system identification, experience replay does not seem to provide an accurate unbiased estimator for reasonable buffer sizes (see Section 6).

In this work, we propose *reverse experience replay* for linear system identification. Our method maintains a small *buffer* of points, but instead of random ordering, we replay the points in a *reverse* order. We show that this algorithm exactly unravels the temporal correlations to obtain a consistent estimator for $A^*$. Similar to the standard linear regression problem with *i.i.d.* samples, we can break the error in two parts: a) bias: that depends on the initial error $\|A_0 - A^*\|$, b) variance: the steady state error due to noise $\eta$. We show that our proposed method, under fairly standard assumptions and with a small buffer size, is able to decrease the bias at fast rate, while the variance error is nearly optimal (see Theorem 1), matching the information theoretic lower bounds [5, Theorem 2.3]. To the best of our knowledge, we provide first non-trivial analysis for a purely streaming SGD-style algorithm with optimal computation complexity and nearly bounded space complexity that is dependent logarithmically on $T$. We note here that the idea of reverse experience replay was independently discovered in experimental reinforcement learning by [16] based on reverse replay observed in Hippocampal place cells [17] in Neurobiology. We also refer to [18] for more on this connection.

In addition to the transition matrix estimation error $\|A - A^*\|$, we also provide analysis of prediction error, i.e., $E[\|AX - A^*X\|^2]$ (see Theorem 2). Here again, we bound the *bias* and the *variance* part of the error separately. We further derive new lower bounds for prediction error (see Theorem 4) and show that our algorithm is minimax optimal, under standard assumptions on the model. As mentioned earlier, our method work with general first order oracles, hence applies to more general problems like *sparse LTI estimation* with known sparsity structure and unlike online OLS methods, SGD − RER has nearly optimal time complexity. Finally, we also provide empirical validation of our method on simulated data, and demonstrate that the proposed method is indeed able to provide error rate similar to the OLS method while methods like SGD and standard experience replay, lead to biased estimates.

**Related Work.** Due to applications in RL, recently LTI system identification has been widely studied. In particular, [19] studied the problem in offline setting under the "stability" condition, i.e., the spectral radius ($\rho(A^*)$) of $A^*$ is a constant bounded away from 1. The sequence of papers [5, 6, 20, 21] provide optimal analyses of the offline OLS estimator beyond assumptions of stability. That is, they show that OLS recovers $A^*$ near optimally even the process defined by (1) is stable but does not mix within time $T$ (when $\rho(A^*)$ is $1 - O(1/T)$) or is unstable (when $\rho(A^*)$ is larger

than 1). Further [5, 22] provide information theoretic lower bounds for the LTI system identification problem. [11, 23, 24] consider the problem of identifying non-linear dynamical systems of the form $X_{t+1} = \phi(A^* X_t) + \eta_t$ where $\phi$ is a one dimensional link function which acts co-ordinate wise. In this setting, however, there is no closed for expressions for the estimator of $A^*$. [23, 24] give offline algorithms whose error guarantees are worse off by factors of mixing time whereas [11] obtains near optimal offline and streaming algorithms for this setting. In fact, [11] uses $\mathsf{SGD} - \mathsf{RER}$ which was first introduced in this work in order to obtain the streaming algorithm.

LTI identification problem has been studied in time series forecasting literature as well. For example, [25] obtains asymptotic consistency results for system identification problem and [26, 27] consider the problem of finite time recovery. Both consider a certain parameterized predictor for a linear system with empirical risk minimization for the parameter and analyzes the deviation from population risk. Similarly, [28] also studies generalization error guarantees. In contrast, our work is able to provide precise bias and variance (similar to generalization error) of the estimator in the streaming setting, and show that the asymptotic error is minimax optimal.

[29] studied SISO systems with observations $(x_\tau, y_\tau) \in \mathbb{R}^2$ and a hidden state $h_\tau$ which is high dimensional, thus their model and applications are significantly different than the LTI system we study. For the SISO system, [29] analyzes SGD to provide error bounds contain (a large) polynomial in the hidden state dimension. Here, the hidden state has an evolution similar to Equation 1 whereas $x_1, \ldots, x_T$ are drawn i.i.d from some distribution.

System identification has been studied in the context of partially observed LTI systems as well. Recent works [19, 30–34] focus on identifying a certain Hankel-like matrix of the system. These are not directly comparable to the fully observed setting in this work since the model parameters are identifiable only upto a similarity transformation in the partially observed setting.

Recently, there has been an exciting line of work in the related domain of online control (see [35–38] and references therein). The state equation studied in these papers also contain an additive term of $Bu_\tau$ for some unknown matrix $B$ and a control signal $u_\tau$ and the noise $\eta_\tau$ is either stochastic (as in [35]) or adversarial (as in [36–38]). The goal is to output control signals $u_\tau$ after observing $X_1, \ldots, X_\tau$, such that the cost $\sum_\tau c_\tau(X_\tau, u_\tau)$ is minimized for some sequence of convex costs $c_\tau$. We focus on the LTI system identification(or estimation) problem while the goal of the above mentioned line of work is to design an online controller.

We also note here another line of works [32, 39–44] focused on online prediction of both fully observed and partially observed LTI systems, and the similar problem of time series forecasting by regret minimization [28, 45]. In particular, the main goal there is to design online prediction algorithms minimizing regret against a certain class (for instance, against a Kalman filter with knowledge of the system parameters in the case of partially observed LTI systems). The situation considered in our work is different in atleast two aspects: 1) we focus significantly on parameter recovery or system identification and 2) our notion of prediction is *prediction at stationarity* which can be thought of as one-step regret (compared to $T$–step regret for instance in [39, 40]).

Finally, [9] considers *offline* sparse linear regression with $\ell_1$ penalty where the feature vector is derived from an auto regressive model. Similarly, [14] considers the problem of linear regression where the feature vectors come from a Markov chain. This line of work is different from ours in that we try to estimate the parameters of the Markov process itself.

**Paper Organization.** We provide the problem definition and introduce the notations in the next section. We then present our algorithm and the key intuition behind it in Section 3. We then present our main result in Section 4 and provide a proof sketch in Section 5. Finally, we present simulation results in Section 6.

## 2 Problem Setting and Notation

In this section, we first introduce the data generation model, the required assumptions and then provide the precision problem definition. Throughout the paper, we use $\|A\|$ to denote the operator norm of $A$ unless otherwise specified. $\|A\|_F$ denotes the Frobenius norm of $A$. $\sigma_i(A)$ denotes the $i$-th largest singular value of $A$, i.e., $\sigma_{\max}(A) = \sigma_1(A)$. $\kappa(A) := \sigma_{\max}(A)/\sigma_{\min}(A)$ denotes the condition number of $A$. $\rho(A)$ denotes the spectral radius of $A$. For two symmetric matrices

$A, B \in \mathbb{R}^{d \times d}$ we say $A \preceq B$ if $B - A$ is positive semidefinite (psd). For notational simplicity, we use $C$ to denote a constant, and it's value can be different in different equations.

**Linear Dynamical System/VAR(1) model.** Given an initial (possibly random) data point $X_0$ which is independent of the noise sequence, we generate the $(X_0, \ldots, X_T)$ from the VAR model as:

$$X_{\tau+1} = A^* X_\tau + \eta_\tau, \quad 0 \le \tau \le T - 1, \tag{2}$$

where $A^* \in \mathbb{R}^{d \times d}$ be the transition matrix. Let $\eta_1, \ldots, \eta_T \in \mathbb{R}^d$ be an i.i.d noise sequence with 0 mean and finite second moment with probability measure $\mu$. We will denote this model by $\mathsf{VAR}(A^*, \mu)$. We also make the following assumptions about $A^*$, $\mu$, and $X_0$:

**Assumption 1.** *External Stability.* $\|A^*\| < 1$

**Assumption 2.** *Sub-Gaussian Noise.* $\mu$ *has co-variance* $\Sigma$ *and for all* $x \in \mathbb{R}^d$, $\langle x, \eta_\tau \rangle$ *is* $C_\mu \langle x, \Sigma \cdot x \rangle$ *sub-Gaussian. Further,* $\Sigma$ *is full rank. Also, let* $\mu_4 := \mathbb{E}\left[\|\eta_\tau\|^4\right]$ *be the fourth moment of the noise.*

**Assumption 3.** *Stationarity.* $X_0 \sim \pi$, *the stationary distribution corresponding to* $(A^*, \mu)$. *Let* $M_4 := \mathbb{E}\left[\|X_0\|^4\right]$.

Due to Assumption 1, we can show that the law of the iterate $X_T$ from the VAR model defined above converges to a stationary distribution $\pi$ as $T \to \infty$ for arbitrary choice of $X_0$ and has a mixing time of the order $\tau_{\mathsf{mix}} = O\left(\frac{1}{1 - \|A^*\|}\right)$. For simplicity, we will absorb $C_\mu$ into other constants. Finally, we will use $(Z_0, \ldots, Z_T) \sim \mathsf{VAR}(A^*, \mu)$ to mean that $Z_0, \ldots, Z_T$ is a stationary sequence corresponding to $\mathsf{VAR}(A^*, \mu)$. We also note that the covariance matrix under stationarity, $G := \mathbb{E}_{X \sim \pi} X X^\top = \sum_{s=0}^\infty A^{*s} \Sigma (A^{*\top})^s \succeq \Sigma$.

**Remark.** *It is indeed possible to replace Assumption 1 with the weaker condition on the spectral radius of* $A^*$: $\rho(A^*) < 1$. *While our results still hold in this case, the bound might have additional condition number factors. See Section A.1 for more details.*

**Remark.** *The full rank assumption on* $\Sigma$ *is needed for polynomial sample complexity [46].*

**Problem Statement.** Let $(X_0, X_1, \cdots, X_T)$ be sampled from $\mathsf{VAR}(A^*, \mu)$ model for a fixed horizon $T$. Then, the goal is to design and analyze an online algorithm that uses only first order gradient oracle to estimate the system matrix $A^*$. That is, at each time-step $\tau$, we obtain gradient for the transition $(X_\tau, X_{\tau+1})$ and output estimate $A_\tau$. The goal is to ensure that each $A_\tau$ has small estimation error wrt $A^*$; naturally, we would expect better estimation error with increasing $\tau$. We quantify estimation error using the following two loss functions:

1. Parameter error: $\mathcal{L}_{\mathsf{op}}(A; A^*, \mu) = \|A - A^*\|$
2. Prediction error at stationarity: $\mathcal{L}_{\mathsf{pred}}(A; A^*, \mu) := \mathbb{E}_{X_\tau \sim \pi} \|X_{\tau+1} - A X_\tau\|^2$

Note that the problem is equivalent to $d$ linear regression problems, but with *dependent* samples, making it significantly more challenging. Whenever Assumption 1 holds, stationary distribution $\pi$ exists, so the prediction error $\mathcal{L}_{\mathsf{pred}}$ is meaningful. Furthermore: $\mathcal{L}_{\mathsf{pred}}(A) - \mathcal{L}_{\mathsf{pred}}(A^*) = \mathrm{Tr}\left[(A - A^*)^\top (A - A^*) G\right]$ where $G := \mathbb{E}_{X \sim \pi} X X^\top$.

## 3 Algorithm

As mentioned in related works, the standard OLS estimator that minimizes the empirical loss is known to be nearly optimal in the *offline setting* [5]:

$$\hat{A}_{OLS} = \arg\min_A \sum_{\tau=0}^{T-1} \|A X_\tau - X_{\tau+1}\|^2. \tag{3}$$

Note that for least squares loss, one can indeed maintain covariance matrix and residual vector to compute the OLS solution *online*. But such a solution does not work if we have access to only gradients and breaks down even for generalized linear models, whereas as the techniques introduced in this work has been extended to non-linear systems [11].

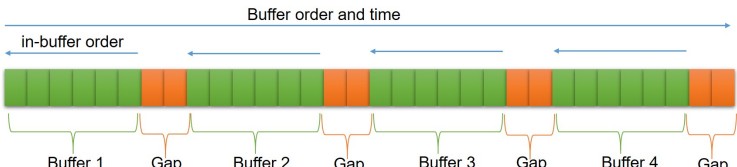

Figure 1: Data Processing Order in SGD − RER. A cell represents a data point. Time goes from left to right, buffers are also considered from left to right. Within each buffer, the data is processed in the reverse order. Gaps ensure that data in successive buffers are approximately independent.

On the other hand, using standard SGD we can obtain update to $A$ efficiently by using gradient at the current point. That is, assuming $A_0 = 0$, we get the following SGD update (for all $\tau \geq 0$):

$$A_{\tau+1} = A_\tau - 2\gamma(A_\tau X_\tau - X_{\tau+1})X_\tau^\top, \tag{4}$$

where $\gamma$ is the stepsize. While SGD is known to be an optimal estimator in certain streaming problems with i.i.d. data, for the $\text{VAR}(A^*, \mu)$ problem the standard SGD does not apply, as samples $(X_\tau, X_{\tau+1})$ and $(X_{\tau+1}, X_{\tau+2})$ are highly correlated. To see why this is the case, let us unroll the recursion for two steps and using Equation (2):

$$A_2 - A^* = (A_0 - A^*)(I - 2\gamma X_0 X_0^\top)(I - 2\gamma X_1 X_1^\top) + 2\gamma\eta_1 X_1^\top + 2\gamma\eta_0 X_0^\top(I - 2\gamma X_1 X_1^\top).$$

Note that the last term does not have 0 mean because $X_1$ depends on $\eta_0$ by Equation (2). Even in the case when $A_0 = A^*$, this means that $\mathbb{E}A_2 \neq A^*$ in general. In fact, in Section 6, we show empirically that SGD with constant step-size converges to a significantly larger error than OLS, even when $T$ is very large. This shows that we cannot naively treat this problem as a collection of $d$ linear regressions. This is consistent with the results in [14, 15] which show a similar behavior for constant step-size SGD with dependent data. Now, one can use techniques like *data drop* that drops a large fraction of points (either explicitly or during the mathematical analysis) from the stream to obtain nearly independent samples [14, 47], but such methods waste a lot of samples and have significantly suboptimal error rate than OLS.

So, the goal is to design a streaming method for the problem of learning dynamical systems that at each time-step $t$ provides an accurate estimate of $A^*$, while also ensuring small space+time complexity. We now present a novel algorithm that addresses the above mentioned problem.

### 3.1 SGD with Reverse Experience Replay

We now discuss a novel algorithm called SGD with Reverse Experience Replay (SGD − RER) that addresses the problem of learning stationary auto-regressive models (or linear dynamical systems) in the streaming setting. Our method is inspired by the experience replay technique [48], used extensively in RL to break temporal correlations between dependent data. We make the following crucial observation. Suppose in Equation (4), instead of processing the samples in the order $(X_1, X_2) \to (X_2, X_3) \to \cdots \to (X_{T-1}, X_T)$, we process it in the reverse order. That is: $(X_{T-1}, X_T) \to (X_{T-2}, X_{T-1}) \to \cdots \to (X_1, X_2)$. Then,

$$A_2 - A^* = (A_0 - A^*)(I - 2\gamma X_{T-1} X_{T-1}^\top)(I - 2\gamma X_{T-2} X_{T-2}^\top) + 2\gamma\eta_{T-2} X_{T-2}^\top$$
$$+ 2\gamma\eta_{T-1} X_{T-1}^\top(I - 2\gamma X_{T-2} X_{T-2}^\top) \tag{5}$$

Now, observe that $(X_{T-2}, X_{T-1})$ are *independent* of $\eta_{T-1}$. Therefore the problematic last term, $2\gamma\eta_{T-1} X_{T-1}^\top(I - 2\gamma X_{T-2} X_{T-2}^\top)$, now has expectation 0. So the updates for *reverse* order SGD would be *unbiased*. This, however, requires us to know all the data points beforehand which is infeasible in the streaming setting. We alleviate this issue by designing SGD − RER, which is the online variant of the above algorithm. SGD − RER uses a buffer of large enough size to store values of consecutive data points and then performs reverse SGD in each of these buffers and then discards this buffer. Experience replay methods also use such (small) buffers of data, but typically samples point randomly from the buffer instead of the reverse order that we propose. We refer to Figure 1 for an illustration of the proposed data processing order.

We present a pseudocode of SGD − RER in Algorithm 1. Note that the algorithm forms non-overlapping buffers of size $S = B + u$. Here $B$ is the actual size of the buffer while $u$ samples are used to interleave between two buffers so that the buffers are *almost independent* of each other. Now

---

**Algorithm 1:** SGD − RER

**Input** : Streaming data $\{X_\tau\}$, horizon $T$, buffer size $B$, buffer gap $u$, bound $R$, tail average
        start: $a$

**Output** : Estimate $\hat{A}_{a,t}$, for all $a < t \leq N - 1$; $N = T/(B + u)$

1 **begin**

2    Step-size: $\gamma \leftarrow \frac{1}{8RB}$, Total buffer size: $S \leftarrow B + u$, Number of buffers: $N \leftarrow T/S$

3    $A_0^0 = 0$ /\*Initialization\*/

4    **for** $t \leftarrow 1$ **to** $N$ **do**

5      Form buffer $\mathsf{Buf}^{t-1} = \{X_0^{t-1}, \ldots, X_{S-1}^{t-1}\}$, where, $X_i^{t-1} \leftarrow X_{(t-1)\cdot S+i}$

6      If $\exists i$, $s.t.$, $\left\|X_i^{t-1}\right\|^2 > R$, then **return** $\hat{A}_{a,t} = 0$

7      **for** $i \leftarrow 0$ **to** $B - 1$ **do**

9        $A_{i+1}^{t-1} \leftarrow A_i^{t-1} - 2\gamma(A_i^{t-1}X_{S-1-i}^{t-1} - X_{S-i}^{t-1})\left(X_{S-1-i}^{t-1}\right)^\top$

10      **end**

11      $A_0^t = A_B^{t-1}$

12      If $t > a$, then $\hat{A}_{a,t} \leftarrow \frac{1}{t-a}\sum_{\tau=a+1}^{t} A_B^{\tau-1}$

13    **end**

14 **end**

---

within a buffer, we perform the usual SGD but with samples read in reverse order. Formally, suppose we index our buffers by $t = 0, 1, 2, \cdots$ and let $S = B + u$ be the total samples (including those that were dropped) in the buffers. Let $N$ denote the total number of buffers in horizon $T$. Within each buffer $t$, we index the samples as $X_i^t$ where $i = 0, 1, 2, \cdots, S - 1$. That is $X_i^t \equiv X_{tS+i}$ is the $i$-th sample in buffer $t$. Similarly $\eta_i^t \equiv \eta_{tS+i}$. Further let $X_{-i}^t \equiv X_{(S-1)-i}^t$. Similarly we set $\eta_{-i}^t \equiv \eta_{(S-1)-i}^t$ Then, the algorithm performs the recursion stated in Line 1 of Algorithm 1. Note that the recursion can also be written as,

$$A_{i+1}^{t-1} - A^* = \left(A_i^{t-1} - A^*\right)\left(I - 2\gamma X_{-i}^{t-1}{X_{-i}^{t-1}}^\top\right) + 2\gamma\eta_{-i}^{t-1}X_{-i}^{t-1}. \tag{6}$$

for $1 \leq t \leq N$ and $0 \leq i \leq B - 1$ with $A_0^t = A_B^{t-1}$ and $A_0^0 = A_0$.

We then ignore the iterates corresponding to first $a$ buffers as part of the *burn-in period*, and output average of the remaining iterates ($t > a$) at each step as that step's estimator (see Line 2 of Algorithm 1). That is, we have the tail-averaged iterate:

$$\hat{A}_{a,t} = \frac{1}{t-a}\sum_{\tau=a+1}^{t} A_B^{\tau-1}. \tag{7}$$

We output the new iterate $\hat{A}_{a,t}$ only at the end of each buffer $t$. At intermediate steps, $(t-1)B+1 \leq \tau \leq tB$, we output $\hat{A}_{a,t-1}$. Also, note that the tail average can be computed in small space and time complexity, by using a running sum of the tail iterates. The update for each point is rank-one, so can be computed in time linear in number of parameters ($O(d^2)$). In the next section, we show that despite using small buffer size $S = B + u$ (that depends logarithmically on $T$), and by throwing away a small constant–independent of *any* problem parameter–fraction of points $u$ in each buffer, we are still able to provide error bound similar to that of OLS.

## 4 Main Results

We now state our main results with leading order terms. For simplicity, we only state the results for the tail average $\hat{A}_{\frac{N}{2},N}$ but a similar result holds for any $\hat{A}_{a,t}$ when $a = \Omega(dB\kappa(G)\log^2 T)$. We refer to Section A for complete statements. Recall the problem setting, and the covariance matrix $G := \mathbb{E}_{X\sim\pi}XX^\top$. Before stating the results, we choose the parameters $B, R, \alpha$ and $u$ as follows, which can be estimated using upper bounds on $\|A^*\|$:

1. $d \leq \mathrm{Poly}(T)$. We use this to bound the norm of covariates in the next item.

2. $\alpha \geq 22$; $R \geq C(\alpha)\frac{\mathrm{Tr}(\Sigma)\log T}{1-\|A^*\|^2} = O(d\tau_{\mathsf{mix}}\log T)$ s.t. $\mathbb{P}\left[\|X_\tau\|^2 \leq R,\ \tau \leq T\right] \geq 1 - \frac{1}{T^\alpha}$. See lemma 9 in appendix.

3. $u \geq \alpha \frac{\log T}{\log\left(\frac{1}{\|A^*\|}\right)} = O(\tau_{\mathsf{mix}} \log T); \quad B = 10u$

For all the results below, we suppose that Assumptions 1, 2 and 3 hold, the stream of samples $X_\tau$ is sampled from $\mathsf{VAR}(A^*, \mu)$ model described in Section 2 and that $R, B, \alpha$ and $u$ are chosen as above. Further we hide some mild conditions on $N$ and $T$.

**Theorem 1** (Informal version of Theorem 5). *Let the step size $\gamma < \min\left(\frac{C}{B\sigma_{\min}(G)}, \frac{1}{8BR}\right)$ for some constant $C$ depending only on $C_\mu$. Then, with probability at least $1 - \frac{1}{T^{100}}$, we have:*

$$\mathcal{L}_{\mathsf{op}}(\hat{A}_{\frac{N}{2},N}, A^*, \mu) \leq C\sqrt{\frac{(d + \log T)\sigma_{\max}(\Sigma)}{T\sigma_{\min}(G)}} + \text{ Lower Order Terms}.$$

**Theorem 2** (Informal version of Theorem 6). *Consider the setting of Theorem 1 but where the step size $\gamma = \min\left(\frac{1}{2R}, \frac{c}{BR}\right)$ for some constant $0 < c < 1$. Then, the following holds:*

$$\mathbb{E}\left[\mathcal{L}_{\mathsf{pred}}(\hat{A}_{\frac{N}{2},N}; A^*, \mu)\right] - \mathsf{Tr}(\Sigma) \leq C\frac{d\,\mathsf{Tr}(\Sigma)}{T} + \text{ Lower Order Terms}$$

*where "lower order" is with respect to $\frac{d}{T}$.*

See Section F.1, Section F.3 for a detailed proof of the parameter error bound and see Section G.1, Section G.2 for a detailed proof of the prediction error bound.

We now make the following observations:

(1) The dominant term in our bound on $\mathcal{L}_{\mathsf{op}}$ (Theorem 1) matches the information theoretically optimal bound (up to logarithmic factors) for the $\mathsf{VAR}(A^*, \mu)$ estimation problem [5] as long as $\|A^*\| \leq 1 - \frac{1}{T^\xi}$ for $\xi \in (0, 1/2)$. Note that despite working with dependent data, leading term in our error bound is nearly independent of mixing time $\tau_{\mathsf{mix}}$. In contrast, most of the existing streaming/SGD style methods for dependent data have strong dependence on $\tau_{\mathsf{mix}}$ [14].

(2) SGD for linear regression with *independent* data [13, 49], but with similar problem setting incurs error $O(\frac{d\,\mathsf{Tr}(\Sigma)}{T})$ for $\mathcal{L}_{\mathsf{pred}}$. So our bound for $\mathsf{SGD} - \mathsf{RER}$ matches the independent data setting bound in the minimax sense.

(3) The space complexity of our method is $O(Bd + d^2)$ where $B = O(\tau_{\mathsf{mix}} \log T)$ is independent of $d$ and only logarithmically dependent on $T$.

(4) **Sparse matrices with known support**: Suppose $A^*$ is known to be sparse and *we know the support* (say by running $L_1$ regularized OLS on a small set of samples). Let $s_j$ denote the sparsity of row $j$ of $A^*$. Then the $\mathsf{SGD} - \mathsf{RER}$ algorithm can be modified to run row by row such that it operates only on the support of row $j$. That is the covariates can be projected onto the support of each row. Then it can be shown that the prediction error is bounded as $O\left(\sum_{j=1}^d \sigma_j^2 s_j / T\right)$ where $\sigma_j^2$ is the $j$-th diagonal entry of $\Sigma$. Note that $\mathsf{SGD} - \mathsf{RER}$ requires only $O(|\mathsf{supp}(A^*)|)$ operations per iteration while applying online version of standard OLS would require $O(d^2)$ operations. In the simple case of $\Sigma = \sigma^2 I$, we note that $G \succeq \sigma^2 I$ and hence the bound for $\mathcal{L}_{\mathsf{pred}}$ becomes $O\left(\frac{|\mathsf{supp}(A^*)|}{T}\right)$. We refer to Section O for a sketch of this extension.

Next, we show that our error bounds are nearly information theoretically optimal. For the lower bound on $\mathcal{L}_{\mathsf{op}}$ we directly use [5, Theorem 2.3].

**Theorem 3.** *Let $\rho < 1$ and $\delta \in (0, 1/4)$. Let $\mu$ be the distribution $\mathcal{N}(0, \sigma^2 I)$. For any estimator $\hat{A} \in \mathcal{F}$, there exists an matrix $A^* \in \mathbb{R}^{d \times d}$ where $A^* = \rho O$ for some orthogonal matrix $O$ such that $|\sigma_{\max}(A^*)| = \rho$ and we have that with probability at least $\delta$:*

$$\|\hat{A} - A^*\| = \Omega\sqrt{\frac{(d + \log(1/\delta))(1 - \rho)}{T}}. \tag{8}$$

Notice that in the setting of Theorem 3, we have $G = \sum_{i=0}^{\infty} \sigma^2 (A^*)^i (A^*)^{i,\top} = \frac{\sigma^2}{1-\rho^2} I$. Therefore, $\sigma_{\min}(G) = \frac{1}{1-\rho^2} \sim \frac{1}{1-\rho}$. The bound in Theorem 1 matches the above minimax bound up to logarithmic factors.

Next we consider the prediction loss. We fix dimension $d$ and horizon $T$ and consider the class of VAR models $\mathcal{M}$ such that Assumptions 1, 2, and 3 hold such that $\mathsf{Tr}(\Sigma(\mu)) = \beta \in \mathbb{R}^+$ be fixed. Let

$\mathcal{F}$ be the class of all estimators for parameter $A^*$ given data $(Z_0, \ldots, Z_T)$. We want to lower bound the minimax error:

$$\mathcal{L}_{\text{minmax}}(\mathcal{M}) := \inf_{f \in \mathcal{F}} \sup_{(A^*, \mu) \in \mathcal{M}} \mathbb{E}_{(Z_t) \sim \text{VAR}(A^*, \mu)} \mathcal{L}_{\text{pred}}(f(Z_0, \ldots, Z_T); A^*, \mu) - \mathcal{L}_{\text{pred}}(A^*; A^*, \mu).$$

**Theorem 4.** *For some universal constant c, we have:*

$$\mathcal{L}_{\text{minmax}}(\mathcal{M}) \geq c\beta(d-1) \min\left(\frac{1}{T}, \frac{1}{d^2}\right), \text{ where } \beta = \text{Tr}(\Sigma(\mu)).$$

Note that the theorem shows that our algorithm is minimax optimal with respect to the prediction loss at stationarity, $\mathcal{L}_{\text{pred}}$. See Section M for a detailed proof of the above lower bound.

## 5   Idea Behind Proofs

In this section, we provide an overview of the key techniques to prove our results. As observed in the discussion following Equation (5), when the data is processed in the reverse order within a buffer, it behaves similar to SGD for linear regression with i.i.d. data. Due to the gaps of size $u$, we can take the buffers to be approximately independent. Therefore, we analyze the algorithm as follows:

1. Analyze reverse order *within* a buffer using the property noted in Equation (5).
2. Treat *different* buffers to be i.i.d. due to gap and present an i.i.d data type analysis.

To execute the proposed proof strategy, we introduce the following technical notions:

**Coupled Process.**    For the real data points $(X_\tau)$, the points in different buffers are *weakly* dependent. In order to make the analysis straight forward, we introduce the *fictitious* coupled process $\tilde{X}_\tau$ such that $\left\| \tilde{X}_\tau - X_\tau \right\| \lesssim \frac{1}{T^\alpha}$ for large enough $\alpha$, for every data point $X_\tau$ used by $\mathsf{SGD} - \mathsf{RER}$. We have the additional property that the successive buffers are actually independent for this coupled process. We refer to Definition 1 in the appendix for the construction of the coupled process $\tilde{X}_\tau$.

Suppose we run $\mathsf{SGD} - \mathsf{RER}$ with the coupled process $\tilde{X}_\tau$ instead of $X_\tau$ to obtain the coupled iterates $\tilde{A}_i^t$. We can then show that $\tilde{A}_i^t \approx A_i^t$. Thus it suffices analyze the coupled iterates $\tilde{A}_i^t$. We refer to Sections B and C for the details.

**Bias Variance Decomposition.**    We consider the standard bias variance decomposition with individual buffers as the basic unit as opposed to individual data points.    We refer to Section D for the details.    We decompose the error in the iterates into the bias part $\left(\tilde{A}_B^{t-1,b} - A^*\right)$ $=$ $(A_0 - A^*) \prod_{s=0}^{t-1} \tilde{H}_{0,B-1}^s$ and the variance part $\left(\tilde{A}_B^{t-1,v}\right)$ $=$ $2\gamma \sum_{r=1}^t \sum_{j=0}^{B-1} \eta_{-j}^{t-r} \tilde{X}_{-j}^{t-r,\top} \tilde{H}_{j+1,B-1}^{t-r} \prod_{s=r-1}^1 \tilde{H}_{0,B-1}^{t-s}$ where the matrices $\tilde{H}_{0,B-1}^s = \prod_{i=0}^{B-1} \left(I - 2\gamma \tilde{X}_{-i}^s \tilde{X}_{-i}^{s,\top}\right)$ are the independent 'contraction' matrices associated with each buffer $s$. This result in the geometric decay of the initial distance between $(A_0 - A^*)$. The variance part is due to the inherent noise present in the data. In Section F.1 we first establish the exponential decay of the 'bias'. We then consider the second moment of the variance term. Observe that the distinct terms in the expression for $\left(\tilde{A}_B^{t-1,v}\right)$ are uncorrelated either due to reverse order *within* a buffer as noted in Equation (5) or due to independence between the data in distinct buffers (due to coupling). This allows us to split the second moment into diagonal terms with non-zero mean and cross terms with zero mean. Diagonal terms are analyzed via a recursive argument in Claim 1 and the following discussion in order to remove dependence on mixing time factors. The analysis for parameter recovery (the result of Theorem 2) is similar but we bound the relevant exponential moments using sub-Gaussianity of the noise sequence $\eta_t$ to obtain high-probability bounds which when combined with standard $\epsilon$-net arguments give us guarantees for the operator norm error $\mathcal{L}_{\text{op}}$.

**Averaged Iterates.**    We then combine the bias and variance bounds obtained for individual iterates in Section F.1 to analyze the tail averaged output. Using techniques standard in the analysis of SGD for linear regression, we finally show that this averaging leads error rates of the order $\frac{d^2}{T}$. We refer to Sections E (for parameter recover) and G (for prediction error) for the detailed results.

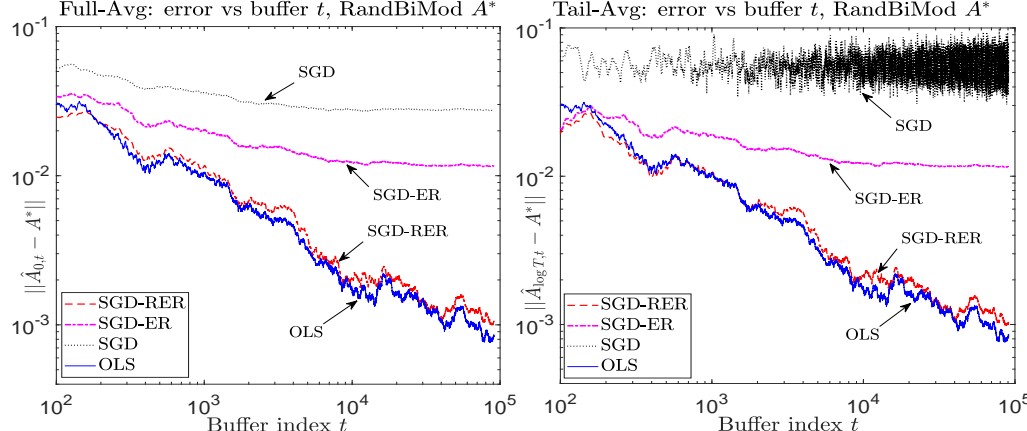

Figure 2: Gaussian $\text{VAR}(A^*, \mu)$: Parameter error for tail averaged and full average iterates of $\text{SGD} - \text{RER}$ and baselines. $\text{SGD} - \text{RER}$ and OLS incur similar parameter error, while error incurred by SGD and $\text{SGD} - \text{ER}$ saturate at significantly higher level, indicating non-zero bias. The parameters used are $\rho = 0.9$, $d = 5$, $T = 10^7$, $B = 100$, $u = 10$. $R$ is estimated and $\gamma = 1/2R$.

**Picking the Step Sizes and Conditioning.** Due to the auto-regressive nature of the data generation, the iterates can grow to be of the size $O(\frac{d}{1-\rho})$. The step sizes need to be set small enough so that the $\gamma \|X_\tau X_\tau^\top\| \leq 1$ in order for the $\text{SGD} - \text{RER}$ iterations to not diverge to infinity. In the statement of Theorem 2, we condition on the event where $\|X_\tau\|^2$ are all bounded by a sufficiently large number $R$ for every $\tau$ in order to ensure this property. The relevant events where the norm is bounded are defined in Section B. Conditioning on these events results in previously zero mean terms to be not zero mean. Routine calculations using triangle inequality and Cauchy-Schwarz inequality ensure that the means are still of the order $\frac{1}{T^\alpha}$ for any fixed constant $\alpha > 0$. Furthermore, we actually require step sizes such that $\gamma \left\| \sum_{\tau \in \text{Buffer}} X_\tau X_\tau^\top \right\| \leq 1$ to show exponential contraction of $\tilde{H}_{0,B-1}^s$ matrices due to the Grammian $G$ as described next.

**Probabilistic Results.** We establish some properties of $\tilde{H}_{0,B-1}^s$, which are products of dependent random matrices in Section L. Specifically we refer to Lemmas 28, 29, 30, and 31 which establish that $\left\| \prod_{s=0}^{t-1} \tilde{H}_{0,B-1}^s \right\| \lesssim (1 - \gamma B \sigma_{\min}(G))^t$ with high probability.

## 6 Experiments

In this section, we compare performance of our $\text{SGD} - \text{RER}$ method on synthetic data against the performance of standard baselines OLS and SGD, along with $\text{SGD} - \text{ER}$ method that applies standard experience replay technique, but where points from a buffer are sampled *randomly*.

**Synthetic data**: We sample data from $\text{VAR}(A^*, \mu)$ with $X_0 = 0$, $\mu \sim \mathcal{N}(0, \sigma^2 I)$ and $A^* \in \mathbb{R}^{d \times d}$ is generated from the "RandBiMod" distribution. That is, $A^* = U \Lambda U^\top$ with random orthogonal $U$, and $\Lambda$ is diagonal with $\lceil d/2 \rceil$ entries on diagonal being $\rho$ and the remaining diagonal entries are set to $\rho/3$. We set $d = 5$, $\rho = 0.9$ and $\sigma^2 = 1$. We fix a horizon $T = 10^7$ and set the buffer size as $B = 100$ and $u = 10$. To estimate $R$ from the data, we use the first $\lfloor 2 \log T \rfloor = 32$ samples and set $R$ as the sum of the norms of these samples. We let the stepsize to be $\gamma = \frac{1}{2R}$ which is *aggressive* compared to our theorems. We start the $\text{SGD} - \text{RER}$ and other $SGD$-like algorithms from the second buffer onward.

For tail averaging, as described in algorithm 1, we ignore the first $\lfloor \log T \rfloor = 16$ buffers, and maintain a running tail average at the end of each of the subsequent buffers. In figure 2, we plot the parameter errors $\left\| \hat{A}_{\log T, t} - A^* \right\|$ and $\left\| \hat{A}_{0,t} - A^* \right\|$ versus the buffer index $t$ as the algorithm runs for horizon $T$. For OLS, we include samples in the first buffer as well (which were used for estimating $R$). Clearly, $\text{SGD} - \text{RER}$ has very similar performance as that of OLS whereas $\text{SGD} - \text{ER}$ and SGD seem to display residual bias for the chosen step-size (which is logarithmic in the horizon $T$) and buffer lengths. We also observe a similar behavior when we choose $A^* = \rho I$.

# 7  Conclusion

In this paper, we studied the problem of linear system identification in streaming setting and provided an efficient algorithm (SGD − RER). We proved that SGD − RER achieves nearly minimax optimal error rate, both in terms of parameter error as well as prediction error. Furthermore, using experiments, we validated that standard SGD as well as SGD with experience replay can have large bias error. Our algorithm and analysis demonstrates that the knowledge of dependency structure can aid us in designing accurate algorithms for dependent data.

This work opens up a myriad of open questions about learning from dependent data in general and Markov processes in particular. Our work currently assumes a specific Markovian dependency structure – extending the intuition and techniques to handle more general data dependencies is an interesting open question. Further, our work does not address the question of recovering a sparse system matrix with unknown sparsity pattern. So online learning of such linear dynamical systems with (unknown) sparsity pattern or low-rank structure is an exciting question with applications to domains like bioinformatics. Moreover, even in our linear setting, extending SGD − RER to the situation of partially observed states with or without control inputs would be another direction to pursue. Finally, it would be interesting to understand how the techniques introduced in this work perform in practical RL settings where learning with data from Markov processes is essential.

## Acknowledgments and Disclosure of Funding

D.N. was supported in part by NSF grant DMS-2022448.
S.S.K was supported in part by Teaching Assistantship (TA) from EECS, MIT.
Part of this work was done when S.S.K was visiting Microsoft Research Lab India Pvt Ltd during summer 2020.

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
