## Organization of the appendix

We provide a map of the results in the appendix.

1. In section A we provide formal statements of theorems 1 and 2. We also discuss the more general spectral gap condition $\max_i |\lambda_i(A)| < 1$ instead of the stronger condition $\|A\| < 1$ and its impact on the results.
2. In section B we construct the coupled process $\tilde{X}_t$ and setup notations used in the rest of the paper. The coupled process has the additional property that the successive buffers are independent.
3. In section C we show that the $\mathsf{SGD-RER}$ iterates generated using the coupled process are close to ones generated by the actual data. After this, we only deal with the coupled iterates.
4. In section D we provide the bias-variance decomposition
5. In section E we provide the proof of the parameter error bound of theorem 1. Required intermediary results are discussed in section L.
6. In section F we present the bounds on the bias and variance terms separately (for last and average iterates), which are necessary to prove theorem 6. Most of the proofs are relegated to sections H, I, J, K and N.
7. In section G we prove theorem 2.
8. In section M, we prove the lower bounds for the prediction error given in theorem 4.
9. In section O we discuss the scenario of $\mathsf{VAR}(A^*, \mu)$ where $A^*$ is sparse with known sparsity pattern. We provide a proof sketch of the bound on prediction error in terms of sparsity.

## A Formal Results and Proof Sketch

In this Section, we formally state the full results and sketch the outline of our proof. Recall the definitions of $\mathcal{L}_{\mathsf{op}}$ and $\mathcal{L}_{\mathsf{pred}}$ from section 2. For all the theorems below, we suppose that Assumptions 1, 2 and 3 hold. Assume that $u, \gamma, B, \alpha$ and $R$ are as chosen in section 4.

Let $t > a$ and let $\hat{A}_{a,t}$ be the tail averaged output of $\mathsf{SGD-RER}$ after buffer $t-1$. Further let $T^{\alpha/2} > cd\kappa(G)$.

**Theorem 5.** *Suppose we pick the step size* $\gamma = \min\left(\frac{C}{B\sigma_{\min}(G)}, \frac{1}{8BR}\right)$ *for some constant $C$ depending only on $C_\mu$. Then, there are constants $C, c_i > 0, 0 \le i \le 4$ such that if $a > c_0\left(d + \alpha \log T\right)$ then with probability at least $1 - \frac{C}{T^\alpha}$, we have:*

$$\mathcal{L}_{\mathsf{op}}(\hat{A}_{a,t}, A^*, \mu) \le c_1\sqrt{\frac{(d + \alpha \log T)\sigma_{\max}(\Sigma)}{(t-a)B\sigma_{\min}(G)}} + \beta_b \|A_0 - A^*\| + c_4\frac{T^2}{B^2}\|A^{*u}\| \qquad (9)$$

*where*

$$\beta_b = c_3 \frac{d\kappa(G)\log T}{t-a} e^{-c_2\frac{a}{d\kappa(G)\log T}} \qquad (10)$$

The techniques for the proof is developed in Section L and the Theorem 5 is proved in Section E.

**Theorem 6.** *Let $R, B, u, \alpha$ be chosen as in section 4. Let $\gamma = \frac{c}{4RB} \le \frac{1}{2R}$ for $0 < c < 1$. Then there are constants $c_1, c_2, c_3, c_4 > 0$ such that for $T^{\alpha/2} > c_1\frac{\sqrt{M_4}}{\sigma_{\min}(G)}$ the expected prediction loss $\mathcal{L}_{\mathsf{pred}}$ is bounded as*

$$
\begin{aligned}
\mathbb{E}\left[\mathcal{L}_{\mathsf{pred}}(\hat{A}_{a,t}; A^*, \mu)\right] - \mathrm{Tr}(\Sigma) \le\ & c_2\left[\frac{d\,\mathrm{Tr}(\Sigma)}{B(t-a)} + \frac{d^2\sigma_{\max}(\Sigma)}{B(t-a)}\frac{\sqrt{\kappa(G)}}{B}\right] + \\
& c_3\left[\frac{d^2\sigma_{\max}(\Sigma)}{B^2(t-a)^2}(\kappa(G))^{3/2}dB\log T + \right. \\
& \beta_b\,\mathrm{Tr}(G)\|A_0 - A^*\|^2 + \\
& \left.\left(\frac{T^3}{B^3}\|A^{*u}\| + \frac{d\sigma_{\max}(\Sigma)}{R}\frac{T^2}{B^2}\frac{1}{T^{\alpha/2}}\right)\mathrm{Tr}(G)\right]
\end{aligned}
$$
$$\qquad (11)$$

*where $\beta_b$ is defined in (10).*

The above theorem is proven only for the case $t = N$. The proof for general $t$ is almost the same. The proof follows by first considering $\mathbb{E}\left[\mathcal{L}_{\text{pred}}(\hat{A}_{a,N}; A^*, \mu)1\left[\mathcal{D}^{0,N-1}\right]\right]$ ($\mathcal{D}^{0,N-1}$ is defined in B.1) and using theorem 20 and theorem 21 along with lemma 12 in the appendix sections G.1, G.2 and C. Then noting that if the norm of any of the covariates $X_t$ exceed $\sqrt{R}$ the algorithm returns the zero matrix we have that $\mathbb{E}\left[\mathcal{L}_{\text{pred}}(\hat{A}_{a,N}; A^*, \mu)1\left[\mathcal{D}^{0,N-1,C}\right]\right] \leq c\|A^*\|\operatorname{Tr}(G)\frac{1}{T^\alpha}$.

**Remark.**

(1) In theorem 6 the term $\frac{d^2\sigma_{\max}(\Sigma)}{B(t-a)}\frac{\sqrt{\kappa(G)}}{B}$ is strictly a lower order term compared to $\frac{d\operatorname{Tr}(\Sigma)}{B(t-a)}$ when $\|A^*\| < c_0 < 1$. To see this note that $\sigma_{\max}(G) \leq \frac{\sigma_{\max}(\Sigma)}{1-\|A^*\|^2}$ and $\sigma_{\min}(G) \geq \sigma_{\min}(\Sigma)$. Hence $\kappa(G) \leq \frac{\kappa(\Sigma)}{1-\|A^*\|^2} = O(\tau_{\text{mix}}\kappa(\Sigma))$. By the choice of $B$ in the section 4 we see that $\frac{\sqrt{\kappa(G)}}{B} = o(1)$ and it *does not depend on condition number of $A^*$.*

(2) If $a = \Omega\left(d\kappa(G)\left(\log T\right)^2\right)$ the $\beta_b$ is a lower order term. Further choosing $u$ and $\alpha$ as in section 4 we see that the terms depending on $\|A^{*u}\|$ and $\frac{1}{T^{\alpha/2}}$ are strictly lower order.

(3) Thus for the choice of $a$ as in the previous remark such that $a < (1+c)t$ (for some $c > 0$), we get minimax optimal rates: $\frac{d\operatorname{Tr}(\Sigma)}{Bt}$ for $\mathcal{L}_{\text{pred}}$ and up to log factors, $\sqrt{\frac{d\sigma_{\max}(\Sigma)}{T\sigma_{\min}(G)}}$ for $\mathcal{L}_{\text{op}}$

### A.1 Spectral Gap Condition

In Assumption 1, we could have used the more general spectral radius condition $\rho(A^*) = \sup_i |\lambda_i(A^*)| < 1$ rather than the one on the operator norm. We have the Gelfand formula for spectral radius which shows that $\lim_{k\to\infty}\|A^{*k}\|^{1/k} = \rho(A^*)$. Now, if $A^*$ is such that $\rho(A^*) < 1$ but $\|A^*\| > 1$ (a case studied by [5]), then we need to make $u$ as large as $Cd\log T$ which would lead to a relatively large buffer size $B$ of $d\log T$. To see this, we verify the proof by [50] (by replacing $A$ with $\frac{A}{\|A\|}$ and $\rho(A)$ with $\frac{\rho}{\|A\|}$ in the proof) to show that $\|A^{*k}\| \leq (2k\|A^*\|)^d\rho^{k-d}$ whenever $k \geq d$. Therefore, in the worst case, we can pick $u = O\left(\left(\log\left(T\sigma_{\max}(G)\right) + d\log d\|A\|\right)/\log 1/\rho\right)$.

In the case of $\rho < 1$ but $\|A^*\| > 1$, $\kappa(G)$ can grow super linearly in $d$. For instance, consider $A^*$ to be nilpotent of order $d$ (i.e. $A^{*d-1} \neq 0$ but $A^{*d} = 0$). Here $\sigma_{\max}(G)$ can grow like $\|A^*\|^d$. So we need exponentially (in $d$) many samples for bias decay. However, in many cases of interest (ex: symmetric matrices, normal matrices etc) the spectral radius is the same as the operator norm.

## B  Basic Lemmas and Notations

Since the covariates $\{X_\tau\}_{\tau \leq T}$ are correlated, we will introduce a coupled process such that we have independence across buffers and that Euclidean distance between the covariates of the original process and the coupled process can be controlled.

**Remark.** *Note that the coupled process is imaginary and we do not actually run the algorithm with the coupled process. We construct it to make the analysis simple by first analyzing the algorithm with the imaginary coupled process and then showing that the output of the actual algorithm cannot deviate too much when run with the actual data.*

**Definition 1** (Coupled process). *Given the covariates $\{X_\tau : \tau = 0, 1, \cdots T\}$ and noise $\{\eta_\tau : \tau = 0, 1, \cdots, T\}$, we define $\{\tilde{X}_\tau : \tau = 0, 1, \cdots, T\}$ as follows:*

1. *For each buffer $t$ generate, independently of everything else, $\tilde{X}_0^t \sim \pi$, the stationary distribution of the $\mathsf{VAR}(A^*, \mu)$ model.*
2. *Then, each buffer has the same recursion as eq (2):*

$$\tilde{X}_{i+1}^t = A^*\tilde{X}_i^t + \eta_i^t, \ i = 0, 1, \cdots S-1, \tag{12}$$

*where the noise vectors as same as in the actual process $\{X_\tau\}$.*

With this definition, we have the following lemma:

**Lemma 7.** *For any buffer t,* $\|X_i^t - \tilde{X}_i^t\| \leq \|A^{*i}\|\|X_0^t - \tilde{X}_0^t\|$, *a.s.. That is,*

$$\|X_i^t X_i^{t^T} - \tilde{X}_i^t \tilde{X}_i^{t^T}\| \leq 2\|X\|\|X_i^t - \tilde{X}_i^t\| \leq (2\|X\|)^2\|A^{*i}\|. \tag{13}$$

*Here* $\|X\|$ *denotes* $\sup_{\tau \leq T}\|X_\tau\|$.

**Lemma 8.** *Suppose* $\mu$ *obeys Assumption 2 and* $A^*$ *obeys Assumption 1. Suppose* $X \sim \pi$, *which is the stationary distribution of* $\mathsf{VAR}(A^*, \mu)$. $\langle X, x \rangle$ *has mean* 0 *and is sub-Gaussian with variance proxy* $C_\mu x^\top G x$

*Proof.* Suppose $\eta_1, \ldots, \eta_n, \ldots$ is a sequence of i.i.d random vectors drawn from the noise distribution $\mu$. We consider the partial sums $\sum_{i=0}^{n} A^{*i}\eta_i$. Call the law of this to be $\pi_n$. Clearly $\pi_n$ converges in distribution to $\pi$ as $n \to \infty$ since $\pi_n$ is the law of the $n+1$-th iterate of $\mathsf{VAR}(A^*, \mu)$ chain stated at $X_0 = 0$. By Skorokhod representation theorem, we can define the infinite sequence $X^{(1)}, \ldots, X^{(n)}, \ldots$, and another random variable $X$ such that $X^{(i)} \sim \pi_i$, $X \sim \pi$ and $\lim_{n\to\infty} X^{(n)} = X$ a.s. Define $G_n = \sum_{i=0}^{n} A^{*i}\Sigma(A^{*i})^T$. Clearly, $G_n \preceq G = \sum_{i=0}^{\infty} A^{*i}\Sigma(A^{*i})^T$. A simple evaluation of Chernoff bound for $\langle X^{(n)}, x \rangle$ by decomposing it into the partial sum of noises shows that:

$$\mathbb{E}\exp(\lambda\langle X^{(n)}, x\rangle) \leq \exp\left(\frac{\lambda^2 C_\mu}{2}\langle x, G_n x\rangle\right) \leq \exp\left(\frac{\lambda^2 C_\mu}{2}\langle x, Gx\rangle\right)$$

We now apply Fatou's lemma, since $X^{(n)} \to X$ almost surely, to the inequality above to conclude that:

$$\mathbb{E}\exp(\lambda\langle X, x\rangle) \leq \exp\left(\frac{\lambda^2 C_\mu}{2}\langle x, Gx\rangle\right).$$

$\square$

Hence $\langle x, X_t \rangle$ is subgaussian with mean 0 and variance proxy $C_\mu\sigma_{\max}(G)\|x\|^2$. This will provide uniform variance for all $x$ such that $\|x\|^2 = 1$.

From subgaussianity and standard $\epsilon$-net argument we have the following lemma.

**Lemma 9.** *For any* $\beta > 0$ *there is a constant* $c > 0$ *such that*

$$\mathbb{P}\left[\exists\tau \leq T : \|X_\tau\|^2 > c\operatorname{Tr}G\log T\right] \leq \frac{d}{T^\beta} \tag{14}$$

*Thus as long as* $d < \operatorname{Poly}(T)$, *for every* $\alpha > 0$ *there is a* $c > 0$ *such that*

$$\mathbb{P}\left[\exists\tau \leq T : \|X_\tau\|^2 > c\operatorname{Tr}G\log T\right] \leq \frac{1}{T^\alpha} \tag{15}$$

## B.1 Notations

Before we analyze this algorithm, we define some notations. We work in a probability space $(\Omega, \mathcal{F}, \mathbb{P})$ and all the random elements are defined on this space. We define the following notations:

$$X_{-i}^t = X_{(S-1)-i}^t,\ 0 \leq i \leq S-1, \quad G = \sum_{s=0}^{\infty} A^{*s}\Sigma(A^{*\top})^s, \quad G_t = \sum_{s=0}^{t-1} A^{*s}\Sigma(A^{*\top})^s,$$

$$\tilde{P}_i^t = \left(I - 2\gamma\tilde{X}_i^t\tilde{X}_i^{t,\top}\right), \quad \tilde{H}_{i,j}^t = \begin{cases} \prod_{s=i}^{j}\tilde{P}_{-s}^t & i \leq j \\ I & i > j \end{cases},$$

$$\hat{\gamma} = 4\gamma(1 - \gamma R), \quad \mathcal{C}_{-j}^t = \left\{\|X_{-j}^t\|^2 \leq R\right\}, \quad \tilde{\mathcal{C}}_{-j}^t = \left\{\|\tilde{X}_{-j}^t\|^2 \leq R\right\},$$

$$\mathcal{D}_{-j}^t = \left\{\|X_{-i}^t\|^2 \leq R : j \leq i \leq B-1\right\} = \bigcap_{i=j}^{B-1}\mathcal{C}_{-i}^t,$$

$$\mathcal{D}^{s,t} = \begin{cases} \bigcap_{r=s}^{t}\mathcal{D}_{-0}^r & s \leq t \\ \Omega & s > t \end{cases}, \quad \tilde{\mathcal{D}}_{-j}^t = \left\{\|\tilde{X}_{-i}^t\|^2 \leq R : j \leq i \leq B-1\right\} = \bigcap_{i=j}^{B-1}\tilde{\mathcal{C}}_{-i}^t,$$

$$\tilde{\mathcal{D}}^{s,t} = \begin{cases} \bigcap_{r=s}^{t}\tilde{\mathcal{D}}_{-0}^r & s \leq t \\ \Omega & s > t \end{cases}, \quad \hat{\mathcal{D}}_{-j}^t = \mathcal{D}_{-j}^t \cap \tilde{\mathcal{D}}_{-j}^t, \quad \hat{\mathcal{D}}^{s,t} = \mathcal{D}^{s,t} \cap \tilde{\mathcal{D}}^{s,t}.$$

Lastly $c$ and $c_i$ for $i = 0, 1, \cdots$ denote absolute constants that can change from line to line in the proofs.

## C   Initial Coupling

We consider the coupled process introduced in Definition 1 and run $\mathsf{SGD} - \mathsf{RER}$ with the fictitious coupled process $\hat{X}_\tau$ instead of $X_\tau$ in order to obtain the iterates $\tilde{A}_i^t$ instead of $A_i^{t-1}$. Using Lemma 7, we can show that $\tilde{A}_i^{t-1} \approx A_i^{t-1}$. It is easier to analyze the iterates $\tilde{A}_i^t$ due to buffer independence.

**Lemma 10.** *Let $\gamma \leq \frac{1}{2R}$. Under the event $\mathcal{D}^{0,N-1}$, for every $t \in [N]$ and $0 \leq i \leq B - 1$ we have:*

$$\|A_i^{t-1}\| \leq 2\gamma RT \,.$$

**Lemma 11.** *Suppose $\gamma < \frac{1}{2R}$. Under the event $\hat{\mathcal{D}}^{0,N-1}$ we have for every $t \in [N]$ and $0 \leq i \leq B-1$.*
$$\|A_i^{t-1} - \tilde{A}_i^{t-1}\| \leq (16\gamma^2 R^2 T^2 + 8\gamma RT) \|A^{*u}\|$$

We can now just analyze the iterates $\tilde{A}_i^{t-1}$ and then use Lemma 11 to infer error bounds for $A_i^{t-1}$. Henceforth, we will only consider $\tilde{A}_i^{t-1}$.

**Lemma 12.** *Consider the algorithmic iterates obtained from the actual process and coupled process $(A_j^t)$ and $(\tilde{A}_j^t)$. Then*

$$\mathbb{E}\left[\left(A_j^{t-1} - A^*\right)^\top \left(A_j^{t-1} - A^*\right) \mathbb{1}\left[\mathcal{D}^{0,t-1}\right]\right] \preceq \mathbb{E}\left[\left(\tilde{A}_j^{t-1} - A^*\right)^\top \left(\tilde{A}_j^{t-1} - A^*\right) \mathbb{1}\left[\tilde{\mathcal{D}}^{0,t-1}\right]\right]$$

$$+ c\left(\gamma^3 R^3 T^3 \|A^{*u}\| + \gamma^2 d\sigma_{\max}(\Sigma)RT^2 \frac{1}{T^{\alpha/2}}\right) I \tag{16}$$

*for some constant $c$. Furthermore, the same conclusion holds for the average iterates. That is let*

$$\hat{A}_{a,N} = \frac{1}{N-a} \sum_{t=a+1}^N A_B^{t-1}$$

$$\hat{\tilde{A}}_{a,N} = \frac{1}{N-a} \sum_{t=a+1}^N \tilde{A}_B^{t-1}$$

*Then*

$$\mathbb{E}\left[\left(\hat{A}_{a,N} - A^*\right)^\top \left(\hat{A}_{a,N} - A^*\right) \mathbb{1}\left[\mathcal{D}^{0,N-1}\right]\right]$$

$$\preceq \mathbb{E}\left[\left(\hat{\tilde{A}}_{a,N} - A^*\right)^\top \left(\hat{\tilde{A}}_{a,N} - A^*\right) \mathbb{1}\left[\tilde{\mathcal{D}}^{0,N-1}\right]\right]$$

$$+ c\left(\gamma^3 R^3 T^3 \|A^{*u}\| + \gamma^2 d\sigma_{\max}(\Sigma)RT^2 \frac{1}{T^{\alpha/2}}\right) I \tag{17}$$

**Remark.** *The above lemma holds as is when $A_j^{t-1}$, $\tilde{A}_j^{t-1}$ is replaced by $A_j^{t-1,v}$, $\tilde{A}_j^{t-1,v}$ respectively.*

We refer to Section N for the proofs of the three lemmas.

## D   Bias Variance Decomposition

Now, we can unroll the recursion in (6), but for the coupled iterates $\tilde{A}_i^{t-1}$ as

$$\tilde{A}_B^{t-1} - A^* = \left(\tilde{A}_B^{t-1,b} - A^*\right) + \left(\tilde{A}_B^{t-1,v}\right), \tag{18}$$

where

$$\left(\tilde{A}_B^{t-1,b} - A^*\right) = (A_0 - A^*)\prod_{s=0}^{t-1} \tilde{H}_{0,B-1}^s \tag{19}$$

is the *bias* term, and the *variance* term is given by:

$$\left(\tilde{A}_B^{t-1,v}\right) = 2\gamma \sum_{r=1}^{t} \sum_{j=0}^{B-1} \eta_{-j}^{t-r} \tilde{X}_{-j}^{t-r,\top} \tilde{H}_{j+1,B-1}^{t-r} \prod_{s=r-1}^{1} \tilde{H}_{0,B-1}^{t-s} \tag{20}$$

Here we use the convention that whenever $r = 1$, the product $\prod_{s=r-1}^{1}$ is empty i.e, equal to 1. The 'bias' term is obtained when the noise terms are set to 0, and captures the movement of the algorithm towards the optimal $A^*$ when we set the initial iterate far away from it. The 'variance' term $\left(A_B^{t,v} - A^*\right)$ capture the uncertainty due to the inherent noise in the data. Our main goal is to understand the performance (estimation and prediction) of the tail-averaged iterates output by $\mathsf{SGD} - \mathsf{RER}$. Here, we consider just the last iterate, but the same technique applies to all the outputs of $\mathsf{SGD} - \mathsf{RER}$. That is, $\hat{\tilde{A}}_{a,N} = \frac{1}{N-a} \sum_{t=a+1}^{N} \tilde{A}_B^{t-1}$, for $a = \lceil \theta N \rceil$ with $0 < \theta < 1$. We can decompose the above into bias and variance as: $\hat{\tilde{A}}_{a,N} = \hat{\tilde{A}}_{a,N}^v + \hat{\tilde{A}}_{a,N}^b$, with,

$$\hat{\tilde{A}}_{a,N}^v = \frac{1}{N-a} \sum_{t=a+1}^{N} \tilde{A}_B^{t-1,v} \tag{21}$$

$$\hat{\tilde{A}}_{a,N}^b = \frac{1}{N-a} \sum_{t=a+1}^{N} \tilde{A}_B^{t-1,b}. \tag{22}$$

Similarly, we can decompose the final error into 'bias' and 'variance' as in Lemma 13 below.

**Lemma 13** (Bias-Variance Decomposition). *We have the following decomposition:*

$$\left(\tilde{A}_B^{t-1} - A^*\right)^\top \left(\tilde{A}_B^{t-1} - A^*\right) \preceq 2\left[\left(\tilde{A}_B^{t-1,b} - A^*\right)^\top \left(\tilde{A}_B^{t-1,b} - A^*\right) + \right.$$
$$\left.\left(\tilde{A}_B^{t-1,v}\right)^\top \left(\tilde{A}_B^{t-1,v}\right)\right].$$

# E   Parameter Error Bound–Proof of Theorem 5

In this section, we formally prove the bounds on $\mathcal{L}_{\mathsf{op}}(; A^*, \mu)$, by combining several operator norm inequalities that we prove in Section L. As mentioned previously, we will just focus on the algorithmic iterates from the coupled process $(\tilde{A}_j^{t-1})$. Recall the output $\tilde{A}_B^{t-1}$ after the $t-1$-th buffer from Equation (18). For any initial buffer index $a \in \{0, 1, \ldots, N-1\}$, the tail averaged output of our algorithm is:

$$\hat{\tilde{A}}_{a,N} := \frac{1}{N-a} \sum_{t=a+1}^{N} \tilde{A}_B^{t-1}.$$

Recall the quantities $\tilde{A}_B^{t-1,v}$ and $\tilde{A}_B^{t-1,b}$ as defined in (19) and (20). We can use this decomposition to write:

$$\hat{\tilde{A}}_{a,N} - A^* = \hat{\tilde{A}}_{a,N}^b - A^* + \hat{\tilde{A}}_{a,N}^v.$$

Here $\hat{\tilde{A}}_{a,N}^b - A^* := \frac{1}{N-a} \sum_{t=a+1}^{N} \left(\tilde{A}_B^{t-1,b} - A^*\right)$ denotes the bias part and $\hat{\tilde{A}}_{a,N}^v := \frac{1}{N-a} \sum_{t=a+1}^{N} \left(\tilde{A}_B^{t-1,v}\right)$ denotes the variance part.

## E.1   Variance

Note that

$$\hat{\tilde{A}}_{a,N}^v = \frac{N}{N-a} \left(\hat{\tilde{A}}_{0,N}^v\right) - \frac{a}{N-a} \left(\hat{\tilde{A}}_{0,a}^v\right) \tag{23}$$

Now, we apply Theorem 33 with $\delta$ in the definition of $\tilde{\mathcal{M}}^{0,N-1}$ to be $\frac{1}{T^v}$ for some fixed $v \geq 1$. We conclude that conditioned on the event $\tilde{\mathcal{M}}^{0,N-1} \cap \tilde{\mathcal{D}}^{0,N-1}$, with probability at least $1 - \frac{1}{T^v}$, we have:

$$\|\hat{\tilde{A}}_{0,N}^v\| \leq C\sqrt{\frac{\gamma(d + \upsilon \log T)^2 \sigma_{\max}(\Sigma)}{N}} + C\sqrt{\frac{(d + \upsilon \log T)\sigma_{\max}(\Sigma)}{NB\sigma_{\min}(G)}} \ .$$

Similarly, applying Theorem 33 with $N = a$ shows that with probability at least $1 - \frac{1}{T^\upsilon}$ conditioned on the event $\tilde{\mathcal{M}}^{0,N-1} \cap \tilde{\mathcal{D}}^{0,N-1}$:

$$\|\hat{\tilde{A}}_{0,a}^v\| \leq C\sqrt{\frac{\gamma(d + \upsilon \log T)^2 \sigma_{\max}(\Sigma)}{a}} + C\sqrt{\frac{(d + \upsilon \log T)\sigma_{\max}(\Sigma)}{aB\sigma_{\min}(G)}} \ .$$

Here, the constant $C$ depends only on $C_\mu$. We also note that when we pick $\gamma BR \leq C_0$ where $R \gtrsim \text{Tr}(G) + \upsilon \log T$, the first term in the equations above becomes smaller than the second term. Therefore, under this assumption we can simplify the expressions to:

$$\|\hat{\tilde{A}}_{0,N}^v\| \leq C\sqrt{\frac{(d + \upsilon \log T)\sigma_{\max}(\Sigma)}{NB\sigma_{\min}(G)}} \ . \tag{24}$$

$$\|\hat{\tilde{A}}_{0,a}^v\| \leq C\sqrt{\frac{(d + \upsilon \log T)\sigma_{\max}(\Sigma)}{aB\sigma_{\min}(G)}} \ . \tag{25}$$

Applying Equations (24) and (25) to Equation (23) we conclude that conditioned on the event $\tilde{\mathcal{M}}^{0,N-1} \cap \tilde{\mathcal{D}}^{0,N-1}$, with probability at least $1 - \frac{2}{T^\upsilon}$, we have:

$$\begin{aligned}
\|\hat{\tilde{A}}_{a,N}^v\| &\leq \frac{N}{N-a}\|\left(\hat{\tilde{A}}_{0,N}^v\right)\| + \frac{a}{N-a}\|\left(\hat{\tilde{A}}_{0,a}^v\right)\| \\
&\leq \frac{CN}{N-a}\sqrt{\frac{(d + \upsilon \log T)\sigma_{\max}(\Sigma)}{NB\sigma_{\min}(G)}} + \frac{Ca}{N-a}\sqrt{\frac{(d + \upsilon \log T)\sigma_{\max}(\Sigma)}{aB\sigma_{\min}(G)}} .
\end{aligned} \tag{26}$$

Choose $a < N/2$. Since

$$\mathbb{P}\left[\tilde{\mathcal{M}}^{0,N-1} \cap \tilde{\mathcal{D}}^{0,N-1}\right] \geq 1 - (\frac{1}{T^\upsilon} + \frac{1}{T^\alpha})$$

we have

$$\begin{aligned}
&\mathbb{P}\left[\|\hat{\tilde{A}}_{a,N}^v\| > C\sqrt{\frac{(d + \upsilon \log T)\sigma_{\max}(\Sigma)}{(N-a)B\sigma_{\min}(G)}}\right] \\
&\leq \frac{1}{T^\alpha} + \frac{3}{T^\upsilon}
\end{aligned} \tag{27}$$

## E.2   Bias

We now consider the bias term: $\hat{\tilde{A}}_{a,N}^b - A^* := \frac{1}{N-a}\sum_{t=a+1}^N \left(\tilde{A}_B^{t-1,b} - A^*\right)$. First note that, from equation (19), we have

$$\left\|\hat{\tilde{A}}_{a,N}^b - A^*\right\| \leq \frac{1}{N-a}\sum_{t=a+1}^N \|A_0 - A^*\| \left\|\prod_{s=0}^{t-1}\tilde{H}_{0,B-1}^s\right\| \tag{28}$$

Now from lemma 31, if $a > c_1\left(d + \log \frac{N}{\delta}\right)$ then conditional on $\tilde{\mathcal{D}}^{0,N-1}$ with probability at least $1 - \delta$, for all $a + 1 \leq t \leq N$ we have

$$\left\|\prod_{s=0}^{t-1}\tilde{H}_{0,B-1}^s\right\| \leq 2\left(1 - \gamma B\sigma_{\min}(G)\right)^{c_2 t} \tag{29}$$

Note that in lemma 31 we only condition on $\tilde{\mathcal{D}}^{0,t-1}$ but due to buffer independence and that $\mathbb{P}\left[\tilde{\mathcal{D}}^{0,N-1}\right] \geq 1 - \frac{1}{T^\alpha}$ we can condition on $\tilde{\mathcal{D}}^{0,N-1}$.

Note that in the proof of lemma 31 the constant $c_2$ is actually at most 1 i.e., $0 < c_2 \leq 1$. Hence from Bernoulli's inequality, for $x < 1$

$$(1-x)^{c_2} \leq 1 - c_2 x$$

Thus conditional on $\tilde{\mathcal{D}}^{0,N-1}$ with probability at least $1 - \delta$

$$
\begin{aligned}
\left\| \hat{\tilde{A}}_{a,N}^b - A^* \right\| &\leq \frac{\|A_0 - A^*\|}{N - a} \sum_{t=a+1}^{\infty} 2 \left(1 - \gamma B \sigma_{\min}(G)\right)^{c_2 t} \\
&= 2 \frac{\|A_0 - A^*\|}{N - a} \frac{(1 - \gamma B \sigma_{\min}(G))^{c_2 a}}{c_2 \gamma B \sigma_{\min}(G)} \\
&\leq c_3 \frac{\|A_0 - A^*\|}{N - a} \frac{e^{-c_2 a \gamma B \sigma_{\min}(G)}}{\gamma B \sigma_{\min}(G)}
\end{aligned}
\tag{30}
$$

Hence choosing $\delta = \frac{1}{T^v}$ we have for $a > c_1 \left(d + \log \frac{N}{\delta}\right)$

$$
\mathbb{P}\left[ \left\| \hat{\tilde{A}}_{a,N}^b - A^* \right\| > c_3 \frac{\|A_0 - A^*\|}{N - a} \frac{e^{-c_2 a \gamma B \sigma_{\min}(G)}}{\gamma B \sigma_{\min}(G)} \right] \leq \frac{1}{T^\alpha} + \frac{1}{T^v}
\tag{31}
$$

Define $\beta_b$ as

$$
\beta_b = c_3 \frac{1}{N - a} \frac{e^{-c_2 a \gamma B \sigma_{\min}(G)}}{\gamma B \sigma_{\min}(G)}
\tag{32}
$$

Thus by union bound and equations (27) and (31) we get

$$
\begin{aligned}
&\mathbb{P}\left[ \left\| \hat{\tilde{A}}_{a,N} - A^* \right\| > C\sqrt{\frac{(d + v \log T)\sigma_{\max}(\Sigma)}{(N - a)B\sigma_{\min}(G)}} + \beta_b \|A_0 - A^*\| \right] \\
&\leq \frac{2}{T^\alpha} + \frac{4}{T^v}
\end{aligned}
\tag{33}
$$

Now from lemma 11 we see that on the event $\hat{\mathcal{D}}^{0,N-1}$

$$
\left\| \hat{A}_{a,N} - \hat{\tilde{A}}_{a,N} \right\| \leq c\gamma^2 R^2 T^2 \|A^{*u}\|
\tag{34}
$$

Since $\mathbb{P}\left[\hat{\mathcal{D}}^{0,N-1}\right] \geq 1 - \frac{1}{T^\alpha}$, we obtain

$$
\mathbb{P}\left[ \left\| \hat{A}_{a,N} - \hat{\tilde{A}}_{a,N} \right\| \leq c\gamma^2 R^2 T^2 \|A^{*u}\| \right] \geq 1 - \frac{1}{T^\alpha}
\tag{35}
$$

Therefore choosing $\delta = \frac{1}{T^v}$ we have for $N/2 > a > c_1 \left(d + \log \frac{N}{\delta}\right)$

$$
\begin{aligned}
&\mathbb{P}\left[ \left\| \hat{A}_{a,N} - A^* \right\| > C\sqrt{\frac{(d + v \log T)\sigma_{\max}(\Sigma)}{(N - a)B\sigma_{\min}(G)}} + \beta_b \|A_0 - A^*\| + c_4 \gamma^2 R^2 T^2 \|A^{*u}\| \right] \\
&\leq \frac{3}{T^\alpha} + \frac{4}{T^v}
\end{aligned}
\tag{36}
$$

where $\beta_b$ is defined in (32).

The theorem follows by adjusting the constants (in choosing $\delta$) such the above probability is at most $\frac{3}{T^\alpha} + \frac{1}{2T^v}$ and then choosing $v$ such that $\frac{3}{T^\alpha} \leq \frac{1}{2T^v}$.

# F   Bias Variance Analysis of Last and Average Iterate

In this section, our goal is to provide a PSD upper bound on

$$
\mathbb{E}\left[ \left(\tilde{A}_B^{t-1} - A^*\right)^\top \left(\tilde{A}_B^{t-1} - A^*\right) \right], \mathbb{E}\left[ \left(\hat{\tilde{A}}_{a,N} - A^*\right)^\top \left(\hat{\tilde{A}}_{a,N} - A^*\right) \right]
$$

using the bias variance decomposition in (18) and (22). This bound leads to Theorem 15 which is critical for our parameter error proof (Theorem 5).

## F.1 Variance of the Last Iterate

The goal of this section is to bound error due to $\left(\tilde{A}_B^{t-1,v}\right)$. For brevity, we will introduce the following notation:

$$\tilde{V}_{t-1} = \mathbb{E}\left[\left(\tilde{A}_B^{t-1,v}\right)^\top \left(\tilde{A}_B^{t-1,v}\right) 1\left[\tilde{\mathcal{D}}^{0,t-1}\right]\right]. \tag{37}$$

The following proposition is the main result of this section.

**Proposition 1.** *Let* $\gamma \leq \frac{1}{2R}$. *Let the noise covariance be* $\mathbb{E}\left[\eta_t \eta_t^T\right] = \Sigma$. *Then,*

$$\tilde{V}_{t-1} \preceq \frac{\gamma \operatorname{Tr}(\Sigma)}{1-\gamma R}\left[I - \mathbb{E}\left[\left(\prod_{s=1}^t \tilde{H}_{0,B-1}^{t-s,\top}\right)\left(\prod_{s=t}^1 \tilde{H}_{0,B-1}^{t-s}\right) 1\left[\tilde{\mathcal{D}}^{0,t-1}\right]\right]\right] + c_1 \gamma^2 d\sigma_{\max}(\Sigma)(Bt)^2 \frac{1}{T^{\alpha/2}}I,$$

$$\tilde{V}_{t-1} \succeq \gamma \operatorname{Tr}(\Sigma)\left[I - \mathbb{E}\left[\left(\prod_{s=1}^t \tilde{H}_{0,B-1}^{t-s,\top}\right)\left(\prod_{s=t}^1 \tilde{H}_{0,B-1}^{t-s}\right) 1\left[\tilde{\mathcal{D}}^{0,t-1}\right]\right]\right] - c_4 \gamma^2 d\sigma_{\max}(\Sigma)(Bt)^2 \frac{1}{T^{\alpha/2}}I,$$

*for some absolute constants* $c_i > 0$, $1 \leq i \leq 4$.

We refer to Section H in the appendix for a full proof. Note that we have, $\frac{1}{1-\gamma\|X\|^2} \leq 2$.

**Corollary 1.** *In the same setting as Proposition 1, we have:*

$$\tilde{V}_{t-1} \preceq c_1 \gamma \operatorname{Tr}(\Sigma)I + c_2 \gamma^2 d\sigma_{\max}(\Sigma)(Bt)^2 \frac{1}{T^{\alpha/2}}I, \tag{38}$$

*for some constants* $c_1, c_2 > 0$. *If* $T^{\alpha/2} > T^2$, *then* $V_{t,1} \preceq c\gamma d\sigma_{\max}I$, *for some constant* $c > 0$.

## F.2 Variance of the Average Iterate

In this section we are interested in bounding: $\mathbb{E}\left[\left(\hat{\tilde{A}}_{a,N}^v\right)^\top \left(\hat{\tilde{A}}_{a,N}^v\right) 1\left[\tilde{\mathcal{D}}^{0,N-1}\right]\right]$, for $a = \theta N$ with $0 \leq \theta < 1$, where,

$$\hat{\tilde{A}}_{a,N}^v = \frac{1}{N-a}\sum_{t=a+1}^N \tilde{A}_B^{t-1,v}, \tag{39}$$

and further, recall that $T = N(B + u)$. The main bound in this section is given in Proposition 2. Note that we have,

$$\mathbb{E}\left[\left(\hat{\tilde{A}}_{a,N}^v\right)^\top \left(\hat{\tilde{A}}_{a,N}^v\right) 1\left[\tilde{\mathcal{D}}^{0,N-1}\right]\right]$$

$$= \frac{1}{(N-a)^2}\sum_{t=a+1}^N \mathbb{E}\left[\left(\tilde{A}_B^{t-1,v}\right)^\top \left(\tilde{A}_B^{t-1,v}\right) 1\left[\tilde{\mathcal{D}}^{0,N-1}\right]\right]$$

$$+ \frac{1}{(N-a)^2}\sum_{t_1 \neq t_2} \mathbb{E}\left[\left(\tilde{A}_B^{t_1-1,v}\right)^\top \left(\tilde{A}_B^{t_2-1,v}\right) 1\left[\tilde{\mathcal{D}}^{0,N-1}\right]\right] \tag{40}$$

**Proposition 2.** *Let* $\gamma \leq \min\{\frac{c}{6RB}\frac{1}{2R}\}$ *for* $0 < c < 1$. *Then for* $\hat{\tilde{A}}_{a,N}^v$ *defined in* (39), *there are constants* $c_1, c_2 > 0$ *such that if* $T^{\alpha/2} > c_1 \frac{\sqrt{M_4}}{\sigma_{\min}(G)}$, *then:*

$$\mathbb{E}\left[\left(\hat{\tilde{A}}_{a,N}^v\right)^\top \left(\hat{\tilde{A}}_{a,N}^v\right) 1\left[\tilde{\mathcal{D}}^{0,N-1}\right]\right]$$

$$\preceq \frac{1}{(N-a)^2}\sum_{t=a+1}^N\left[\tilde{V}_{t-1}\left(\sum_{s=0}^{N-t}\mathcal{H}^s\right) + \left(\sum_{s=0}^{N-t}\mathcal{H}^s\right)^\top \tilde{V}_{t-1}\right] + c_2\delta I \tag{41}$$

$$= \frac{1}{(N-a)^2}\sum_{t=a+1}^N\left[\tilde{V}_{t-1}\left(I-\mathcal{H}\right)^{-1} + \left(I-\mathcal{H}^\top\right)^{-1}\tilde{V}_{t-1}\right] + c_2\delta I +$$

$$\frac{1}{(N-a)^2}\sum_{t=a+1}^N\left[\tilde{V}_{t-1}\left(I-\mathcal{H}\right)^{-1}\mathcal{H}^{N-t+1} + \left(\mathcal{H}^\top\right)^{N-t+1}\left(I-\mathcal{H}^\top\right)^{-1}\tilde{V}_{t-1}\right] \tag{42}$$

*and,*

$$\delta \equiv \delta(N, B, R) = \gamma^2 T^2 R d\sigma_{\max}(\Sigma) \frac{1}{T^{\alpha/2}} \tag{43}$$

*and $\mathcal{H}$ is given by,*

$$\mathcal{H} = \mathbb{E}\left[\prod_{j=0}^{B-1} \left(I - 2\gamma \tilde{X}_{-j}^0 \tilde{X}_{-j}^{0,\top}\right) \mathbf{1}\left[\cap_{j=0}^{B-1}\left\{\|\tilde{X}_{-j}^0\|^2 \leq R\right\}\right]\right], \tag{44}$$

*with $\tilde{X}_0$ sampled from the stationary distribution $\pi$ and $\tilde{X}_t$ follows the $\mathsf{VAR}(A^*, \mu)$.*

See section I in the appendix for the proof.

### F.3 Bias of the Last Iterate

In this we will analyze the bias term of the last iterate. That is we want to bound:

$$\mathbb{E}\left[\left(\tilde{A}_B^{t-1,b} - A^*\right)^\top \left(\tilde{A}_B^{t-1,b} - A^*\right) \mathbf{1}\left[\tilde{\mathcal{D}}^{0,t-1}\right]\right].$$

Where $\left(\tilde{A}_B^{t-1,b} - A^*\right)$ is defined in (19).

**Theorem 14.** *Let $\gamma RB \leq \frac{c}{6}$ for some $0 < c < 1$ with $B$ such that $\gamma R \leq \frac{1}{2}$. Then there are constants $c_1, c_2, c_3 > 0$ such that if $T^{\alpha/2} > c_1 \frac{\sqrt{M_4}}{\sigma_{\min}(G)}$ (where $M_4 = \mathbb{E}\left[\|\tilde{X}_{-0}^0\|^4\right]$) then*

$$\mathbb{E}\left[\left(\tilde{A}_B^{t-1,b} - A^*\right)^\top \left(\tilde{A}_B^{t-1,b} - A^*\right) \mathbf{1}\left[\tilde{\mathcal{D}}^{0,t-1}\right]\right] \preceq \|A_0 - A^*\|^2 \left(1 - c_2\gamma B\sigma_{\min}(G)\right)^t I \tag{45}$$

See section J for the proof.

### F.4 Bias of the Tail-Averaged Iterate

We define the tail averaged bias as

$$\hat{\tilde{A}}_{a,N}^b = \frac{1}{N-a} \sum_{t=a+1}^{N} \tilde{A}_B^{t-1,b} \tag{46}$$

**Theorem 15.** *Let $\gamma RB \leq \frac{c}{6}$ for some $0 < c < 1$ and $B$ such that $\gamma R \leq \frac{1}{2}$. There exist constants $c_1, c_2 > 0$ such that if $T = N(B + u)$ satisfies $T^{\alpha/2} > c_1 \frac{\sqrt{M_4}}{\sigma_{\min}(G)}$ then for $a = \theta N$ with $0 < \theta < 1$ we have*

$$\left\|\mathbb{E}\left[\left(\hat{\tilde{A}}_{a,N}^b - A^*\right)^\top \left(\hat{\tilde{A}}_{a,N}^b - A^*\right) \mathbf{1}\left[\tilde{\mathcal{D}}^{0,N-1}\right]\right]\right\| \leq$$

$$c_2 \frac{1}{B(N-a)} \frac{e^{-c_3 B\gamma\sigma_{\min}(G)a}}{\gamma\sigma_{\min}(G)} \|A_0 - A^*\|^2 \tag{47}$$

See section K for the proof.

## G  Prediction Error

Recall the definition of the prediction error at stationarity.

$$\mathcal{L}_{\mathsf{pred}}(\hat{A}; A^*, \mu) := \mathbb{E}_{X_t \sim \pi}\|X_{t+1} - \hat{A}X_t\|^2 \tag{48}$$

where $\pi$ is the stationary distribution.

Note that the prediction loss is a function of possibly random estimator $\hat{A}$. Hence the expectation in (48) is only with respect to the process $(X_t)$ (which is considered independent of $\hat{A}$). Letting $G = \mathbb{E}\left[X_t X_t^\top\right]$ as the covariance matrix of the process at stationarity, we can write

$$\mathcal{L}_{\mathsf{pred}}(\hat{A}; A^*, \mu) = \mathrm{Tr}(G(\hat{A} - A^*)^\top(\hat{A} - A^*)) + \mathrm{Tr}(\Sigma) \tag{49}$$

We are interested in bounding the expected prediction loss of the estimator which is the average iterate $\hat{A}_{a,N}$ of our algorithm $\mathsf{SGD} - \mathsf{RER}$ (with $a = \theta N$). Note that $\hat{A}_{a,N} = \hat{A}^b_{a,N} + \hat{A}^v_{a,N}$ where the superscripts $b$ and $v$ correspond to bias and variance respectively (c.f. (22))

Hence

$$
\begin{aligned}
\mathbb{E}\left[\mathcal{L}_{\mathsf{pred}}(\hat{A}_{a,N}; A^*, \mu)\right] &= \mathrm{Tr}(\Sigma) + \mathrm{Tr}\left(G^{1/2}\mathbb{E}\left[\left(\hat{A}_{a,N} - A^*\right)^\top \left(\hat{A}_{a,N} - A^*\right)\right]G^{1/2}\right) \\
&\leq \mathrm{Tr}(\Sigma) + 2\,\mathrm{Tr}\left(G^{1/2}\mathbb{E}\left[\left(\hat{A}^v_{a,N}\right)^\top \left(\hat{A}^v_{a,N}\right)\right]G^{1/2}\right) \\
&\quad + 2\,\mathrm{Tr}\left(G^{1/2}\mathbb{E}\left[\left(\hat{A}^b_{a,N} - A^*\right)^\top \left(\hat{A}^b_{a,N} - A^*\right)\right]G^{1/2}\right) \quad (50)
\end{aligned}
$$

But we will only bound $\mathbb{E}\left[\mathcal{L}_{\mathsf{pred}}(\hat{A}_{a,N}; A^*, \mu)\mathbf{1}\left[\mathcal{D}^{0,N-1}\right]\right]$ so that we have a tight upper bound on the conditional expectation of $\mathcal{L}_{\mathsf{pred}}$ over a high probability event.

As before we will just focus on the prediction error obtained using the algorithmic iterates from the coupled process, i.e., we will bound $\mathbb{E}\left[\mathcal{L}_{\mathsf{pred}}(\tilde{\hat{A}}_{a,N}; A^*, \mu)\mathbf{1}\left[\tilde{\mathcal{D}}^{0,N-1}\right]\right]$

### G.1 Variance of prediction error

In this section we will focus on analyzing the variance part of the expected prediction loss under the coupled process

$$
\tilde{\mathcal{L}}^v = \mathrm{Tr}\left(G^{1/2}\mathbb{E}\left[\left(\tilde{\hat{A}}^v_{a,N}\right)^\top \left(\tilde{\hat{A}}^v_{a,N}\right)\mathbf{1}\left[\tilde{\mathcal{D}}^{0,N-1}\right]\right]G^{1/2}\right) \quad (51)
$$

where $T = N(B + u)$.

We begin with few lemmata which would be useful in bounding $\tilde{\mathcal{L}}^v$. Recall the definition of $\mathcal{H}$

$$
\mathcal{H} = \mathbb{E}\left[\prod_{j=0}^{B-1}\left(I - 2\gamma\tilde{X}^0_{-j}\tilde{X}^{0,\top}_{-j}\right)\mathbf{1}[\tilde{\mathcal{D}}^0_{-0}]\right] \quad (52)
$$

with $\tilde{X}_0$ sampled from the stationary distribution $\pi$.

**Lemma 16.** *Let $\gamma \leq \frac{1}{8RB}$. Then*

$$
\mathcal{H} + \mathcal{H}^\top \preceq 2\left(I - \frac{4}{3}\gamma BG\right) + \frac{8}{3}\gamma B\sqrt{M_4}\frac{1}{T^{\alpha/2}}I \quad (53)
$$

*where $M_4 = \mathbb{E}\left[\|\tilde{X}^0_{-0}\|^4\right]$. For simplicity, we just say that for $\gamma RB < \frac{c}{4}$ with $0 < c < 1$ then*

$$
\mathcal{H} + \mathcal{H}^\top \preceq 2\left(I - c_1\gamma BG\right) + c_2\gamma B\sqrt{M_4}\frac{1}{T^{\alpha/2}}I \quad (54)
$$

*for some absolute constants $c_1, c_2 > 0$.*

The proof is similar to the combined proofs of Lemmas 28 and 29. We therefore skip it.

Next we will bound $\mathrm{Tr}(G(I - \mathcal{H})^{-1})$.

**Lemma 17.** *Let $\gamma RB < \frac{c_1}{4}$ with $0 < c_1 < 1$. Then for $T$ such that $T^{\alpha/2} > c_2\frac{\sqrt{M_4}}{\sigma_{\min}(G)}$ we have*

$$
\mathrm{Tr}\left(G(I - \mathcal{H})^{-1}\right) \leq c\frac{d}{\gamma B} \quad (55)
$$

*for some absolute constant $c > 0$.*

*Proof.* First note that

$$\text{Tr}\left(G(I - \mathcal{H})^{-1})\right) = \text{Tr}\left(G^{1/2}(I - \mathcal{H})^{-1}G^{1/2})\right)$$

$$= \text{Tr}\left(\left(G^{-1} - G^{-1/2}\mathcal{H}G^{-1/2}\right)^{-1}\right)$$

$$\leq d\left\|\left(G^{-1} - G^{-1/2}\mathcal{H}G^{-1/2}\right)^{-1}\right\|$$

$$= \frac{d}{\sigma_{\min}\left(G^{-1} - G^{-1/2}\mathcal{H}G^{-1/2}\right)} \tag{56}$$

Let $Q = \left(G^{-1} - G^{-1/2}\mathcal{H}G^{-1/2}\right)$. Let $\text{Sym}(Q) = Q + Q^\top$. We will relate $\sigma_{\min}(Q)$ with $\sigma_{\min}\left(\frac{\text{Sym}(Q)}{2}\right)$. From AM-GM inequality, for any $\theta > 0$, we have

$$\frac{Q^\top Q}{\theta} + \theta I \succeq \text{Sym}(Q) \tag{57}$$

Also

$$\sigma_{\min}^2(Q) = \inf_{x:\|x\|=1} x^\top Q^\top Q x \tag{58}$$

Further, from lemma 16 we have

$$\text{Sym}(Q) = G^{-1} - G^{-1/2}\frac{\mathcal{H} + \mathcal{H}^T}{2}G^{-1/2}$$

$$\succeq c_1\gamma B I - c_2\gamma B\sqrt{M_4}\frac{1}{T^{\alpha/2}}G^{-1}$$

$$\succeq c_1\gamma B I - c_2\gamma B\sqrt{M_4}\frac{1}{T^{\alpha/2}}\frac{1}{\sigma_{\min}(G)}I \tag{59}$$

Hence combining equations (57), (58) and (59) we have:

$$\frac{\sigma_{\min}^2(Q)}{\theta} + \theta \succeq c_1\gamma B - c_2\gamma B\sqrt{M_4}\frac{1}{T^{\alpha/2}}\frac{1}{\sigma_{\min}(G)}. \tag{60}$$

Now choosing $\theta = \frac{1}{2}c_1\gamma B$ we get:

$$\sigma_{\min}^2(Q) \geq \frac{c_1^2}{4}\gamma^2 B^2 - \frac{c_2 c_1}{2}\gamma^2 B^2\sqrt{M_4}\frac{1}{T^{\alpha/2}}\frac{1}{\sigma_{\min}(G)}. \tag{61}$$

Now choose $T$ large enough such that $\frac{c_2 c_1}{2}\sqrt{M_4}\frac{1}{T^{\alpha/2}}\frac{1}{\sigma_{\min}(G)} \leq \frac{c_1^2}{8}$. Then, $\sigma_{\min}^2(Q) \geq c_3\gamma^2 B^2$, for some constant $c_3 > 0$. Hence from (56),

$$\text{Tr}\left(G(I - \mathcal{H})^{-1}\right) \leq c_4\frac{d}{\gamma B}.$$

$\square$

Next we bound $\text{Tr}(\Delta(I - \mathcal{H})^{-1}G)$ for any symmetric matrix $\Delta$. Let $\kappa(G) = \frac{\sigma_{\max(G)}}{\sigma_{\min(G)}}$ denote the condition number of $G$.

**Lemma 18.** *Let* $\gamma RB \leq \frac{c_1}{4}$ *with* $0 < c_1 < 1$. *Then for* $T$ *such that* $T^{\alpha/2} > c_2\frac{\sqrt{M_4}}{\sigma_{\min}(G)}$ *we have*

$$\left|\text{Tr}\left(\Delta(I - \mathcal{H})^{-1}G\right)\right| \leq c\frac{d}{\gamma B}\|\Delta\|\sqrt{\kappa(G)} \tag{62}$$

*for some absolute constant* $c > 0$.

*Proof.* We have

$$\left|\text{Tr}\left(\Delta(I - \mathcal{H})^{-1}G\right)\right| = \left|\text{Tr}\left(G^{1/2}\Delta G^{-1/2}G^{1/2}(I - \mathcal{H})^{-1}G^{1/2}\right)\right|$$

$$\leq d\left\|G^{1/2}\Delta G^{-1/2}\right\|\left\|G^{1/2}(I - \mathcal{H})^{-1}G^{1/2}\right\|$$

$$\leq d\sqrt{\kappa(G)}\|\Delta\|\left\|G^{1/2}(I - \mathcal{H})^{-1}G^{1/2}\right\| \tag{63}$$

From the proof of lemma 17, we know that

$$\left\|G^{1/2}(I - \mathcal{H})^{-1}G^{1/2}\right\| \le c\frac{1}{\gamma B} \tag{64}$$

for $T$ satisfying the condition the statement of the lemma.

Hence:

$$\left|\text{Tr}\left(\Delta(I - \mathcal{H})^{-1}G\right)\right| \le c\sqrt{\kappa(G)}\,\|\Delta\|\,\frac{d}{\gamma B} \tag{65}$$

$$\square$$

Our goal is to bound $\text{Tr}(\tilde{V}_{t-1}(I - \mathcal{H})^{-1}G)$. From proposition 1 we can decompose $\tilde{V}_{t-1}$ as:

$$\tilde{V}_{t-1} = \gamma\,\text{Tr}(\Sigma)I + (\tilde{V}_{t-1} - \gamma\,\text{Tr}(\Sigma)I), \tag{66}$$

and hence,

$$\text{Tr}(\tilde{V}_{t-1}(I - \mathcal{H})^{-1}G) = \gamma\,\text{Tr}(\Sigma)\,\text{Tr}((I - \mathcal{H})^{-1}G) + \text{Tr}\left((\tilde{V}_{t-1} - \gamma\,\text{Tr}(\Sigma))(I - \mathcal{H})^{-1}G\right). \tag{67}$$

To bound the second term in (67) we want to use lemma 18. Hence we need to bound the norm of $\tilde{V}_{t-1} - \gamma\,\text{Tr}(\Sigma)$.

**Lemma 19.** *Let $\gamma \le \min\left\{\frac{c}{4RB}, \frac{1}{2R}\right\}$ for $0 < c < 1$. Then there are constants $c_1, c_2, c_3 > 0$ such that for $T^{\alpha/2} > c_1 \frac{\sqrt{M_4}}{\sigma_{\min}(G)}$ we have*

$$\left\|\tilde{V}_{t-1} - \gamma\,\text{Tr}(\Sigma)\right\| \le c_2\gamma d\sigma_{\max}\left[\frac{1}{B} + (1 - c_3\gamma B\sigma_{\min}(G))^t\right] \tag{68}$$

*for some constant $c_1 > 0$.*

*Proof.* From proposition 1 we have

$$\left\|\tilde{V}_{t-1} - \gamma\,\text{Tr}(\Sigma)I\right\| \le \gamma\,\text{Tr}(\Sigma)\frac{\gamma R}{1 - \gamma R} +$$

$$c_1\gamma\,\text{Tr}(\Sigma)\left\|\mathbb{E}\left[\left(\prod_{s=1}^{t}\tilde{H}_{0,B-1}^{t-s,\top}\right)\left(\prod_{s=t}^{1}\tilde{H}_{0,B-1}^{t-s}\right)1\left[\tilde{\mathcal{D}}^{0,t-1}\right]\right]\right\|$$

$$+ c_2\gamma d\sigma_{\max}(\Sigma)T^2\frac{1}{T^{\alpha/2}}. \tag{69}$$

From lemma 26 equation (111) we can show that

$$\left\|\mathbb{E}\left[\left(\prod_{s=1}^{t}\tilde{H}_{0,B-1}^{t-s,\top}\right)\left(\prod_{s=t}^{1}\tilde{H}_{0,B-1}^{t-s}\right)1\left[\tilde{\mathcal{D}}^{0,t-1}\right]\right]\right\| \le (1 - c_3\gamma B\sigma_{\min}(G))^t. \tag{70}$$

Hence

$$\left\|\tilde{V}_{t-1} - \gamma\,\text{Tr}(\Sigma)I\right\| \le c_4\gamma d\sigma_{\max}(\Sigma)\left[\frac{\gamma R}{1 - \gamma R} + (1 - c_3\gamma B\sigma_{\min}(G))^t\right]$$

$$\le c_5\gamma d\sigma_{\max}\left[\gamma R + (1 - c_3\gamma B\sigma_{\min}(G))^t\right] \le c_6\gamma d\sigma_{\max}\left[\frac{1}{B} + (1 - c_3\gamma B\sigma_{\min}(G))^t\right]. \tag{71}$$

$$\square$$

Now we have all required ingredients for the main theorem of this section

**Theorem 20.** *Let $\gamma \le \min\left\{\frac{c}{4RB}, \frac{1}{2R}\right\}$ for $0 < c < 1$. Then there are constants $c_1, c_2, c_3, c_4 > 0$ such that for $T^{\alpha/2} > c_1 \frac{\sqrt{M_4}}{\sigma_{\min}(G)}$ the variance part of the expected prediction loss $\tilde{\mathcal{L}}^v$ (defined in (51)) for $a = \theta N$ is bounded as*

$$\tilde{\mathcal{L}}^v \le c_1\frac{d\,\text{Tr}(\Sigma)}{NB(1-\theta)} + c_2\frac{d^2\sigma_{\max}(\Sigma)}{NB(1-\theta)}\frac{\sqrt{\kappa(G)}}{B} + c_3\frac{d^2\sigma_{\max}(\Sigma)}{(NB)^2(1-\theta)^2}\sqrt{\kappa(G)}\frac{1}{\gamma\sigma_{\min}(G)}$$

$$+ c_4\gamma^2 Rd\sigma_{\max}(\Sigma)T^2\frac{1}{T^{\alpha/2}}\,\text{Tr}(G) \tag{72}$$

*Proof.* From (51) and proposition 2 equation (42) we have

$$\tilde{\mathcal{L}}^v \le \frac{2}{(N-a)^2} \sum_{t=a+1}^{N} \mathrm{Tr}\left(\tilde{V}_{t-1}(I-\mathcal{H})^{-1}G\right) \tag{73}$$

$$+ \frac{2}{(N-a)^2} \sum_{t=a+1}^{N} \mathrm{Tr}\left(\tilde{V}_{t-1}(I-\mathcal{H})^{-1}\mathcal{H}^{N-t+1}G\right) \tag{74}$$

$$+ c\delta\,\mathrm{Tr}(G) \tag{75}$$

where $\delta = \gamma^2 T^2 R d \sigma_{\max}(\Sigma)\frac{1}{T^{\alpha/2}}$ as defined in (43)

For the first term (73) we have from (67), lemma 17, lemma 18 and lemma 19

$$\mathrm{Tr}\left(\tilde{V}_{t-1}(I-\mathcal{H})^{-1}G\right) \le c_1\gamma\,\mathrm{Tr}(\Sigma)\frac{d}{\gamma B} +$$
$$c_2\frac{d}{\gamma B}\sqrt{\kappa(G)}\gamma d\sigma_{\max}(\Sigma)\left[\frac{1}{B} + (1-c_3\gamma B\sigma_{\min}(G))^t\right]$$
$$= c_1\frac{d\,\mathrm{Tr}(\Sigma)}{B} + c_2\frac{d^2\sigma_{\max}(\Sigma)}{B}\frac{\sqrt{\kappa(G)}}{B} +$$
$$c_4\frac{d^2\sigma_{\max}(\Sigma)}{B}\sqrt{\kappa(G)}\left(1-c_3\gamma B\sigma_{\min}(G)\right)^t \tag{76}$$

Therefore

$$\frac{2}{(N-a)^2}\sum_{t=a+1}^{N}\mathrm{Tr}\left(\tilde{V}_{t-1}(I-\mathcal{H})^{-1}G\right) \le c_1\frac{d\,\mathrm{Tr}(\Sigma)}{NB(1-\theta)} + c_2\frac{d^2\sigma_{\max}(\Sigma)}{NB(1-\theta)}\frac{\sqrt{\kappa(G)}}{B} +$$
$$c_5\frac{d^2\sigma_{\max}(\Sigma)}{N^2B(1-\theta)^2}\sqrt{\kappa(G)}\frac{(1-c_3\gamma B\sigma_{\min}(G))^{a+1}}{\gamma B\sigma_{\min}(G)} \tag{77}$$

Similarly, for the second term (74), from corollary 1, lemma 18, lemma 26 and the fact that $(I-\mathcal{H})^{-1}$ and $\mathcal{H}^{N-t+1}$ commute, we get

$$\left|\mathrm{Tr}\left(\tilde{V}_{t-1}(I-\mathcal{H})^{-1}\mathcal{H}^{N-t+1}G\right)\right| \le c_1\frac{d}{\gamma B}\sqrt{\kappa}\|\tilde{V}_{t-1}\|\|\mathcal{H}^{N-t+1}\|$$
$$\le c_2\frac{d}{\gamma B}\sqrt{\kappa(G)}\gamma d\sigma_{\max}(\Sigma)\left(1-c_3\gamma B\sigma_{\min}(G)\right)^{(N-t+1)}$$
$$= c_2\frac{d^2\sigma_{\max}(\Sigma)}{B}\sqrt{\kappa(G)}\left(1-c_3\gamma B\sigma_{\min}(G)\right)^{(N-t+1)} \tag{78}$$

Therefore

$$\left|\frac{2}{(N-a)^2}\sum_{t=a+1}^{N}\mathrm{Tr}\left(\tilde{V}_{t-1}(I-\mathcal{H})^{-1}\mathcal{H}^{N-t+1}G\right)\right| \le c\frac{d^2\sigma_{\max}(\Sigma)}{N^2B(1-\theta)^2}\sqrt{\kappa(G)}\frac{1}{\gamma B\sigma_{\min}(G)} \tag{79}$$

Hence we obtain,

$$\tilde{\mathcal{L}}^v \le c_1\frac{d\,\mathrm{Tr}(\Sigma)}{NB(1-\theta)} + c_2\frac{d^2\sigma_{\max}(\Sigma)}{NB(1-\theta)}\frac{\sqrt{\kappa(G)}}{B} +$$
$$c_3\frac{d^2\sigma_{\max}(\Sigma)}{N^2B^2(1-\theta)^2}\sqrt{\kappa(G)}\frac{1}{\gamma\sigma_{\min}(G)} + c_4\gamma^2 R d\sigma_{\max}(\Sigma)T^2\frac{1}{T^{\alpha/2}}\,\mathrm{Tr}(G). \tag{80}$$

$\square$

## G.2 Bias of prediction error

In this section we will focus on analyzing the (tail-averaged) bias part of the expected prediction loss from the coupled process

$$\tilde{\mathcal{L}}^b = \mathrm{Tr}\left(G^{1/2}\mathbb{E}\left[\left(\left(\hat{\tilde{A}}_{a,N}^b - A^*\right)\right)^{\top}\left(\left(\hat{\tilde{A}}_{a,N}^b - A^*\right)\right)\mathbf{1}\left[\tilde{\mathcal{D}}^{0,N-1}\right]\right]G^{1/2}\right) \tag{81}$$

where $T = N(B+u)$ and $a = \theta N$ for $0 < \theta < 1$.

**Theorem 21.** *Let $\gamma RB \leq \frac{c}{6}$ for some $0 < c < 1$ and $B$ such that $\gamma R \leq \frac{1}{2}$. There exist constants $c_1, c_2, c_3, c_4 > 0$ such that if $T$ satisfies $T^{\alpha/2} > c_1 \frac{\sqrt{M_4}}{\sigma_{\min}(G)}$ then for $a = \theta N$ with $0 < \theta < 1$ we have*

$$\tilde{\mathcal{L}}^b \leq c_2 \frac{1}{NB(1-\theta)} \frac{\mathrm{Tr}(G)}{\gamma \sigma_{\min}(G)} e^{-c_3 NB\gamma \sigma_{\min}(G)\theta} \|A_0 - A^*\|^2 \tag{82}$$

*Proof.* Proof follows directly from (81) and theorem 15. $\qquad\square$

### G.3 Overall Prediction Error

Combining theorem 20 and theorem 21 along with lemma 12 we obtain the main theorem on prediction error of SGD − RER

**Theorem 22.** *Let $R, B, u, \alpha$ be chosen as in section 4. Let $\gamma = \frac{c}{4RB} \leq \frac{1}{2R}$ for $0 < c < 1$. Then there are constants $c_1, c_2, c_3, c_4 > 0$ such that for $T^{\alpha/2} > c_1 \frac{\sqrt{M_4}}{\sigma_{\min}(G)}$ the expected prediction loss $\mathcal{L}$ (defined in (49)) is bounded as*

$$\mathbb{E}\left[\mathcal{L}_{\mathsf{pred}}(\hat{A}_{a,N}; A^*, \mu)\mathbb{1}\left[\mathcal{D}^{0,N-1}\right]\right] \leq c_2 \left[ \frac{d\,\mathrm{Tr}(\Sigma)}{B(N-a)} + \frac{d^2\sigma_{\max}(\Sigma)}{B(N-a)} \frac{\sqrt{\kappa(G)}}{B} \right] +$$

$$c_3 \left[ \frac{d^2\sigma_{\max}(\Sigma)}{B^2(N-a)^2} \sqrt{\kappa(G)} \frac{1}{\gamma\sigma_{\min}(G)} + \right.$$

$$\frac{1}{B(N-a)} d\kappa(G)RB e^{-c_4 \frac{\sigma_{\min}(G)}{R} a} \|A_0 - A^*\|^2 +$$

$$\left. \left( \frac{T^3}{B^3} \|A^{*u}\| + \frac{d\sigma_{\max}(\Sigma)}{R} \frac{T^2}{B^2} \frac{1}{T^{\alpha/2}} \right) \mathrm{Tr}(G) \right] \tag{83}$$

*Hence, if $\|A^*\| < c_0 < 1$ then choosing $a \geq C \frac{R \log T}{\sigma_{\min}(G)}$ such that $B(N-a) = \Theta(T)$ and $B, u$ as in section 4 we get*

$$\mathbb{E}\left[\mathcal{L}_{\mathsf{pred}}(\hat{A}_{a,N}; A^*, \mu)\mathbb{1}\left[\mathcal{D}^{0,N-1}\right]\right] \leq c_2 \frac{d\,\mathrm{Tr}(\Sigma)}{T} + o\left(\frac{1}{T}\right) \tag{84}$$

## H  Proof of Proposition 1

*Proof of Proposition 1.* First note that

$$\left(\tilde{A}_b^{t-1,v}\right)^\top \left(\tilde{A}_b^{t-1,v}\right) = \sum_{r=1}^{t} \sum_{j=0}^{B-1} \widetilde{\mathrm{Dg}}(t,r,j) + \sum_{r_1,r_2=1}^{t} \sum_{j_1,j_2=0}^{B-1} \widetilde{\mathrm{Cr}}(t,r_1,j_1,r_2,j_2) \tag{85}$$

where

$$\widetilde{\mathrm{Dg}}(t,r,j) = 4\gamma^2 \left\|\eta_{-j}^{t-r}\right\|^2 \cdot$$
$$\left(\prod_{s=1}^{r-1} \tilde{H}_{0,B-1}^{t-s,\top}\right) \tilde{H}_{j+1,B-1}^{t-r,\top} \tilde{X}_{-j}^{t-r} \tilde{X}_{-j}^{t-r,\top} \tilde{H}_{j+1,B-1}^{t-r} \left(\prod_{s=r-1}^{1} \tilde{H}_{0,B-1}^{t-s}\right) \tag{86}$$

$$\widetilde{\mathrm{Cr}}(t,r_1,j_1,r_2,j_2) = 4\gamma^2 \left(\eta_{-j_1}^{t-r_1} \tilde{X}_{-j_1}^{t-r_1,\top} \tilde{H}_{j_1+1,B-1}^{t-r_1} \prod_{s=r_1-1}^{1} \tilde{H}_{0,B-1}^{t-s}\right)^\top \cdot$$
$$\left(\eta_{-j_2}^{t-r_2} \tilde{X}_{-j_2}^{t-r_2,\top} \tilde{H}_{j_2+1,B-1}^{t-r_2} \prod_{s=r_2-1}^{1} \tilde{H}_{0,B-1}^{t-s}\right) \tag{87}$$

denote the diagonal and cross terms respectively.

We begin by noting the following two facts about $\left(\tilde{A}_b^{t-1,v}\right)$:

- It has zero mean

$$\mathbb{E}\left[\left(\tilde{A}_B^{t-1,v}\right)\right] = 0 \tag{88}$$

- Let $(r_1, j_1) \neq (r_2, j_2)$. Then

$$\mathbb{E}\left[\widetilde{\mathrm{Cr}}(t, r_1, j_1, r_2, j_2)\right] = 0 \tag{89}$$

This follows because, assuming $r_1 > r_2$, the term $\eta_{-j_1}^{t-r_1} \tilde{X}_{-j_1}^{t-r_1,\top} \tilde{H}_{j_1+1,B-1}^{t-r_1}$ is independent of everything else in that expression, and that $\eta_{-j_1}^{t-r_1}$ is independent of $\tilde{X}_{-j_1}^{t-r_1,\top} \tilde{H}_{j_1+1,B-1}^{t-r_1}$. A similar argument can be made for the case when $r_1 = r_2$ but $j_1 \neq j_2$.

But we are interested in expectation on the event $\tilde{\mathcal{D}}^{0,t-1}$.

We will bound the expectation of cross terms in the following lemma.

**Lemma 23.** *We have*

$$\left\| \mathbb{E}\left[ \sum_{r_1,r_2} \sum_{j_1,j_2} \widetilde{\mathrm{Cr}}(t, r_1, j_1, r_2, j_2) \right] 1\left[\tilde{\mathcal{D}}^{0,t-1}\right] \right\| \leq 8(Bt)^2 \gamma^2 R \operatorname{Tr}(\Sigma) \frac{1}{T^{\alpha/2}} \tag{90}$$

*Proof.* Let

Consider a single cross term: $\widetilde{\mathrm{Cr}}(t, r_1, j_1, r_2, j_2)$ and without loss of generality, assume that either $r_1 > r_2$ or $r_1 = r_2$ but $j_1 < j_2$. In either case, we note that $\eta_{-j_1}^{t-r_1}$ is unconditionally independent of all other terms present in $\widetilde{\mathrm{Cr}}(t, r_1, j_1, r_2, j_2)$. The main problem here is to bound the expectation over the event $\tilde{\mathcal{D}}^{0,t-1}$. For the sake of convenience, only in this proof, we will define the following notation:

$$\widetilde{\mathrm{Cr}}(t, r_1, j_1, r_2, j_2) = E_1 \eta_{-j_1}^{t-r_1,\top} \eta_{-j_2}^{t-r_2} E_2$$

Where $E_1$ and $E_2$ are random matrices defined according to the definition of $\widetilde{\mathrm{Cr}}(t, r_1, j_1, r_2, j_2)$ and are unconditionally independent of $\eta_{-j_1}^{t-r_1,\top}$. Let $\mathcal{F}_E = \sigma(E_1, E_2, \eta_{-j_2}^{t-r_2})$. Note that when conditioned on the event $\tilde{\mathcal{D}}^{0,t-1}$, we must have the event $\mathcal{M} := \{\|E_1\| \leq 4\gamma^2\sqrt{R}\} \cap \{\|E_2\| \leq \sqrt{R}\}$ almost surely. Therefore, we conclude:

$$\mathbb{E}\left[\widetilde{\mathrm{Cr}}(t, r_1, j_1, r_2, j_2) 1\left[\tilde{\mathcal{D}}^{0,t-1}\right]\right] = \mathbb{E}\left[\widetilde{\mathrm{Cr}}(t, r_1, j_1, r_2, j_2) 1\left[\tilde{\mathcal{D}}^{0,t-1}\right] 1\left[\mathcal{M}\right]\right]$$

$$= \mathbb{E}\left[1\left[\mathcal{M}\right] E_1 \mathbb{E}\left[\eta_{-j_1}^{t-r_1,\top} 1\left[\tilde{\mathcal{D}}^{0,t-1}\right] \middle| \mathcal{F}_E\right] \eta_{-j_2}^{t-r_2} E_2\right]$$

$$\leq \mathbb{E}\left[1\left[\mathcal{M}\right] \|E_1\| \left\| \mathbb{E}\left[\eta_{-j_1}^{t-r_1,\top} 1\left[\tilde{\mathcal{D}}^{0,t-1}\right] \middle| \mathcal{F}_E\right] \right\| \|\eta_{-j_2}^{t-r_2}\| \|E_2\|\right]$$

$$\leq 4\gamma^2 R \mathbb{E}\left[\left\| \mathbb{E}\left[\eta_{-j_1}^{t-r_1,\top} 1\left[\tilde{\mathcal{D}}^{0,t-1}\right] \middle| \mathcal{F}_E\right] \right\| \|\eta_{-j_2}^{t-r_2}\|\right] \tag{91}$$

In the third step, we have used the fact that under the event $\mathcal{M}$, the norms $\|E_1\|, \|E_2\|$ are bounded. We will now bound $\mathbb{E}\left[\eta_{-j_1}^{t-r_1,\top} 1\left[\tilde{\mathcal{D}}^{0,t-1}\right] \middle| \mathcal{F}_E\right]$. Clearly, due to the unconditional independence, we must have:

$$\mathbb{E}\left[\eta_{-j_1}^{t-r_1,\top} \middle| \mathcal{F}_E\right] = 0$$

$$\implies \mathbb{E}\left[\eta_{-j_1}^{t-r_1,\top} 1\left[\tilde{\mathcal{D}}^{0,t-1}\right] \middle| \mathcal{F}_E\right] = -\mathbb{E}\left[\eta_{-j_1}^{t-r_1,\top} 1\left[\tilde{\mathcal{D}}^{0,t-1,C}\right] \middle| \mathcal{F}_E\right]$$

$$\implies \left\| \mathbb{E}\left[\eta_{-j_1}^{t-r_1,\top} 1\left[\tilde{\mathcal{D}}^{0,t-1}\right] \middle| \mathcal{F}_E\right] \right\| \leq \sqrt{\operatorname{Tr}\Sigma} \sqrt{\mathbb{P}\left(\tilde{\mathcal{D}}^{0,t-1,C} \middle| \mathcal{F}_E\right)} \tag{92}$$

In the last step, we have used Cauchy Schwarz inequality and the fact that $\eta_{-j_1}^{t-r_1,\top}$ is independent of $\mathcal{F}_E$. We combine the Equation above with Equation (91) and apply Jensen's inequality once again to conclude:

$$\left\|\mathbb{E}\left[\widetilde{\mathrm{Cr}}(t,r_1,j_1,r_2,j_2)1\left[\tilde{\mathcal{D}}^{0,t-1}\right]\right]\right\| \le 4\gamma^2 R\,\mathrm{Tr}(\Sigma)\sqrt{\mathbb{P}\left[\tilde{\mathcal{D}}^{0,t-1,C}\right]} \le 4\gamma^2 R\frac{\mathrm{Tr}(\Sigma)}{T^{\alpha/2}} \qquad (93)$$

In the last step, we have used Lemma 9 to bound $\mathbb{P}\left(\tilde{\mathcal{D}}^{0,t-1,C}\right)$. Summing over all the indices $(r_1,j_1,r_2,j_2)$, we conclude the statement of the lemma.

$\square$

**Lemma 24.** *We have:*

$$\mathbb{E}\left[\sum_{r=1}^{t}\sum_{j=0}^{B-1}\widetilde{\mathrm{Dg}}(t,r,j)1\left[\tilde{\mathcal{D}}^{0,t-1}\right]\right] \preceq 4\gamma^2\,\mathrm{Tr}(\Sigma)\mathbb{E}\left[\sum_{r=1}^{t}\sum_{j=0}^{B-1}\left(\prod_{s=1}^{r-1}\tilde{H}_{0,B-1}^{t-s,\top}\right)\tilde{H}_{j+1,B-1}^{t-r,\top}\tilde{X}_{-j}^{t-r}.\right.$$
$$\left.\tilde{X}_{-j}^{t-r,\top}\tilde{H}_{j+1,B-1}^{t-r}\left(\prod_{s=r-1}^{1}\tilde{H}_{0,B-1}^{t-s}\right)1\left[\tilde{\mathcal{D}}^{0,t-1}\right]\right] + \delta_{\mathrm{Dg}}I \qquad (94)$$

*and*

$$\mathbb{E}\left[\sum_{r=1}^{t}\sum_{j=0}^{B-1}\widetilde{\mathrm{Dg}}(t,r,j)1\left[\tilde{\mathcal{D}}^{0,t-1}\right]\right] \succeq 4\gamma^2\,\mathrm{Tr}(\Sigma)\mathbb{E}\left[\sum_{r=1}^{t}\sum_{j=0}^{B-1}\left(\prod_{s=1}^{r-1}\tilde{H}_{0,B-1}^{t-s,\top}\right)\tilde{H}_{j+1,B-1}^{t-r,\top}\tilde{X}_{-j}^{t-r}.\right.$$
$$\left.\tilde{X}_{-j}^{t-r,\top}\tilde{H}_{j+1,B-1}^{t-r}\left(\prod_{s=r-1}^{1}\tilde{H}_{0,B-1}^{t-s}\right)1\left[\tilde{\mathcal{D}}^{0,t-1}\right]\right] - \delta_{\mathrm{Dg}}I \qquad (95)$$

*where*

$$\delta_{\mathrm{Dg}} \equiv \delta_{\mathrm{Dg}}(T,\Sigma,R,\mu_4) = 4\gamma^2(Bt)R\sqrt{\mu_4}\frac{1}{T^{\alpha/2}} \qquad (96)$$

*Proof.* The evaluation of expectations is clear when there is no indicator $1\left[\tilde{\mathcal{D}}^{0,t-1}\right]$ within the expectation. We will now deal with it just like in the proof of Lemma 23. Consider $\widetilde{\mathrm{Dg}}(t,r,j)$. For the sake of convenience, only in this proof, we will use the following notation:

$$\widetilde{\mathrm{Dg}}(t,r,j) = 4\gamma^2\left\|\eta_{-j}^{t-r}\right\|^2 E\,.$$

Where the random PSD matrix $E$ is unconditionally independent of $\eta_{-j}^{t-r}$. Let $\mathcal{M} = \{\|E\| \le R\}$. Conditioned on the event $\tilde{\mathcal{D}}^{0,t-1}$, the event $\mathcal{M}$ holds almost surely. Let $\mathcal{F}_E = \sigma(E)$.

Now consider:

$$\mathbb{E}\left[\widetilde{\mathrm{Dg}}(t,r,j)1\left[\tilde{\mathcal{D}}^{0,t-1}\right]\right] = \mathbb{E}\left[\widetilde{\mathrm{Dg}}(t,r,j)1\left[\tilde{\mathcal{D}}^{0,t-1}\right]1\left[\mathcal{M}\right]\right]$$
$$= 4\gamma^2\mathbb{E}\left[\left\|\eta_{-j}^{t-r}\right\|^2 E1\left[\tilde{\mathcal{D}}^{0,t-1}\right]1\left[\mathcal{M}\right]\right]$$
$$= 4\gamma^2\mathbb{E}\left[\mathbb{E}\left[\left\|\eta_{-j}^{t-r}\right\|^2 1\left[\tilde{\mathcal{D}}^{0,t-1}\right]\Big|\mathcal{F}_E\right]E1\left[\mathcal{M}\right]\right] \qquad (97)$$

It can be easily shown via similar techniques used in Lemma 23 that:

$$\mathrm{Tr}(\Sigma) - \sqrt{\mu_4}\sqrt{\mathbb{P}\left(\tilde{\mathcal{D}}^{0,t-1,C}\big|\mathcal{F}_E\right)} \le \mathbb{E}\left[\left\|\eta_{-j}^{t-r}\right\|^2 1\left[\tilde{\mathcal{D}}^{0,t-1}\right]\Big|\mathcal{F}_E\right] \le \mathrm{Tr}(\Sigma)$$

Using this in Equation (97), we conclude:

$$\mathbb{E}\left[\widetilde{\mathrm{Dg}}(t,r,j)1\left[\tilde{\mathcal{D}}^{0,t-1}\right]\right] \preceq 4\gamma^2\,\mathrm{Tr}(\Sigma)\mathbb{E}\left[E1\left[\mathcal{M}\right]\right]$$

$$= 4\gamma^2\,\mathrm{Tr}(\Sigma)\mathbb{E}\left[E1\left[\mathcal{M}\right]1\left[\tilde{\mathcal{D}}^{0,t-1}\right] + E1\left[\mathcal{M}\right]1\left[\tilde{\mathcal{D}}^{0,t-1,C}\right]\right]$$

$$= 4\gamma^2\,\mathrm{Tr}(\Sigma)\mathbb{E}\left[E1\left[\tilde{\mathcal{D}}^{0,t-1}\right] + E1\left[\mathcal{M}\right]1\left[\tilde{\mathcal{D}}^{0,t-1,C}\right]\right]$$

$$\preceq 4\gamma^2\,\mathrm{Tr}\,\Sigma\mathbb{E}\left[E1\left[\hat{\mathcal{D}}^{0,t-1}\right]\right] + 4\gamma^2\,\mathrm{Tr}(\Sigma)R\frac{I}{T^\alpha} \qquad (98)$$

In the third step, we have used the fact that $\tilde{\mathcal{D}}^{0,t-1} \subseteq \mathcal{M}$. In the last step we have used the fact that $E$ is PSD and over the event $\mathcal{M}$, $E \preceq RI$. We have used Lemma 9 to bound $\mathbb{P}(\tilde{\mathcal{D}}^{0,t-1,C})$. Using a similar technique as above, we can show that:

$$\mathbb{E}\left[\widetilde{\mathrm{Dg}}(t,r,j)1\left[\tilde{\mathcal{D}}^{0,t-1}\right]\right] \succeq 4\gamma^2\,\mathrm{Tr}\,\Sigma\mathbb{E}\left[E1\left[\tilde{\mathcal{D}}^{0,t-1}\right]\right] - 4\gamma^2\frac{\sqrt{\mu_4}R}{T^{\alpha/2}}I \qquad (99)$$

Note that $\frac{\sqrt{\mu_4}R}{T^{\alpha/2}} \geq \frac{\mathrm{Tr}(\Sigma)R}{T^\alpha}$. Summing over $r,j$ and combining Equations (99) and (98), we conclude the result.

$\square$

For convenience, define $K^s := \sum_{j=0}^{B-1}\tilde{H}_{j+1,B-1}^{s,\top}\tilde{X}_{-j}^s\tilde{X}_{-j}^{s,\top}\tilde{H}_{j+1,B-1}^s$

**Claim 1.** *Suppose $\gamma < \frac{1}{R}$. Under the event $\tilde{\mathcal{D}}^{0,t-1}$, for every $s \leq t-1$ we must have:*

$$\frac{I - \tilde{H}_{0,B-1}^{s,\top}\tilde{H}_{0,B-1}^s}{4\gamma} \preceq K^s \preceq \frac{I - \tilde{H}_{0,B-1}^{s,\top}\tilde{H}_{0,B-1}^s}{\hat{\gamma}}$$

*Where $\hat{\gamma} = 4\gamma(1-\gamma R)$*

*Proof.* In the entire proof, we suppose that the event $\tilde{\mathcal{D}}^{0,t-1}$ holds. Consider:

$$\tilde{H}_{j,B-1}^{s,\top}\tilde{H}_{j,B-1}^s + 4\gamma\tilde{H}_{j+1,B-1}^{s,\top}\tilde{X}_{-j}^s\tilde{X}_{-j}^{s,\top}\tilde{H}_{j+1,B-1}^s$$

$$= \tilde{H}_{j+1,B-1}^{s,\top}\left(I - \left(4\gamma - 4\gamma^2\|\tilde{X}_{-j}^s\|^2\right)\tilde{X}_{-j}^s\tilde{X}_{-j}^{s,\top}\right)\tilde{H}_{j+1,B-1}^s + 4\gamma\tilde{H}_{j+1,B-1}^{s,\top}\tilde{X}_{-j}^s\tilde{X}_{-j}^{s,\top}\tilde{H}_{j+1,B-1}^s$$

$$= \tilde{H}_{j+1,B-1}^{s,\top}\left(I + 4\gamma^2\|\tilde{X}_{-j}^s\|^2\tilde{X}_{-j}^s\tilde{X}_{-j}^{s,\top}\right)\tilde{H}_{j+1,B-1}^s$$

$$\succeq \tilde{H}_{j+1,B-1}^{s,\top}\tilde{H}_{j+1,B-1}^s \qquad (100)$$

Using the recursion in Equation (100), we show that:

$$\tilde{H}_{0,B-1}^{s,\top}\tilde{H}_{0,B-1}^s + 4\gamma K^s \succeq I.$$

This establishes the lower bound. To establish the upper bound, we consider

$$\tilde{H}_{j,B-1}^{s,\top}\tilde{H}_{j,B-1}^s + \hat{\gamma}\tilde{H}_{j+1,B-1}^{s,\top}\tilde{X}_{-j}^s\tilde{X}_{-j}^{s,\top}\tilde{H}_{j+1,B-1}^s.$$

Following similar technique used to establish Equation (100), using the fact that under the event $\tilde{\mathcal{D}}^{0,t-1}$ we have $\|\tilde{X}_{-j}^s\|^2 \leq R$ we show that:

$$\tilde{H}_{j,B-1}^{s,\top}\tilde{H}_{j,B-1}^s + \hat{\gamma}\tilde{H}_{j+1,B-1}^{s,\top}\tilde{X}_{-j}^s\tilde{X}_{-j}^{s,\top}\tilde{H}_{j+1,B-1}^s \preceq \tilde{H}_{j+1,B-1}^{s,\top}\tilde{H}_{j+1,B-1}^s.$$

Using a similar recursion as before, we establish that:

$$\tilde{H}_{0,B-1}^{s,\top}\tilde{H}_{0,B-1}^s + \hat{\gamma}K^s \preceq I.$$

$\square$

We are now ready to bound the first term in (94):

$$\mathbb{E}\left[\sum_{r=1}^{t}\left(\prod_{s=1}^{r-1}\tilde{H}_{0,B-1}^{t-s,\top}\right)K^{t-r}\left(\prod_{s=r-1}^{1}\tilde{H}_{0,B-1}^{t-s}\right)1\left[\tilde{\mathcal{D}}^{0,t-1}\right]\right] \tag{101}$$

It is easy to show via. telescoping sum argument that:

$$\sum_{r=1}^{t}\left(\prod_{s=1}^{r-1}\tilde{H}_{0,B-1}^{t-s,\top}\right)\left(I-\tilde{H}_{0,B-1}^{t-r,\top}\tilde{H}_{0,B-1}^{t-r}\right)\left(\prod_{s=r-1}^{1}\tilde{H}_{0,B-1}^{t-s}\right) = I-\left(\prod_{s=1}^{t}\tilde{H}_{0,B-1}^{t-s,\top}\right)\left(\prod_{s=t}^{1}\tilde{H}_{0,B-1}^{t-s}\right) \tag{102}$$

We then use Claim 1 to show that under the event $\tilde{\mathcal{D}}^{0,t-1}$, we must have:

$$\frac{I-\left(\prod_{s=1}^{t}\tilde{H}_{0,B-1}^{t-s,\top}\right)\left(\prod_{s=t}^{1}\tilde{H}_{0,B-1}^{t-s}\right)}{4\gamma} \preceq \sum_{r=1}^{t}\left(\prod_{s=1}^{r-1}\tilde{H}_{0,B-1}^{t-s,\top}\right)K^{t-r}\left(\prod_{s=r-1}^{1}\tilde{H}_{0,B-1}^{t-s}\right) \tag{103}$$

And:

$$\sum_{r=1}^{t}\left(\prod_{s=1}^{r-1}\tilde{H}_{0,B-1}^{t-s,\top}\right)K^{t-r}\left(\prod_{s=r-1}^{1}\tilde{H}_{0,B-1}^{t-s}\right) \preceq \frac{I-\left(\prod_{s=1}^{t}\tilde{H}_{0,B-1}^{t-s,\top}\right)\left(\prod_{s=t}^{1}\tilde{H}_{0,B-1}^{t-s}\right)}{\hat{\gamma}} \tag{104}$$

Finally, combining Lemma 23, Lemma 24, claim 1, Equations (103), (104) and the bound on $\mu_4$ (stated after assumption 3 in section 2) along with $\hat{\gamma} = 4\gamma(1-\gamma R)$ we get the statement of the proposition.

$\square$

# I  Proof of Proposition 2

Before delving into the proof, we note some useful results below.

**Lemma 25.** *For any random matrix $B \in \mathbb{R}^{d \times d}$ we have that*

$$\mathbb{E}\left[B^{\top}\right]\mathbb{E}\left[B\right] \preceq \mathbb{E}\left[B^{\top}B\right] \tag{105}$$

*Hence*

$$\|\mathbb{E}\left[B\right]\| \leq \sqrt{\|\mathbb{E}\left[B^{\top}B\right]\|} \tag{106}$$

*Proof.* Note that for any vector $x \in \mathbb{R}^d$ we have

$$x^{\top}\mathbb{E}\left[B^{\top}\right]\mathbb{E}\left[B\right]x = \|\mathbb{E}\left[Bx\right]\|^2 \leq \mathbb{E}\left[\|Bx\|^2\right] = x^{\top}\mathbb{E}\left[B^{\top}B\right]x \tag{107}$$

$\square$

**Lemma 26.** *Let $\gamma RB \leq \frac{c}{6}$ for $0 < c < 1$. The there are constants $c_1, c_2 > 0$ such that for $T^{\alpha/2} > c_1\frac{\sqrt{M_4}}{\sigma_{\min}(G)}$ we have*

$$\|\mathcal{H}\| \leq \sqrt{1-c_2\gamma B\sigma_{\min}(G)} \leq 1-\frac{c_2}{2}\gamma B\sigma_{\min}(G) \tag{108}$$

*with $1-c_2\gamma B\sigma_{\min}(G) > 0$.*

*Proof.* Note that $\mathcal{H}$ can be written as $\mathcal{H} = \mathbb{E}\left[\tilde{H}_{0,B-1}^{0}1[\tilde{\mathcal{D}}_{-0}^{0}]\right]$. First we use Lemma 25 to get

$$\|\mathcal{H}\| \leq \sqrt{\left\|\mathbb{E}\left[\tilde{H}_{0,B-1}^{0,\top}\tilde{H}_{0,B-1}^{0}1[\tilde{\mathcal{D}}_{-0}^{0}]\right]\right\|} \tag{109}$$

Then, from Lemma 29 we can show that there are constants $c_1, c_2 > 0$ such that

$$\left\| \mathbb{E}\left[ \tilde{H}_{0,B-1}^{0,\top} \tilde{H}_{0,B-1}^0 1[\tilde{\mathcal{D}}_{-0}^0] \right] \right\| \le \left( 1 - c_1 \gamma B \sigma_{\min}(G) + c_2 \gamma B \sqrt{M_4} \frac{1}{T^{\alpha/2}} \right) \tag{110}$$

Now choosing $T$ such that $T^{\alpha/2} > \frac{c_2 \sqrt{M_4}}{2 c_1 \sigma_{\min}(G)}$ we get

$$\left\| \mathbb{E}\left[ \tilde{H}_{0,B-1}^{0,\top} \tilde{H}_{0,B-1}^0 1[\tilde{\mathcal{D}}_{-0}^0] \right] \right\| \le (1 - c_3 \gamma B \sigma_{\min}(G)) \tag{111}$$

where $c_3$ is such that the RHS in (111) is positive. Hence the claim follows.

$\square$

*Proof of Proposition 2.* We will prove the proposition only for $a = 0$. The arguments for general $a$ are exactly the same.

For simplicity, we denote

$$\hat{\tilde{A}}_N^v \equiv \left( \hat{\tilde{A}}_{0,N}^v \right) \tag{112}$$

From recursion (6) we have the following relation between $\left( \tilde{A}_B^{t_2-1,v} \right)$ and $\left( \tilde{A}_B^{t_1-1,v} \right)$ for $t_2 > t_1$

$$\left( \tilde{A}_B^{t_2-1,v} \right) = \left( \tilde{A}_B^{t_1-1,v} \right) \left( \prod_{s=t_2-t_1}^1 \tilde{H}_{0,B-1}^{t_2-s} \right) +$$
$$2\gamma \sum_{r=1}^{t_2-t_1} \sum_{j=0}^{B-1} \eta_{-j}^{t_2-r} \tilde{X}_{-j}^{t_2-r,\top} \tilde{H}_{j+1,B-1}^{t_2-r} \left( \prod_{s=r-1}^1 \tilde{H}_{0,B-1}^{t_2-s} \right). \tag{113}$$

Hence we have

$$\left( \tilde{A}_B^{t_1-1,v} \right)^\top \left( \tilde{A}_B^{t_2-1,v} \right) = \left( \tilde{A}_B^{t_1-1,v} \right)^\top \left( \tilde{A}_B^{t_1-1,v} \right) \left( \prod_{s=t_2-t_1}^1 \tilde{H}_{0,B-1}^{t_2-s} \right) +$$
$$2\gamma \left( \tilde{A}_B^{t_1-1,v} \right)^\top \sum_{r=1}^{t_2-t_1} \sum_{j=0}^{B-1} \eta_{-j}^{t_2-r} \tilde{X}_{-j}^{t_2-r,\top} \tilde{H}_{j+1,B-1}^{t_2-r} \left( \prod_{s=r-1}^1 \tilde{H}_{0,B-1}^{t_2-s} \right). \tag{114}$$

The second term in (114) is bounded in claim 2

The first term in (114) can be analyzed using independence as follows.

$$\mathbb{E}\left[ \left( \tilde{A}_B^{t_1-1,v} \right)^\top \left( \tilde{A}_B^{t_1-1,v} \right) 1\left[ \tilde{\mathcal{D}}^{0,t_1-1} \right] \left( \prod_{s=t_2-t_1}^1 \tilde{H}_{0,B-1}^{t_2-s} \right) 1\left[ \tilde{\mathcal{D}}^{t_1,N-1} \right] \right]$$

$$= \tilde{V}_{t_1-1} \mathbb{E}\left[ \left( \prod_{s=t_2-t_1}^1 \tilde{H}_{0,B-1}^{t_2-s} \right) 1\left[ \tilde{\mathcal{D}}^{t_1,N-1} \right] \right]$$

$$= \tilde{V}_{t_1-1} \mathbb{E}\left[ \left( \prod_{s=t_2-t_1}^1 \tilde{H}_{0,B-1}^{t_2-s} \right) 1\left[ \tilde{\mathcal{D}}^{t_1,t_2-1} \right] \right] \mathbb{E}\left[ 1\left[ \tilde{\mathcal{D}}^{t_2,N-1} \right] \right]$$

$$= \tilde{V}_{t_1-1} \left( \prod_{s=t_2-t_1}^1 \mathbb{E}\left[ \tilde{H}_{0,B-1}^{t_2-s} 1\left[ \tilde{\mathcal{D}}^{t_1,t_2-1} \right] \right] \right) \mathbb{E}\left[ 1\left[ \tilde{\mathcal{D}}^{t_2,N-1} \right] \right] = \tilde{V}_{t_1-1} \mathcal{H}^{t_2-t_1} \mathbb{E}\left[ 1\left[ \tilde{\mathcal{D}}^{t_2,N-1} \right] \right]$$

$$= \tilde{V}_{t_1-1} \mathcal{H}^{t_2-t_1} - \tilde{V}_{t_1-1} \mathcal{H}^{t_2-t_1} \mathbb{E}\left[ 1\left[ \tilde{\mathcal{D}}^{t_2,N-1,C} \right] \right]. \tag{115}$$

Note that,

$$\left( \tilde{A}_B^{t_1-1,v} \right)^\top \left( \tilde{A}_B^{t_1-1,v} \right) \preceq 4\gamma^2 (Bt_1) \sum_{r=1}^{t_1} \sum_{j=0}^{B-1} \left\| \eta_{-j}^{t_1-r} \right\|^2 \cdot$$

$$\left( \prod_{s=1}^{r-1} \tilde{H}_{0,B-1}^{t_1-s,\top} \right) \tilde{H}_{j+1,B-1}^{t_1-r,\top} \tilde{X}_{-j}^{t_1-r} \tilde{X}_{-j}^{t_1-r,\top} \tilde{H}_{j+1,B-1}^{t_1-r} \left( \prod_{s=r-1}^1 \tilde{H}_{0,B-1}^{t_1-s} \right). \tag{116}$$

From equation (116), we have:

$$\left\| \tilde{V}_{t_1-1} \right\| \le c\gamma^2 (Bt_1)^2 Rd\sigma_{\max}, \tag{117}$$

and further, $\|\mathcal{H}\| < 1$ from Lemma 26. Hence,

$$\left\| \tilde{V}_{t_1-1} \mathcal{H}^{t_2-t_1} \mathbb{E}\left[ 1\left[ \tilde{\mathcal{D}}^{t_2,N-1,C} \right] \right] \right\| \le \left\| \tilde{V}_{t_1-1} \mathcal{H}^{t_2-t_1} \right\| \frac{1}{T^\alpha} \le c\gamma^2 (Bt_1)^2 Rd\sigma_{\max} \frac{1}{T^\alpha}.$$

For brevity, given a matrix $Q \in \mathbb{R}^{d \times d}$, let,

$$\mathrm{Sym}\,(Q) = Q + Q^\top. \tag{118}$$

Combining everything so far, we have, for $t_2 > t_1$:

$$\mathrm{Sym}\left( \mathbb{E}\left[ \left( \tilde{A}_B^{t_1-1,v} \right)^\top \left( \tilde{A}_B^{t_2-1,v} \right) 1\left[ \tilde{\mathcal{D}}^{0,N-1} \right] \right] \right)$$
$$\preceq \mathrm{Sym}\left( \tilde{V}_{t_1-1} \mathcal{H}^{t_2-t_1} \right) + c_1 \gamma^2 (Bt_1)^2 Rd\sigma_{\max} \frac{1}{T^\alpha} I +$$
$$\left( c_3 \gamma^2 B^2 t_1 t_2 Rd\sigma_{\max} \frac{1}{T^{\alpha/2}} \right) I \tag{119}$$

Since $Bt_2 \le T$ we get:

$$\mathrm{Sym}\left( \mathbb{E}\left[ \left( \tilde{A}_B^{t_1-1,v} \right)^\top \left( \tilde{A}_B^{t_2-1,v} \right) 1\left[ \tilde{\mathcal{D}}^{0,N-1} \right] \right] \right) \preceq \mathrm{Sym}\left( \tilde{V}_{t_1-1} \mathcal{H}^{t_2-t_1} \right) +$$
$$c_3 \gamma^2 T^2 Rd\sigma_{\max} \frac{1}{T^{\alpha/2}} I. \tag{120}$$

Therefore we have,

$$\frac{1}{N^2} \sum_{t_1 \ne t_2} \mathbb{E}\left[ \left( \tilde{A}_B^{t_1-1,v} \right)^\top \left( \tilde{A}_B^{t_2-1,v} \right) \right] \preceq \frac{1}{N^2} \sum_{t_1=1}^{N-1} \mathrm{Sym}\left( \tilde{V}_{t_1-1} \left( \sum_{t_2>t_1} \mathcal{H}^{t_2-t_1} \right) \right)$$
$$+ c_3 \gamma^2 T^2 Rd\sigma_{\max} \frac{1}{T^{\alpha/2}} I.$$

Next observe that,

$$\frac{1}{N^2} \sum_{t=1}^N \tilde{V}_{t-1} + \frac{1}{N^2} \sum_{t_1=1}^{N-1} \mathrm{Sym}\left( \tilde{V}_{t_1-1} \left( \sum_{t_2>t_1} \mathcal{H}^{t_2-t_1} \right) \right)$$
$$= \frac{1}{N^2} \sum_{t=1}^N \tilde{V}_{t-1} + \frac{1}{N^2} \sum_{t_1=1}^{N-1} \mathrm{Sym}\left( \tilde{V}_{t_1-1} \left( \sum_{s=1}^{N-t_1} \mathcal{H}^s \right) \right)$$
$$\preceq \frac{1}{N^2} \sum_{t=1}^N \mathrm{Sym}\left( \tilde{V}_{t-1} \left( \sum_{s=0}^{N-t} \mathcal{H}^s \right) \right).$$

Hence, substituting in (40), we obtain:

$$\mathbb{E}\left[ \left( \hat{A}_N^v \right)^\top \left( \hat{A}_N^v \right) 1\left[ \tilde{\mathcal{D}}^{0,N-1} \right] \right] \preceq \frac{1}{N^2} \sum_{t=1}^N \mathrm{Sym}\left( \tilde{V}_{t-1} \left( \sum_{s=0}^{N-t} \mathcal{H}^s \right) \right) + \tag{121}$$
$$c_3 \gamma^2 T^2 Rd\sigma_{\max} \frac{1}{T^{\alpha/2}} I. \tag{122}$$

From Equations (121)-(122) we obtain (41).

Now $\sum_{s=0}^{N-t} \mathcal{H}^s = (I - \mathcal{H})^{-1}(I - \mathcal{H}^{N-t+1})$ since from Lemma 26 we know that $\|\mathcal{H}\| < 1$ for large $T$. Thus we get (42).

$\square$

## I.1 Claims

**Claim 2.** *For $\gamma \leq \frac{1}{2R}$ we have*

$$\left\| \mathbb{E}\left[ 2\gamma \left(\tilde{A}_B^{t_1-1,v}\right)^\top \sum_{r=1}^{t_2-t_1} \sum_{j=0}^{B-1} \eta_{-j}^{t_2-r} \tilde{X}_{-j}^{t_2-r,\top} \tilde{H}_{j+1,B-1}^{t_2-r} \left(\prod_{s=r-1}^{1} \tilde{H}_{0,B-1}^{t_2-s}\right) \right] 1\left[\tilde{\mathcal{D}}^{0,N-1}\right] \right\|$$

$$\leq c_1 \gamma^2 B^2 t_1 t_2 R d\sigma_{\max} \frac{1}{T^{\alpha/2}} \tag{123}$$

*for some constant $c_1 > 0$.*

*Proof.* The proof is similar to the proof of Lemma 23.

$\square$

## J  Proof of Theorem 14

*Proof of Theorem 14.* We start with the following

$$\left(\tilde{A}_b^{t-1,b} - A^*\right)^\top \left(\tilde{A}_b^{t-1,b} - A^*\right) = \left(\prod_{s=1}^{t} \tilde{H}_{0,B-1}^{t-s,\top}\right) (A_0 - A^*)^\top (A_0 - A) \left(\prod_{s=t}^{1} \tilde{H}_{0,B-1}^{t-s}\right)$$

$$\preceq \|A_0 - A^*\|^2 \left(\prod_{s=1}^{t} \tilde{H}_{0,B-1}^{t-s,\top}\right) \left(\prod_{s=t}^{1} \tilde{H}_{0,B-1}^{t-s}\right) \tag{124}$$

From Lemma 29 we can show that there are constants $c_1, c_2 > 0$ such that

$$\left\| \mathbb{E}\left[ \left(\prod_{s=1}^{t} \tilde{H}_{0,B-1}^{t-s,\top}\right) \left(\prod_{s=t}^{1} \tilde{H}_{0,B-1}^{t-s}\right) 1\left[\tilde{\mathcal{D}}^{0,t-1}\right] \right] \right\|$$

$$\leq \left(1 - c_1 \gamma B \sigma_{\min}(G) + c_2 \gamma B \sqrt{M_4} \frac{1}{T^{\alpha/2}}\right)^t. \tag{125}$$

Now choosing $T$ such that $T^{\alpha/2} > \frac{c_2 \sqrt{M_4}}{2 c_1 \sigma_{\min}(G)}$ we get,

$$\left\| \mathbb{E}\left[ \left(\prod_{s=1}^{t} \hat{H}_{0,B-1}^{t-s,\top}\right) \left(\prod_{s=t}^{1} \hat{H}_{0,B-1}^{t-s}\right) \right] \right\| \leq \left(1 - c_3 \gamma B \sigma_{\min}(G)\right)^t. \tag{126}$$

Thus we get the theorem.

$\square$

## K  Proof of Theorem 15

*Proof of Theorem 15.* We use the following inequality that is obtained from Lemma 25

$$\left(\hat{\tilde{A}}_{a,N}^b - A^*\right)^\top \left(\hat{\tilde{A}}_{a,N}^b - A^*\right) \preceq \frac{1}{N-a} \sum_{t=a+1}^{N} \left(\tilde{A}_B^{t-1,b} - A^*\right)^\top \left(\tilde{A}_B^{t-1,b} - A^*\right) \tag{127}$$

Therefore

$$\mathbb{E}\left[ \left(\hat{\tilde{A}}_{a,N}^b - A^*\right)^\top \left(\hat{\tilde{A}}_{a,N}^b - A^*\right) 1\left[\tilde{\mathcal{D}}^{0,N-1}\right] \right]$$

$$\preceq \frac{1}{N-a} \sum_{t=a+1}^{N} \mathbb{E}\left[ \left(\tilde{A}_B^{t-1,b} - A^*\right)^\top \left(\tilde{A}_B^{t-1,b} - A^*\right) 1\left[\tilde{\mathcal{D}}^{0,N-1}\right] \right]$$

$$\preceq \frac{1}{N-a} \sum_{t=a+1}^{N} \mathbb{E}\left[ \left(\tilde{A}_B^{t-1,b} - A^*\right)^\top \left(\tilde{A}_B^{t-1,b} - A^*\right) 1\left[\tilde{\mathcal{D}}^{0,t-1}\right] \right] \tag{128}$$

Now using theorem 14, we get

$$
\mathbb{E}\left[\left(\hat{\tilde{A}}^b_{a,N} - A^*\right)^\top \left(\hat{\tilde{A}}^b_{a,N} - A^*\right) \mathbb{1}\left[\tilde{\mathcal{D}}^{0,N-1}\right]\right] \preceq
$$

$$
\left(\frac{1}{N-a}\frac{(1-c_1\gamma B\sigma_{\min}(G))^{a+1}}{c_1\gamma B\sigma_{\min}(G)}\right)\|A_0 - A^*\|^2 I \tag{129}
$$

Hence using $1 - x \le e^{-x}$ we get

$$
\left\|\mathbb{E}\left[\left(\hat{\tilde{A}}^b_{a,N} - A^*\right)^\top \left(\hat{\tilde{A}}^b_{a,N} - A^*\right) \mathbb{1}\left[\tilde{\mathcal{D}}^{0,N-1}\right]\right]\right\|
$$

$$
\le c\frac{1}{B(N-a)}\frac{e^{-cB\gamma\sigma_{\min}(G)a}}{\gamma\sigma_{\min}(G)}\|A_0 - A^*\|^2 \tag{130}
$$

$\square$

## L  Operator Norm Inequalities

In this section, we develop the concentration inequalities necessary to obtain bounds on $\mathcal{L}_{\text{op}}$. Consider Equation (20)

$$
\left(\tilde{A}^{t-1,v}_B\right) = 2\gamma\sum_{r=1}^{t}\sum_{j=0}^{B-1}\eta^{t-r}_{-j}\tilde{X}^{t-r,\top}_{-j}\tilde{H}^{t-r}_{j+1,B-1}\prod_{s=r-1}^{1}\tilde{H}^{t-s}_{0,B-1} \tag{131}
$$

Splitting the sum into $r = 1$ and $r = 2, \ldots, t$, it is easy to show the following recursion:

$$
\left(\tilde{A}^{t-1,v}_B\right) = 2\gamma\sum_{j=0}^{B-1}\eta^{t-1}_{-j}\tilde{X}^{t-1,\top}_{-j}\tilde{H}^{t-1}_{j+1,B-1} + \left(\tilde{A}^{t-2,v}_B\right)\tilde{H}^{t-1}_{0,B-1} \tag{132}
$$

We will consider the matrix $\Delta_{t-1} := 2\gamma\sum_{j=0}^{B-1}\eta^{t-1}_{-j}\tilde{X}^{t-1,\top}_{-j}\tilde{H}^{t-1}_{j+1,B-1}$. Recall the sequence of events $\tilde{\mathcal{D}}^{t-1}_{-j}$ for $j = 0, 1, \ldots, B-1$ as defined in Section B.1. We will pick $R$ as in Section 4 so that $\mathbb{P}(\tilde{\mathcal{D}}^{t-1}_{-0})$ is close to 1.

For the sake of clarity, we drop the dependence on $t$ while stating and proving some of the technical results since the events and random variables considered there are identically distributed for every $t$. That is, consider $\tilde{\mathcal{D}}_{-j}$ instead of $\tilde{\mathcal{D}}^{t-1}_{-j}$ and

$$
\Delta := 2\gamma\sum_{j=0}^{B-1}\eta_{-j}\tilde{X}^\top_{-j}\tilde{H}_{j+1,B-1}
$$

We will bound the exponential moment generating function of $\Delta$:

**Lemma 27.** *Suppose Assumption 2 holds and that $\gamma R < 1$. Let $\lambda \in \mathbb{R}$ and $x, y \in \mathbb{R}^d$ are arbitrary. Then, we have:*

  *1.*

$$
\mathbb{E}\left[\exp(\gamma\lambda^2 C_\mu\langle x, \Sigma x\rangle\langle y, \tilde{H}^\top_{0,B-1}\tilde{H}_{0,B-1}y\rangle + \lambda\langle x, \Delta y\rangle)|\tilde{\mathcal{D}}_{-0}\right]
$$

$$
\le \frac{\exp\left(\gamma\lambda^2 C_\mu\langle x, \Sigma x\rangle\|y\|^2\right)}{\mathbb{P}(\tilde{\mathcal{D}}_{-0})}
$$

  *2.*

$$
\mathbb{E}\left[\exp(\lambda\langle x, \Delta y\rangle)|\tilde{\mathcal{D}}_{-0}\right] \le \frac{\exp\left(\gamma\lambda^2 C_\mu\langle x, \Sigma x\rangle\|y\|^2\right)}{\mathbb{P}(\tilde{\mathcal{D}}_{-0})}
$$

*Where $C_\mu$ is as given in Assumption 2*

*Proof.* We will just prove item 1 since item 2 follows from it trivially as

$$\gamma \lambda^2 C_\mu \langle x, \Sigma x \rangle \langle y, \tilde{H}_{0,B-1}^\top \tilde{H}_{0,B-1} y \rangle \geq 0 \,.$$

For the sake of clarity, we will take:

$$\Xi_0 := \gamma \lambda^2 C_\mu \langle x, \Sigma x \rangle \langle y, \tilde{H}_{0,B-1}^\top \tilde{H}_{0,B-1} y \rangle$$

and more generally,

$$\Xi_k = \gamma \lambda^2 C_\mu \langle x, \Sigma x \rangle \langle y, \tilde{H}_{k,B-1}^\top \tilde{H}_{k,B-1} y \rangle$$

Consider $\Delta_{-k} := 2\gamma \sum_{j=k}^{B-1} \eta_{-j} \tilde{X}_{-j}^\top \tilde{H}_{j+1,B-1}$. We will first prove the following claim before bounding the exponential moment:

**Claim 3.** *Whenever* $\|\tilde{X}_{-k}\|^2 \leq R$ *and* $\gamma R < 1/2$, *we have:*

$$\Xi_k + 2\gamma^2 \lambda^2 C_\mu \langle x, \Sigma x \rangle \langle y, \tilde{H}_{k+1,B-1}^\top \tilde{X}_{-k} \tilde{X}_{-k}^\top \tilde{H}_{k+1,B-1} y \rangle \leq \Xi_{k+1}$$

*Proof.* We use the fact that $\tilde{H}_{k,B-1}^\top \tilde{H}_{k,B-1} = \tilde{H}_{k+1,B-1}^\top (I - 2\gamma \tilde{X}_{-k} \tilde{X}_{-k}^\top)^2 \tilde{H}_{k+1,B-1}$ to conclude that:

$$\Xi_k + 2\gamma^2 \lambda^2 C_\mu \langle x, \Sigma x \rangle \langle y, \tilde{H}_{k+1,B-1}^\top \tilde{X}_{-k} \tilde{X}_{-k}^\top \tilde{H}_{k+1,B-1} y \rangle$$
$$= \gamma \lambda^2 C_\mu \langle x, \Sigma x \rangle \langle y, \tilde{H}_{k+1,B-1}^\top \left( I - 2\gamma \tilde{X}_{-k} \tilde{X}_{-k}^\top + 4\gamma^2 \|\tilde{X}_{-k}\|^2 \tilde{X}_{-k} \tilde{X}_{-k}^\top \right) \tilde{H}_{k+1,B-1} y \rangle$$
$$\leq \gamma \lambda^2 C_\mu \langle x, \Sigma x \rangle \langle y, \tilde{H}_{k+1,B-1}^\top \tilde{H}_{k+1,B-1} y \rangle = \Xi_{k+1} \qquad (133)$$

In the second step we have used the fact that when $\gamma \|\tilde{X}_{-k}\|^2 \leq 1/2$, we have that

$$I - 2\gamma \tilde{X}_{-k} \tilde{X}_{-k}^\top + 4\gamma^2 \|\tilde{X}_{-k}\|^2 \tilde{X}_{-k} \tilde{X}_{-k}^\top \preceq I$$

$\square$

First note that $\Delta = 2\gamma \eta_0 \tilde{X}_0^\top \tilde{H}_{1,B-1} + \Delta_{-1}$. Now,

$$\mathbb{E}\left[ \exp(\Xi_0 + \lambda \langle x, \Delta y \rangle) | \tilde{\mathcal{D}}_{-0} \right] = \frac{1}{\mathbb{P}(\tilde{\mathcal{D}}_{-0})} \mathbb{E}\left[ \exp(\Xi_0 + \lambda \langle x, \Delta y \rangle) \mathbb{1}\left( \tilde{\mathcal{D}}_{-0} \right) \right]$$
$$= \frac{1}{\mathbb{P}(\tilde{\mathcal{D}}_{-0})} \mathbb{E}\left[ \exp\left( \Xi_0 + 2\lambda \gamma \langle x, \eta_{-0} \rangle \langle \tilde{X}_{-0}, \tilde{H}_{1,B-1} y \rangle + \lambda \langle x, \Delta_{-1} y \rangle \right) \mathbb{1}\left( \tilde{\mathcal{D}}_{-0} \right) \right]$$
$$\leq \frac{1}{\mathbb{P}(\tilde{\mathcal{D}}_{-0})} \mathbb{E}\left[ \exp\left( \Xi_0 + 2\gamma^2 \lambda^2 C_\mu \langle x, \Sigma x \rangle \langle y, \tilde{H}_{1,B-1}^\top \tilde{X}_{-0} \tilde{X}_{-0}^\top \tilde{H}_{1,B-1} y \rangle + \lambda \langle x, \Delta_{-1} y \rangle \right) \mathbb{1}\left( \tilde{\mathcal{D}}_{-0} \right) \right]$$
$$\leq \frac{1}{\mathbb{P}(\tilde{\mathcal{D}}_{-0})} \mathbb{E}\left[ \exp\left( \Xi_1 + \lambda \langle x, \Delta_{-1} y \rangle \right) \mathbb{1}\left( \tilde{\mathcal{D}}_{-0} \right) \right]$$
$$\leq \frac{1}{\mathbb{P}(\tilde{\mathcal{D}}_{-0})} \mathbb{E}\left[ \exp\left( \Xi_1 + \lambda \langle x, \Delta_{-1} y \rangle \right) \mathbb{1}\left( \tilde{\mathcal{D}}_{-1} \right) \right] \qquad (134)$$

In the first step we have used the definition of conditional expectation, in the third step we have used the fact that $\eta_{-0}$ is independent of $\tilde{\mathcal{D}}_{-0}$, $\Delta_{-1}$, $\tilde{X}_{-0}^\top \tilde{H}_{1,B-1}$, and $\Delta_{-1}$ and have applied the sub-Gaussianity from Assumption 2. In the fourth step, using the fact under the event $\tilde{\mathcal{D}}_{-0}$, $\|\tilde{X}_{-0}\|^2 \leq R$ we have applied Claim 3. In the final step, we have used the fact that $\tilde{\mathcal{D}}_{-0} \subseteq \tilde{\mathcal{D}}_{-1}$. We proceed by induction over Equation (134) to conclude the result.

$\square$

We now consider the matrix $\tilde{H}_{0,B-1}$ under the event $\tilde{\mathcal{D}}_{-0}$.

**Lemma 28.** *Suppose that* $\gamma RB < \frac{1}{6}$. *Then, under the event* $\tilde{\mathcal{D}}_{-0}$, *we have:*

$$I - 4\gamma \left( 1 + \frac{2\gamma BR}{1-4\gamma BR} \right) \sum_{i=0}^{B-1} \tilde{X}_{-i} \tilde{X}_{-i}^\top \preceq \tilde{H}_{0,B-1}^\top \tilde{H}_{0,B-1} \preceq I - 4\gamma \left( 1 - \frac{2\gamma BR}{1-4\gamma BR} \right) \sum_{i=0}^{B-1} \tilde{X}_{-i} \tilde{X}_{-i}^\top$$

*Proof.* By definition, we have: $\tilde{H}_{0,B-1} = \prod_{j=0}^{B-1}(I - 2\gamma\tilde{X}_{-j}\tilde{X}_{-j}^\top)$. Expanding out the product, we get an expression of the form:

$$\tilde{H}_{0,B-1}^\top \tilde{H}_{0,B-1} = I - 4\gamma\sum_{i=0}^{B-1}\tilde{X}_{-i}\tilde{X}_{-i}^\top + (2\gamma)^2\sum_{i,j}\tilde{X}_{-i}\tilde{X}_{-i}^\top\tilde{X}_{-j}\tilde{X}_{-j}^\top + \ldots \quad (135)$$

Here, the summation $\sum_{i,j}$ is over all possible combinations possible when the product is expanded and $\ldots$ denotes higher order terms of the form $\tilde{X}_{-i_1}\tilde{X}_{-i_1}^\top \ldots \tilde{X}_{-i_k}\tilde{X}_{-i_k}^\top$

**Claim 4.** *Assume $k \geq 2$ and $i_1, \ldots, i_k \in \{0, \ldots, B-1\}$. Under the event $\tilde{\mathcal{D}}_{-0}$, for any $x \in \mathbb{R}^d$, we have:*

$$\left| x^\top\tilde{X}_{-i_1}\tilde{X}_{-i_1}^\top \ldots \tilde{X}_{-i_k}\tilde{X}_{-i_k}^\top x \right| \leq \frac{R^{k-1}}{2}\left[ x^\top\tilde{X}_{-i_1}\tilde{X}_{-i_1}^\top x + x^\top\tilde{X}_{-i_k}\tilde{X}_{-i_k}^\top x \right]$$

*Proof.* This follows from an application of AM-GM inequality. It is clear by Cauchy-Schwarz inequality that $|\langle\tilde{X}_{i_l}, \tilde{X}_{i_{l+1}}\rangle| \leq R$, which implies:

$$\left| x^\top\tilde{X}_{-i_1}\tilde{X}_{-i_1}^\top \ldots \tilde{X}_{-i_k}\tilde{X}_{-i_k}^\top x \right| \leq R^{k-1}\left| \left[ x^\top\tilde{X}_{-i_1}\tilde{X}_{-i_k}^\top x \right] \right| \leq \frac{R^{k-1}}{2}\left[ \langle x, \tilde{X}_{-i_1}\rangle^2 + \langle\tilde{X}_{-i_k}, x\rangle^2 \right].$$

Where the last inequality follows from an application of the AM-GM inequality. $\qquad\square$

From Claim 4, we conclude that:

$$\sum_{i_1,\ldots,i_k}\tilde{X}_{-i_1}\tilde{X}_{-i_1}^\top \ldots \tilde{X}_{-i_k}\tilde{X}_{-i_k}^\top \preceq (2B)^{k-1}R^{k-1}\sum_{i=0}^{B-1}\tilde{X}_{-i}\tilde{X}_{-i}^\top$$

Plugging this into Equation (135), we have that under the event $\tilde{\mathcal{D}}_{-0}$:

$$\tilde{H}_{0,B-1}^\top\tilde{H}_{0,B-1} \preceq I - 4\gamma\sum_{i=0}^{B-1}\sum_{i=0}^{B-1}\tilde{X}_{-i}\tilde{X}_{-i}^\top + \sum_{k=2}^{2B}(2\gamma)^k(2B)^{k-1}R^{k-1}\sum_{i=0}^{B-1}\tilde{X}_{-i}\tilde{X}_{-i}^\top$$

$$\preceq I - 4\gamma\sum_{i=0}^{B-1}\sum_{i=0}^{B-1}\tilde{X}_{-i}\tilde{X}_{-i}^\top + 2\gamma\frac{4\gamma BR}{1-4\gamma BR}\sum_{i=0}^{B-1}\sum_{i=0}^{B-1}\tilde{X}_{-i}\tilde{X}_{-i}^\top \quad (136)$$

Here we have used the fact that $4\gamma BR < 1$ to convert the finite sum to an infinite sum. Using the bound on $\gamma$, we conclude the upper bound. The lower bound follows with a similar proof.

$\qquad\square$

**Lemma 29.** *Suppose $\gamma BR < \frac{1}{6}$. Let $G := \mathbb{E}\tilde{X}_{-i}\tilde{X}_{-i}^\top$ and $M_4 := \mathbb{E}\|\tilde{X}_{-i}\|^4$. Then, we have:*

$$\mathbb{E}\left[\tilde{H}_{0,B-1}^\top\tilde{H}_{0,B-1}|\tilde{\mathcal{D}}_{-0}\right] \preceq I - \frac{4\gamma B}{\mathbb{P}(\tilde{\mathcal{D}}_{-0})}\left(1 - \frac{2\gamma BR}{1-4\gamma BR}\right)G +$$

$$\frac{4\gamma B\sqrt{M_4(1-\mathbb{P}(\tilde{\mathcal{D}}_{-0}))}}{\mathbb{P}(\tilde{\mathcal{D}}_{-0})}\left(1 - \frac{2\gamma BR}{1-4\gamma BR}\right)I$$

*Proof.* The result follows from the statement of Lemma 28, once we show the following inequality via Cauchy Schwarz inequality and the definition of conditional expectation:

$$\mathbb{E}\left[\tilde{X}_{-i}\tilde{X}_{-i}^\top|\tilde{\mathcal{D}}_{-0}\right] \succeq \frac{G}{\mathbb{P}(\tilde{\mathcal{D}}_{-0})} - I\frac{\sqrt{\mathbb{E}\|\tilde{X}_{-i}\|^4}\sqrt{1-\mathbb{P}(\tilde{\mathcal{D}}_{-0})}}{\mathbb{P}(\tilde{\mathcal{D}}_{-0})}.$$

$\qquad\square$

Now we will show that $\tilde{H}_{0,B-1}$ contracts any given vector with probability at-least $p_0 > 0$. For this we will refer to lemma 8 where it is shown that if $X \sim \pi$ then $\langle X, x \rangle$ has mean 0 and is sub-Gaussian with variance proxy $C_\mu x^\top G x$. Using this will show that the matrix $\tilde{H}_{0,B-1}$ operating on a given vector $x$ contracts it with a high enough probability.

**Lemma 30.** *Suppose $\gamma R B < \frac{1}{8}$ and that $\mu$ obeys Assumption 2. There exists a constant $c_0 > 0$ which depends only on $C_\mu$ such that whenever $1 - \mathbb{P}(\tilde{\mathcal{D}}_{-0}) \leq c_0$, then for any arbitrary $x \in \mathbb{R}^2$*

$$\mathbb{P}\left( \|\tilde{H}_{0,B-1} x\|^2 \geq \|x\|^2 - B\gamma x^\top G x \big| \tilde{\mathcal{D}}_{-0} \right) \leq 1 - p_0 < 1 \, .$$

*Where $p_0 > 0$ depends only on $C_\mu$.*

*Proof.* Initially we do not condition on $\tilde{\mathcal{D}}_{-0}$. Consider the quantity: $Y := \sum_{i=0}^{B-1} \langle x, \tilde{X}_{-i} \rangle^2$.

**Claim 5.**
$$\mathbb{P}\left( Y \geq 1/2 B x^\top G x \right) \geq q_0$$

*where $q_0 > 0$ depends only on sub-Gaussianity parameter $C_\mu$*

*Proof.* We consider the Payley-Zygmund inequality which states that for any positive random variable $Y$ with a finite second moment, we have:

$$\mathbb{P}\left( Y > \tfrac{1}{2} \mathbb{E} Y \right) \geq \frac{1}{4} \frac{(\mathbb{E} Y)^2}{\mathbb{E} Y^2} \, .$$

Note that $\mathbb{E} Y = B x^\top G x$. The statement of the lemma follows once we lower bound the quantity $\frac{(\mathbb{E} Y)^2}{\mathbb{E} Y^2}$. Clearly, $(\mathbb{E} Y)^2 = B^2 x^\top G x$. Now,

$$\mathbb{E} Y^2 = \sum_{i,j} \mathbb{E}\langle x, X_i \rangle^2 \langle x, X_j \rangle^2 \leq \sum_{i,j} \sqrt{\mathbb{E}\langle x, X_i \rangle^4} \sqrt{\mathbb{E}\langle x, X_j \rangle^2} = B^2 \mathbb{E}\langle x, X_i \rangle^4$$
$$\leq B^2 c_1 C_\mu^2 (x^\top G x)^2 \tag{137}$$

Here, the second step follows from Cauchy-Schwarz inequality. The third step follows from the fact that $X_i$ are all identically distributed. The fourth step follows from Lemma 8 and Theorem 2.1 from [51]. The statement of the claim follows once we apply Payley-Zygmund inequality. $\qquad\square$

Now, by definition of conditional probabililty and Claim 5, we have:

$$\mathbb{P}\left( \sum_{i=0}^{B-1} \langle x, \tilde{X}_{-i} \rangle^2 \leq \frac{B}{2} x^T G x \bigg| \tilde{\mathcal{D}}_{-0} \right) \leq \frac{(1 - q_0)}{\mathbb{P}(\tilde{\mathcal{D}}_{-0})}$$

Now the statement of the lemma follows from an application of Lemma 28 $\qquad\square$

Now we want to bound the operator norm of $\prod_{s=a}^{a+b} \tilde{H}_{0,B-1}^s$ with high probability under the event $\cap_{s=a}^{a+b} \tilde{\mathcal{D}}_{-0}^s$.

**Lemma 31.** *Suppose the conditions in Lemma 30 hold. Let $\sigma_{\min}(G)$ denote the smallest eigenvalue of $G$. We also assume that $\mathbb{P}(\tilde{\mathcal{D}}^{a,b}) > 1/2$. Conditioned on the event $\tilde{\mathcal{D}}^{a,b}$,*

1. $\| \prod_{s=a}^{b} \tilde{H}_{0,B-1}^s \| \leq 1$ *almost surely*
2. *Whenever $b - a + 1$ is larger than some constant which depends only on $C_\mu$, we have:*

$$\mathbb{P}\left( \| \prod_{s=a}^{b} \tilde{H}_{0,B-1}^s \| \geq 2(1 - \gamma B \sigma_{\min}(G))^{c_4(b-a+1)} \bigg| \tilde{\mathcal{D}}^{a,b} \right) \leq \exp(-c_3(b-a+1) + c_5 d)$$

*Where $c_3, c_4$ and $c_5$ are constants which depend only on $C_\mu$*

*Proof.*

1. The proof follows from an application of Lemma 28.
2. We will prove this with an $\epsilon$ net argument over the sphere in $\mathbb{R}^d$ dimensions.

Suppose we have arbitrary $x \in \mathbb{R}^d$ such that $\|x\| = 1$. Conditioned on the event $\tilde{\mathcal{D}}^{a,b}$, the matrices $\tilde{H}^s_{0,B-1}$ are all independent for $a \leq s \leq b$. We also note that $\tilde{H}^s_{0,B-1}$ is independent of $\tilde{\mathcal{D}}^t$ for $t \neq s$. Let $K_v := \prod_{s=v}^b \tilde{H}^s_{0,B-1}$. When $v \geq b+1$, we take this product to be identity. Consider the set of events $\mathcal{G}_v := \{\|\tilde{H}^v_{0,B-1} K_{v+1} x\|^2 \leq \|K_{v+1} x\|^2 (1 - \gamma B \sigma_{\min}(G))\}$. From Lemma 30, we have that whenever $v \in (a, b)$:

$$\mathbb{P}(\mathcal{G}^c_v | \tilde{\mathcal{D}}^v, \tilde{H}^s_{0,B-1} : s \neq v) \leq 1 - p_0 \tag{138}$$

Where $p_0$ is given in Lemma 30

Let $D \subseteq \{a, \ldots, b\}$ such that $|D| = r$. It is also clear from item 1 and the definitions above that whenever the event $\cap_{v \in D} \mathcal{G}_v$ holds, we have:

$$\| \prod_{s=a}^b \tilde{H}^s_{0,B-1} x\| \leq (1 - \gamma B \sigma_{\min}(G))^{\frac{r}{2}}. \tag{139}$$

Therefore, whenever Equation (139) is violated, we must have a set $D^c \subseteq \{a, \ldots, b\}$ such that $|D^c| \geq b - a - r$ and the event $\cap_{v \in D^c} \mathcal{G}^c_v$ holds. We will union bound all such events indexed by $D^c$ to obtain an upper bound on the probability that Equation (139) is violated. Therefore, using Equation (138) along with the union bound, we have:

$$\mathbb{P}\left( \| \prod_{s=a}^b \tilde{H}^s_{0,B-1} x\| \geq (1 - \gamma B \sigma_{\min}(G))^{\frac{r}{2}} \middle| \tilde{\mathcal{D}}^{a,b} \right) \leq \binom{b-a+1}{b-a-r}(1-p_0)^{b-a-r}$$

Whenever $b - a + 1$ is larger than some constant depending only on $C_\mu$, we can pick $r = c_2(b - a + 1)$ for some constant $c_2 > 0$ small enough such that:

$$\mathbb{P}\left( \| \prod_{s=a}^b \tilde{H}^s_{0,B-1} x\| \geq (1 - \gamma B \sigma_{\min}(G))^{\frac{r}{2}} \middle| \tilde{\mathcal{D}}^{a,b} \right) \leq \exp(-c_3(b - a + 1))$$

Now, let $\mathcal{N}$ be a $1/2$-net of the sphere $\mathcal{S}^{d-1}$. Using Corollary 4.2.13 in [52], we can choose $|\mathcal{N}| \leq 6^d$. By Lemma 4.4.1 in [52] we show that:

$$\| \prod_{s=a}^b \tilde{H}^s_{0,B-1}\| \leq 2 \sup_{x \in \mathcal{N}} \| \prod_{s=a}^b \tilde{H}^s_{0,B-1} x\| \tag{140}$$

By union bounding Equation (140) for every $x \in \mathcal{N}$, we conclude that:

$$\mathbb{P}\left( \| \prod_{s=a}^b \tilde{H}^s_{0,B-1}\| \geq 2(1 - \gamma B \sigma_{\min}(G))^{c_4(b-a+1)} \middle| \tilde{\mathcal{D}}^{a,b} \right) \leq |\mathcal{N}| \exp(-c_3(b - a + 1))$$
$$= \exp(-c_3(b - a + 1) + c_5 d) \tag{141}$$

$\square$

Now we will give a high probability bound for the following operator:

$$Fa, N := \sum_{r=a}^{N-1} \prod_{s=a+1}^r \tilde{H}^s_{0,B-1} \tag{142}$$

Here, we use the convention that $\prod_{s=a+1}^a \tilde{H}^s_{0,B-1} = I$

**Lemma 32.** *Suppose $c_4 \gamma B \sigma_{\min}(G) < \frac{1}{4}$ for the constant $c_4$ as given in Lemma 31. Suppose all the conditions given in the statement of Lemma 31 hold. Then, for any $\delta \in (0, 1)$, we have:*

$$\mathbb{P}\left( \|F_{a,N}\| \geq C\left(d + \log \frac{N}{\delta} + \frac{1}{\gamma B \sigma_{\min}(G)}\right) \middle| \tilde{\mathcal{D}}^{a,N-1} \right) \leq \delta$$

*Where $C$ is a constant which depends only on $C_\mu$*

*Proof.* We consider the triangle inequality: $\|F_{a,N}\| \leq \sum_{t=a}^{N-1} \left\| \prod_{s=a+1}^{t} \tilde{H}_{0,B-1}^s \right\|$. By Lemma 31, we have that whenever $t - a \geq \frac{c_5 d}{c_3} + \frac{\log \frac{N}{\delta}}{c_3}$:

$$\mathbb{P} \left( \| \prod_{s=a+1}^{t} \tilde{H}_{0,B-1}^s \| \geq 2(1 - \gamma B \sigma_{\min}(G))^{c_4(t-a)} \middle| \tilde{\mathcal{D}}^{a,N-1} \right) \leq \frac{\delta}{N}$$

Using union bound, we show that when conditioned on $\tilde{\mathcal{D}}^{a,N-1}$, with probability at least $1 - \delta$ the following holds:

1. For all $a \leq t \leq N - 1$ such that $t - a \geq \frac{c_5 d}{c_3} + \frac{\log \frac{N}{\delta}}{c_3}$:

$$\| \prod_{s=t}^{N} \tilde{H}_{0,B-1}^s \| \leq 2(1 - \gamma B \sigma_{\min}(G))^{c_4(t-a)}$$

2. For all $t$ such that $t - a < \frac{c_5 d}{c_3} + \frac{\log \frac{N}{\delta}}{c_3}$, we have: $\| \prod_{s=t}^{N} \tilde{H}_{0,B-1}^s \| \leq 1$. For this, we use the almost sure bound given in item 1 of Lemma 31

Therefore, when conditioned on $\tilde{\mathcal{D}}^{a,N-1}$, with probability at least $1 - \delta$ we have:

$$\|F_{a,N}\| \leq C(d + \log \frac{N}{\delta}) + 2 \sum_{j=0}^{\infty} (1 - \gamma B \sigma_{\min}(G))^{c_4 j}$$

$$\leq C(d + \log \frac{N}{\delta}) + 2 \sum_{j=0}^{\infty} \exp(-c_4 j \gamma B \sigma_{\min}(G))$$

$$\leq C(d + \log \frac{N}{\delta}) + \frac{2}{1 - \exp(-c_4 \gamma B \sigma_{\min}(G))}$$

$$\leq C(d + \log \frac{N}{\delta}) + \frac{2}{c_4 \gamma B \sigma_{\min}(G) - \frac{c_4^2 \gamma^2 B \sigma_{\min}(G)}{2}}$$

$$\leq C \left( d + \log \frac{N}{\delta} + \frac{1}{\gamma B \sigma_{\min}(G)} \right) \tag{143}$$

In the first step, we have used the event described above to bound the operator norm via. the infinite geometric series. In the second step, we have used the inequality $(1 - x)^a \leq \exp(-ax)$ whenever $x \in [0,1]$ and $a > 0$. In the fourth step, we have used the inequality $\exp(-x) \leq 1 - x + \frac{x^2}{2}$ whenever $x \in [0,1]$. In the last step, we have absorbed constants into a single constant $C$ □

We will now consider the averaged iterate of the coupled process as defined in Equation (21) with $a = 0$.

$$\hat{\tilde{A}}_{0,N}^v := \frac{1}{N} \sum_{t=1}^{N} \left( \tilde{A}_B^{t-1,v} \right) \tag{144}$$

We recall the definition of $\Delta_{t-1}$ from the beginning of the Section L and the recursion shown in Equation (132). We combine these with Equation (144) to show:

$$\hat{\tilde{A}}_{0,N}^v = \frac{1}{N} \sum_{t=1}^{N} \Delta_{t-1} F_{t-1,N} \tag{145}$$

Where $F_{a,N}$ is as defined in Equation (142). Using the results in Lemma 27 and a similar proof technique we show the following theorem. We define the following event as considered in Lemma (32):

$$\tilde{\mathcal{M}}^{t-1} := \left\{ \|F_{t-1,N}\| \leq C \left( d + \log \frac{N}{\delta} + \frac{1}{\gamma B \sigma_{\min}(G)} \right) \right\}$$

Define the event $\tilde{\mathcal{M}}^{0,N-1} = \cap_{t=0}^{N-1} \tilde{\mathcal{M}}^t$ and recall the definition of the event $\tilde{\mathcal{D}}^{0,N-1}$.

**Theorem 33.** *We suppose that the conditions in Lemmas 27, 32 and 28 hold. We also assume that* $\mathbb{P}(\tilde{\mathcal{M}}^{0,N-1} \cap \tilde{\mathcal{D}}^{0,N-1}) \geq \frac{1}{2}$. *Define* $\alpha := C(d + \log \frac{N}{\delta} + \frac{1}{\gamma B \sigma_{\min}(G)})$ *as in the definition of the event* $\tilde{\mathcal{M}}^t$

$$
\mathbb{P}\left( \|\hat{\tilde{A}}_{0,N}^v\| > \beta \,\middle|\, \tilde{\mathcal{M}}^{0,N-1} \cap \tilde{\mathcal{D}}^{0,N-1} \right) \leq \exp\left( c_1 d - \frac{\beta^2 N}{16\gamma C_\mu \sigma_{\max}(\Sigma)(1+2\alpha)} \right) .
$$

*Proof.* Recall the events $\tilde{\mathcal{D}}^{t,N-1}$ and define $\tilde{\mathcal{M}}^{t,N-1} := \cap_{s=t}^{N-1} \tilde{\mathcal{M}}^t$. We recall that $\Delta_{t-1}$ is independent of $F_{t-1,N}$ and $\tilde{\mathcal{D}}^{t,N-1}$. Now consider arbitrary $x, y \in \mathbb{R}^d$ such that $\|x\| = \|y\| = 1$. Define $\Gamma_{t-1,N-1} := \frac{1}{N}\sum_{s=t}^N \Delta_{s-1} F_{s-1,N}$. For any $\lambda > 0$, consider the following exponential moment:

$$
\mathbb{E}\left[ \exp\left( \lambda \langle x, (\hat{\tilde{A}}_{0,N}^v)y \rangle \right) \,\middle|\, \tilde{\mathcal{M}}^{0,N-1} \cap \tilde{\mathcal{D}}^{0,N-1} \right]
$$

$$
= \frac{\mathbb{E}\left[ \exp\left( \lambda \langle x, (\hat{\tilde{A}}_{0,N}^v)y \rangle \right) \mathbb{1}\left( \tilde{\mathcal{M}}^{0,N-1} \cap \tilde{\mathcal{D}}^{0,N-1} \right) \right]}{\mathbb{P}\left( \tilde{\mathcal{M}}^{0,N-1} \cap \tilde{\mathcal{D}}^{0,N-1} \right)}
$$

$$
= \frac{\mathbb{E}\left[ \exp\left( \frac{\lambda}{N} \langle x, \Delta_0 F_{0,N} y \rangle + \lambda \langle x, \Gamma_{1,N-1} y \rangle \right) \mathbb{1}\left( \tilde{\mathcal{M}}^{0,N-1} \cap \tilde{\mathcal{D}}^{0,N-1} \right) \right]}{\mathbb{P}\left( \tilde{\mathcal{M}}^{0,N-1} \cap \tilde{\mathcal{D}}^{0,N-1} \right)} \tag{146}
$$

Here, we note that $\Delta_0$ is independent of $\tilde{\mathcal{M}}^{0,N-1}$, $F_{0,N}$ and $\tilde{D}^{1,N-1}$. We integrate out $\Delta_0$ in Equation (146) using item 2 of Lemma 27 by using the fact that $\tilde{\mathcal{D}}^{0,N-1} = \tilde{\mathcal{D}}^{1,N-1} \cap \tilde{\mathcal{D}}^0_{-0}$ to show:

$$
\mathbb{E}\left[ \exp\left( \lambda \langle x, (\hat{\tilde{A}}_{0,N}^v)y \rangle \right) \,\middle|\, \tilde{\mathcal{M}}^{0,N-1} \cap \tilde{\mathcal{D}}^{0,N-1} \right]
$$

$$
\leq \frac{\mathbb{E}\left[ \exp\left( \gamma \frac{\lambda^2 C_\mu}{N^2} \langle x, \Sigma x \rangle \|F_{0,N} y\|^2 + \lambda \langle x, \Gamma_{1,N-1} y \rangle \right) \mathbb{1}\left( \tilde{\mathcal{M}}^{0,N-1} \cap \tilde{\mathcal{D}}^{1,N-1} \right) \right]}{\mathbb{P}\left( \tilde{\mathcal{M}}^{0,N-1} \cap \tilde{\mathcal{D}}^{0,N-1} \right)} \tag{147}
$$

We use the fact that $F_{0,N} = I + \tilde{H}^1_{0,B-1} F_{1,N}$ to conclude: $\|F_{0,N} y\|^2 = \|y\|^2 + 2\langle y, \tilde{H}^1_{0,B-1} F_{1,N} y \rangle + \langle y, F_{1,N}^T \tilde{H}^{1,\top}_{0,B-1} \tilde{H}^1_{0,B-1} F_{1,N} y \rangle$. Under the event $\tilde{\mathcal{M}}^{0,N-1} \cap \tilde{\mathcal{D}}^{1,N-1}$, we have: $\|\tilde{H}^1_{0,B-1}\| \leq 1$ and $\|F_{1,N}\| \leq \alpha$. Therefore, $\|F_{0,N} y\|^2 \leq \|y\|^2 (1+2\alpha) + \langle y, F_{1,N}^T \tilde{H}^{1,\top}_{0,B-1} \tilde{H}^1_{0,B-1} F_{1,N} y \rangle$. Using this in Equation (147), we conclude:

$$
\mathbb{P}\left( \tilde{\mathcal{M}}^{0,N-1} \cap \tilde{\mathcal{D}}^{0,N-1} \right) \mathbb{E}\left[ \exp\left( \lambda \langle x, (\hat{\tilde{A}}_{0,N}^v)y \rangle \right) \,\middle|\, \tilde{\mathcal{M}}^{0,N-1} \cap \tilde{\mathcal{D}}^{0,N-1} \right]
$$

$$
\leq \mathbb{E}\left[ \exp\left( \Omega + \lambda \langle x, \Gamma_{1,N-1} y \rangle \right) \mathbb{1}\left( \tilde{\mathcal{M}}^{0,N-1} \cap \tilde{\mathcal{D}}^{1,N-1} \right) \right]
$$

$$
\leq \mathbb{E}\left[ \exp\left( \Omega + \lambda \langle x, \Gamma_{1,N-1} y \rangle \right) \mathbb{1}\left( \tilde{\mathcal{M}}^{1,N-1} \cap \tilde{\mathcal{D}}^{1,N-1} \right) \right], \tag{148}
$$

where $\Omega := \gamma \frac{\lambda^2 C_\mu}{N^2} \langle x, \Sigma x \rangle (1+2\alpha) \|y\|^2 + \gamma \frac{\lambda^2 C_\mu}{N^2} \langle x, \Sigma x \rangle \langle y, F_{1,N}^T \tilde{H}^{1,\top}_{0,B-1} \tilde{H}^1_{0,B-1} F_{1,N} y \rangle$. In the last step we have used the fact that $\tilde{\mathcal{M}}^{0,N-1} \cap \tilde{\mathcal{D}}^{1,N-1} \subseteq \tilde{\mathcal{M}}^{1,N-1} \cap \tilde{\mathcal{D}}^{1,N-1}$. We continue just like before but use item 1 of Lemma 27 instead of item 2 to keep peeling terms of the form $\langle x, \Delta_{t-1} F_{t-1,N} y \rangle$ to conclude:

$$
\mathbb{E}\left[ \exp\left( \lambda \langle x, (\hat{\tilde{A}}_{0,N}^v)y \rangle \right) \,\middle|\, \tilde{\mathcal{M}}^{0,N-1} \cap \tilde{\mathcal{D}}^{0,N-1} \right] \leq 2 \exp\left( \gamma \frac{\lambda^2 C_\mu}{N} \langle x, \Sigma x \rangle (1+2\alpha) \|y\|^2 \right)
$$

$$
\leq 2 \exp\left( \gamma \frac{\lambda^2 C_\mu}{N} \sigma_{\max}(\Sigma)(1+2\alpha) \right) \tag{149}
$$

Where $\sigma_{\max}(\Sigma)$ is the maximum eigenvalue of the covariance matrix $\Sigma$. Here we have used the assumption that $\mathbb{P}\left(\tilde{\mathcal{M}}^{0,N-1} \cap \tilde{\mathcal{D}}^{0,N-1}\right) \geq \frac{1}{2}$ and the fact that $\|x\| = \|y\| = 1$. We apply Chernoff bound to $\langle x, (\hat{\tilde{A}}^v_{0,N})y\rangle$ using Equation (149) to conclude that for any $\beta, \lambda \in \mathbb{R}^+$

$$\mathbb{P}\left(\langle x, (\hat{\tilde{A}}^v_{0,N})y\rangle > \beta \,\bigg|\, \tilde{\mathcal{M}}^{0,N-1} \cap \tilde{\mathcal{D}}^{0,N-1}\right) \leq 2\exp\left(\gamma\frac{\lambda^2 C_\mu}{N}\sigma_{\max}(\Sigma)\rangle(1+2\alpha) - \beta\lambda\right) \quad (150)$$

Choose $\lambda = \frac{N\beta}{2\gamma C_\mu \sigma_{\max}(\Sigma)(1+2\alpha)}$ to conclude:

$$\mathbb{P}\left(\langle x, (\hat{\tilde{A}}^v_{0,N})y\rangle > \beta \,\bigg|\, \tilde{\mathcal{M}}^{0,N-1} \cap \tilde{\mathcal{D}}^{0,N-1}\right) \leq 2\exp\left(-\frac{\beta^2 N}{4\gamma C_\mu \sigma_{\max}(\Sigma)(1+2\alpha)}\right)$$

We now apply an $\epsilon$ net argument just like in Lemma 31. Suppose $\mathcal{N}$ is a $1/4$-net of the sphere in $\mathbb{R}^d$. By Corollary 4.2.13 in [52], we can choose $|\mathcal{N}| \leq 12^d$. By Exercise 4.4.3 in [52], we conclude that:

$$\|\hat{\tilde{A}}^v_{0,N}\| \leq 2 \sup_{x,y\in\mathcal{N}} \langle x, (\hat{\tilde{A}}^v_{0,N})y\rangle.$$

Therefore,

$$\mathbb{P}\left(\|\hat{\tilde{A}}^v_{0,N}\| > \beta \,\bigg|\, \tilde{\mathcal{M}}^{0,N-1} \cap \tilde{\mathcal{D}}^{0,N-1}\right)$$

$$\leq \mathbb{P}\left(\sup_{x,y\in\mathcal{N}} \langle x, (\hat{\tilde{A}}^v_{0,N})y\rangle > \frac{\beta}{2} \,\bigg|\, \tilde{\mathcal{M}}^{0,N-1} \cap \tilde{\mathcal{D}}^{0,N-1}\right)$$

$$\leq |\mathcal{N}|^2 \sup_{x,y\in\mathcal{N}} \mathbb{P}\left(\langle x, (\hat{\tilde{A}}^v_{0,N})y\rangle > \frac{\beta}{2} \,\bigg|\, \tilde{\mathcal{M}}^{0,N-1} \cap \tilde{\mathcal{D}}^{0,N-1}\right)$$

$$\leq 2(12)^{2d} \exp\left(-\frac{\beta^2 N}{16\gamma C_\mu \sigma_{\max}(\Sigma)(1+2\alpha)}\right) \leq \exp\left(c_1 d - \frac{\beta^2 N}{16\gamma C_\mu \sigma_{\max}(\Sigma)(1+2\alpha)}\right) \quad (151)$$

$\square$

# M  Lower Bounds

Consider the notations as defined in Section 4. The idea behind the proof is to consider an appropriate Bayesian error lower bound to the minimax error. To construct such a prior distribution, we consider binary tuples $M = (M_{ij} \text{ for } i,j \in [d], i < j) \in \{0,1\}^{d(d-1)/2}$ and $\epsilon \in (0, \frac{1}{4d})$. We construct the symmetric matrix corresponding to $M$, denoted by $A(M)$ as:

$$A(M)_{ij} = \begin{cases} \frac{1}{2} & \text{if } i = j \\ \frac{1}{4d} - \epsilon M_{ij} & \text{if } i < j \end{cases} \quad (152)$$

For the sake of clarity, we denote $\mathcal{L}_{\text{pred}}(\cdot; A(M), \mathcal{N}(0, \sigma^2 I))$ by $\mathcal{L}_{\text{pred}}(\cdot; M)$. We use $\pi_M$ to denote the stationary distribution of $\mathsf{VAR}(A(M), \mathcal{N}(0, \sigma^2 I))$ and the data co-variance matrix at stationarity to be $G_M := \mathbb{E}_{X\sim\pi_M} XX^\top$. By $(Z_t) \sim M$, we mean $(Z_1, \ldots, Z_T) \sim \mathsf{VAR}(A(M), \mathcal{N}(0, \sigma^2 I))$. We will first list some useful results in the following Lemmas:

**Lemma 34.** *Suppose Assumption 1 holds for $\mathsf{VAR}(A^*, \mu)$ and let its stationary distribution be $\pi$. Let $G := \mathbb{E}_{X\sim\pi} XX^\top$. Then,*

$$\mathcal{L}_{\text{pred}}(A) - \mathcal{L}_{\text{pred}}(A^*) = \text{Tr}\left[(A - A^*)^\top (A - A^*)G\right]$$

**Lemma 35.** *For every $M \in \{0,1\}^{d(d-1)/2}$ we have:*

$$\sigma^2 I \preceq G_M \preceq 3\sigma^2 I$$

*Proof.* First we note by Gershgorin circle theorem that $\|A(M)\| \leq \frac{3}{4}$. Given a stationary sequence $(Z_0, \ldots, Z_T) \sim M$ and the corresponding noise sequence $\eta_0, \ldots, \eta_T \sim \mathcal{N}(0, \sigma^2 I)$ i.i.d, we have by stationarity definition: $Z_{t+1} = A(M)Z_t + \eta_t$ and $Z_{t+1}, Z_t$ are both stationary. Therefore:

$$G_M = \mathbb{E}Z_{t+1}Z_{t+1}^\top = A(M)\mathbb{E}Z_t Z_t^\top A(M)^\top + \mathbb{E}\eta_t \eta_t^\top = A(M)G_M A(M)^\top + \sigma^2 I\,.$$

From this we conclude that $G_M \succeq \sigma^2 I$. Now, expanding the recursion above, we have:

$$G_M = \sigma^2 \sum_{i=0}^\infty A(M)^i (A(M)^\top)^i \preceq \sigma^2 \sum_{i=0}^\infty \left(\frac{9}{16}\right)^i I = \frac{16\sigma^2}{7} I \tag{153}$$

In the second step we have the fact that $\|A(M)\| \leq \frac{3}{4}$ to show that $A(M)^i (A(M)^\top)^i \preceq \left(\frac{9}{16}\right)^i I$ $\quad \square$

Suppose $M$ and $M'$ are such that their Hamming distance is 1 (i.e, $A(M)$ and $A(M')$ differ in exactly two places). We want to bound the total variation distance between the corresponding stationary sequences $(Z_0, Z_1, \ldots, Z_T) \sim \mathsf{VAR}(A(M), \mathcal{N}(0, \sigma^2 I))$ and $(Z_0', Z_1', \ldots, Z_T') \sim \mathsf{VAR}(A(M'), \mathcal{N}(0, \sigma^2 I))$.

**Lemma 36.** *Let the quantities be as defined above. For some universal constant $c$, whenever $\epsilon < c\min(\frac{1}{\sqrt{T}}, \frac{1}{d})$, we have:*

$$TV\left((Z_0, \ldots, Z_T), (Z_0', \ldots, Z_T')\right) \leq \frac{1}{2}$$

*By the existence of maximal coupling (see Chapter I, Theorem 5.2 in [53]), we conclude that we can define $(Z_0, \ldots, Z_T)$ and $(Z_0', \ldots, Z_T')$ on a common probability space such that:*

$$\mathbb{P}((Z_0, \ldots, Z_T) = (Z_0', \ldots, Z_T')) \geq \frac{1}{2}$$

*Proof.* We will first bound the KL divergence between the two distributions and infer the bound on TV distance from Pinsker's inequality. Consider $p_{M,T}$ and $p_{M',T}$ to be the respective probability density functions of $(Z_0, \ldots, Z_T) \sim M$ and $(Z_0', \ldots, Z_T') \sim M'$ respectively. In this proof, we will use $Z_{t,-}$ to denote the tuple $(Z_0, \ldots, Z_t)$. Now, by definition of KL divergence, we have:

$$\begin{aligned}
\mathsf{KL}(p_{M,T}\|p_{M',T}) &= \mathbb{E}_{Z \sim p_{M,T}} \log \frac{p_{M,T}(Z_0, \ldots, Z_T)}{p_{M',T}(Z_0, \ldots, Z_T)} \\
&= \mathbb{E}_{Z \sim p_{M,T}} \log \frac{p_{M,T}(Z_T|Z_{T-1,-})}{p_{M',T}(Z_T|Z_{T-1,-})} + \mathbb{E}_{Z \sim p_{M,T}} \log \frac{p_{M,T-1}(Z_0, \ldots, Z_{T-1})}{p_{M',T-1}(Z_0, \ldots, Z_{T-1})} \\
&= \mathbb{E}_{Z \sim p_{M,T}} \log \frac{p_{M,T}(Z_T|Z_{T-1,-})}{p_{M',T}(Z_T|Z_{T-1,-})} + \mathsf{KL}(p_{M,T-1}\|p_{M',T-1}) \\
&= \mathbb{E}_{Z \sim p_{M,T}} \log \frac{p_{M,T}(Z_T|Z_{T-1})}{p_{M',T}(Z_T|Z_{T-1})} + \mathsf{KL}(p_{M,T-1}\|p_{M',T-1}) \tag{154}
\end{aligned}$$

The first 3 steps above follow from the definition of KL divergence and conditional density. In the last step we have used the Markov property of the sequence $Z_0, \ldots, Z_T$ which in this case shows that the law of $Z_T|Z_{T-1}$ is the same as the law $Z_T|Z_{T-1,-}$. Using Equation (154) recursively and noting that $(Z_t, Z_{t-1})$ are identically distributed for every $t \in \{1, \ldots, T\}$, we conclude:

$$\mathsf{KL}(p_{M,T}\|p_{M',T}) = T\mathbb{E}_{(Z_0, Z_1) \sim p_{M,1}} \log \frac{p_{M,1}(Z_1|Z_0)}{p_{M',1}(Z_1|Z_0)} + \mathsf{KL}(\pi_M\|\pi_{M'}) \tag{155}$$

We will first bound $\mathbb{E}_{(Z_0, Z_1) \sim p_{M,1}} \log \frac{p_{M,1}(Z_1|Z_0)}{p_{M',1}(Z_1|Z_0)}$. Conditioned on $Z_0$, the law of $Z_1$ under the model $M$ is $\mathcal{N}(A(M)Z_0, \sigma^2 I)$. Similarly, the conditional law of $Z_1$ under the model $M'$ is $\mathcal{N}(A(M')Z_0, \sigma^2 I)$. Therefore, a simple calculation shows that:

$$\mathbb{E}_{(Z_0,Z_1)\sim p_{M,1}} \log \frac{p_{M,1}(Z_1|Z_0)}{p_{M',1}(Z_1|Z_0)} = \mathbb{E}_{Z_0\sim\pi_M} \frac{\|(A(M)-A(M'))Z_0\|^2}{2\sigma^2}$$

$$= \mathbb{E}_{Z_0\sim\pi_M} \text{Tr}\left((A(M)-A(M'))^\top (A(M)-A(M')) \frac{Z_0 Z_0^\top}{2\sigma^2}\right)$$

$$= \frac{1}{2\sigma^2} \text{Tr}\left((A(M)-A(M'))^\top (A(M)-A(M')) G_M\right)$$

$$\leq \frac{3}{2} \text{Tr}\left((A(M)-A(M'))^\top (A(M)-A(M'))\right)$$

$$= \frac{3}{2}\|A(M)-A(M')\|_{\mathsf{F}}^2 = 3\epsilon^2. \tag{156}$$

In the first step, we have used standard KL formula for Gaussians with different mean but same variance. In the third step we have used the fact that $Z_0 \sim \pi_M$. In the fourth step, we have used the upper bound on $G_M$ from Lemma 35. In the last step we have used the definition of $A(M)$ and the fact that the Hamming distance between $M$ and $M'$ is 1. Now we consider: $\text{KL}(\pi_M\|\pi_{M'})$

Clearly, $\pi_M = \mathcal{N}(0, G_M)$. By standard formula for KL divergence between Gaussians,

$$\text{KL}(\pi_M\|\pi_{M'}) = \frac{1}{2}\left[\text{Tr}(G_{M'}^{-1}G_M) - d + \log \frac{\det G_{M'}}{\det G_M}\right]. \tag{157}$$

First we consider $\text{Tr}(G_{M'}^{-1}G_M)$. Clearly, $G_M = \sigma^2(I - A(M)^2)^{-1}$ and $G_{M'} = \sigma^2(I - A(M')^2)^{-1}$. Therefore, $G_{M'}^{-1} = G_M^{-1} + \frac{A(M)^2-A(M')^2}{\sigma^2}$. We have:

$$\text{Tr}(G_{M'}^{-1}G_M) = \text{Tr}(I) + \text{Tr}\left(\frac{A(M)^2-A(M')^2}{\sigma^2}G_M\right) \leq d + d\left\|\frac{A(M)^2-A(M')^2}{\sigma^2}G_M\right\|$$

$$\leq d + d\frac{\|G_M\|}{\sigma^2}\|A(M)^2-A(M')^2\| \leq d + 3d\|A(M)^2-A(M')^2\|$$

$$= d + 3d\|(A(M)-A(M'))A(M)+A(M')(A(M)-A(M'))\|$$

$$\leq d + 3d\left[\|A(M)-A(M')\|\|A(M)\| + \|A(M')\|\|A(M)-A(M')\|\right]$$

$$\leq d + \frac{9}{2}d\epsilon. \tag{158}$$

In the second step we have used the fact that $tr(B) \leq d\|B\|$. In the future steps, we have made use of the sub-multiplicativity of the operator norm and the upper bound on $\|G_M\|$ given by Lemma 35. We have also used the fact that by Gershgorin theorem $\|A(M)\| \leq \frac{3}{4}$ and $\|A(M)-A(M')\| = \epsilon$.

Next, we will bound $\log \frac{\det G_{M'}}{\det G_M}$. Suppose $\mu_1 \geq \cdots \geq \mu_d$ be the eigenvalues of $A(M)$ and $\mu_1' \geq \cdots \geq \mu_d'$ be the eigenvalues of $A(M')$. We conclude that:

$$\log \frac{\det G_{M'}}{\det G_M} = \sum_{i=1}^{d} \log\left(\frac{1-\mu_i^2}{1-(\mu_i')^2}\right).$$

Now, $\|A(M)-A(M')\| \leq \epsilon$. Therefore, we conclude by Weyl inequalities that $|\mu_i - \mu_i'| \leq \epsilon$. By Gershgorin circle theorem, we also conclude that $\frac{1}{4} \leq \mu_i' \leq \frac{3}{4}$

Plugging this into the equation above, we have:

$$\log \frac{\det G_{M'}}{\det G_M} = \sum_{i=1}^{d} \log\left(\frac{1-\mu_i^2}{1-(\mu_i')^2}\right) \leq \sum_{i=1}^{d} \log\left(\frac{1-(\mu_i'-\epsilon)^2}{1-(\mu_i')^2}\right) = \sum_{i=1}^{d} \log\left(1+\frac{2\mu_i'-\epsilon^2}{1-(\mu_i')^2}\right)$$

$$\leq \sum_{i=1}^{d} \log\left(1+4\epsilon\right) \leq 4\epsilon d \tag{159}$$

Combining Equations (158) and (159) along with Equation (157) we conclude:

$$\text{KL}(\pi_M\|\pi_{M'}) \leq 5\epsilon d.$$

Using this along with Equations (156) and (155), we conclude:
$$\mathsf{KL}(p_{M,T}\|p_{M',T}) = 3\epsilon^2 T + 5\epsilon d. \tag{160}$$
From this we conclude that when $\epsilon$ is as given in the statement of the lemma, we have:
$$\mathsf{KL}(p_{M,T}\|p_{M',T}) \leq \frac{1}{8}. \tag{161}$$
By Pinsker's inequality, which states that $\mathsf{TV} \leq \sqrt{2\mathsf{KL}}$, we conclude the result of the lemma. $\qquad\square$

*Theorem 4.* We first note that when we choose $\sigma^2$ such that $d\sigma^2 = \beta$, we have
$$\mathsf{VAR}(A(M), \mathcal{N}(0, \sigma^2 I)) \in \mathcal{M}$$
for every $M \in \{0,1\}^{d(d-1)/2}$. We pick $\epsilon = c\min(\frac{1}{\sqrt{T}}, \frac{1}{d})$ so that Lemma 36 is satisfied.

We draw $M$ randomly from the uniform measure over $\{0,1\}^{d(d-1)/2}$ and lower bound the minimax error by Bayesian error.

$$\mathcal{L}_{\mathsf{minmax}}(\mathcal{M}) \geq \inf_{f \in \mathcal{F}} \mathbb{E}_M \mathbb{E}_{(Z_t)\sim M} \mathcal{L}_{\mathsf{pred}}(f(Z_0, \ldots, Z_T); M) - \mathcal{L}_{\mathsf{pred}}(A(M); M) \tag{162}$$

We will now uniformly lower bound $\mathbb{E}_M \mathbb{E}_{(Z_t)\sim M} \mathcal{L}_{\mathsf{pred}}(f(Z_0, \ldots, Z_T); M) - \mathcal{L}_{\mathsf{pred}}(A(M); M)$ for every fixed choice of $f \in \mathcal{F}$ to conclude the statement of the theorem from Equation (162). Henceforth, we will denote $f(Z_0, \ldots, Z_T)$ by $\hat{A}(M)$ whenever $(Z_t) \sim M$. By Lemma 34, we conclude that:
$$\mathcal{L}_{\mathsf{pred}}(\hat{A}(M); M) - \mathcal{L}_{\mathsf{pred}}(A(M); M) = \mathrm{Tr}\left[(\hat{A}(M) - A(M))^\top (\hat{A}(M) - A(M))G_M\right].$$
$(\hat{A}(M) - A(M))^\top (\hat{A}(M) - A(M))$ is a PSD matrix and by Lemma 35, $G_M \geq \sigma^2 I$ for every $M$. Therefore, we conclude that with probability 1 we have:

$$\mathcal{L}_{\mathsf{pred}}(\hat{A}(M); M) - \mathcal{L}_{\mathsf{pred}}(A(M); M) \geq \sigma^2 \mathrm{Tr}\left[(\hat{A}(M) - A(M))^\top (\hat{A}(M) - A(M))\right]$$
$$= \sigma^2 \|\hat{A}(M) - A(M)\|_{\mathsf{F}}^2 \geq 2\sigma^2 \sum_{\substack{i,j \in [d] \\ i < j}} (\hat{A}(M)_{ij} - A(M)_{ij})^2. \tag{163}$$

Therefore, we conclude that:
$$\mathbb{E}_M \mathbb{E}_{Z_t \sim M} \mathcal{L}_{\mathsf{pred}}(\hat{A}(M); M) - \mathcal{L}_{\mathsf{pred}}(A(M); M) \geq 2 \sum_{\substack{i,j \in [d] \\ i < j}} \mathbb{E}_M \mathbb{E}_{(Z_t)\sim M} (\hat{A}(M)_{ij} - A(M)_{ij})^2. \tag{164}$$

We will now lower bound every term in the summation in the RHS of Equation (164). Fix $(i, j)$. Let $M_{\sim ij}$ denote all the co-ordinates of $M$ other than $(i, j)$. We define $M^+, M^- \in \{0,1\}^{d(d-1)/2}$ so that $M_{\sim ij}^+ = M_{\sim ij}$ and $M_{ij}^+ = 1$. Similarly, let $M_{\sim ij}^- = M_{\sim ij}$ and $M_{ij}^- = 0$. Therefore, we have:

$$\mathbb{E}_M \mathbb{E}_{(Z_t)\sim M} (\hat{A}(M)_{ij} - A(M)_{ij})^2 = \frac{1}{2} \mathbb{E}_{M_{\sim ij}} \mathbb{E}_{(Z_t)\sim M^+} (\hat{A}(M^+)_{ij} - A(M^+)_{ij})^2$$
$$+ \frac{1}{2} \mathbb{E}_{M_{\sim ij}} \mathbb{E}_{(Z_t)\sim M^-} (\hat{A}(M^-)_{ij} - A(M^-)_{ij})^2. \tag{165}$$

Now, $M^+$ and $M^-$ differ in exactly one co-ordinate. We invoke Lemma 36 to show that there exists a coupling between $(Z_t^+) \sim M^+$ and $Z_t^- \sim M^-$ such that $\mathbb{P}(Z_t^+ = Z_t^-) \geq \frac{1}{2}$. Call this event $\Gamma$ (we ignore the dependence on $M_{\sim ij}$ for the sake of clarity). In this event, we must have $\hat{A}(M^+) = \hat{A}(M^-)$ since our estimator $f \in \mathcal{F}$ is a measurable function of the data. For any fixed $M_{\sim ij}$, we have:
$$\mathbb{E}_{(Z_t)\sim M^+} (\hat{A}(M^+)_{ij} - A(M^+)_{ij})^2 + \mathbb{E}_{(Z_t)\sim M^-} (\hat{A}(M^-)_{ij} - A(M^-)_{ij})^2$$
$$\geq \mathbb{E}_{(Z_t)} \mathbb{1}(\Gamma) \left[(\hat{A}(M^+)_{ij} - A(M^+)_{ij})^2 + (\hat{A}(M^+)_{ij} - A(M^-)_{ij})^2\right]$$
$$\geq \mathbb{P}(\Gamma)(A(M^-)_{ij} - A(M^+)_{ij})^2 \geq \frac{1}{2}(A(M^-)_{ij} - A(M^+)_{ij})^2 = \frac{\epsilon^2}{2}. \tag{166}$$

In the second line we have used the fact that under event $\Gamma$, $\hat{A}(M^+) = \hat{A}(M^-)$. In the third line, we have used the inequality $(x - y)^2 + (x - z)^2 \geq \frac{1}{2}(y - z)^2$. In the fourth line, we have used the fact that $\mathbb{P}(\Gamma) \geq 1/2$. Using Equation (166) along with Equations (165) and (164), we conclude that for every estimator $f \in \mathcal{F}$ the following holds:

$$\mathbb{E}_M \mathbb{E}_{Z_t \sim M}[\mathcal{L}_{\text{pred}}(\hat{A}(M); M) - \mathcal{L}_{\text{pred}}(A(M); M)] \geq \frac{d(d-1)\epsilon^2\sigma^2}{4}.$$

Using above equation with Equation (162), we conclude the statement of the theorem. $\qquad\square$

**Remark.** *We can show a similar lower bound by considering a discrete prior over the space of orthogonal matrices. In particular taking $A^*$ to be an orthogonal matrix scaled by $\rho$, we can endow the orthogonal (or special orthogonal) group with metric induced by the Frobenius norm. Then from [54, Proposition 7], we can construct an $\epsilon$-cover of cardinality $d^{\frac{d(d-1)}{2}}$. But then from the proof of [55, Proposition 3], for $\alpha \in (0, 1)$, there exists a local packing of the space with packing distance $\alpha\epsilon$ and cardinality at least $c^{d(d-1)/2}$ where $c > 1$. Further the diameter of this local packing is at most $2\epsilon$ (in Frobenius norm). Now using standard arguments from Fano's inequality (c.f.[55, Proposition 3]) or Birge's inequality (c.f.[5, Lemma F.1]) we can get a similar lower bound on the prediction error as Theorem 4 but with explicit dependence on $\rho$.*

# N    Techincal Proofs

## N.1    Proof of Lemma 10

*Proof.* Consider the SGD − RER iteration:

$$A_{i+1}^{t-1} = A_i^{t-1} - 2\gamma(A_i^{t-1}X_{-i}^{t-1} - X_{-(i+1)}^{t-1})X_{-i}^{t-1,\top}$$
$$= A_i^{t-1}(I - 2\gamma X_{-i}^{t-1}X_{-i}^{t-1,\top}) + 2\gamma X_{-(i-1))}^{t-1}X_{-(i+1)}^{t-1,\top} \qquad (167)$$

Observe that for our choice of $\gamma$ and under the event $\mathcal{D}^{0,N-1}$, we have $\|(I - 2\gamma X_{-i}^{t-1}X_{-i}^{t-1,\top})\| \leq 1$ and $\|X_{-(i+1)}^{t-1}X_{-i}^{t-1,\top}\| \leq R$. Therefore, triangle inequality implies:

$$\|A_{i+1}^{t-1}\| \leq \|A_i^{t-1}\| + 2\gamma R$$

We conclude the bound in the Lemma.

$\qquad\square$

## N.2    Proof of Lemma 11

*Proof.* We again consider the evolution equation: $\tilde{X}_{-i}^{t-1}$

$$A_{i+1}^{t-1} = A_i^{t-1} - 2\gamma(A_i^{t-1}X_{-i}^{t-1} - X_{-(i+1)}^{t-1})X_{-i}^{t-1,\top}$$
$$= A_i^{t-1} - 2\gamma(A_i^{t-1}\tilde{X}_{-i}^{t-1} - \tilde{X}_{-(i+1)}^{t-1})\tilde{X}_{-i}^{t-1,\top} + \Delta_{t,i} \qquad (168)$$

Where

$$\Delta_{t,i} = 2\gamma A_i^{t-1}\left(\tilde{X}_{-i}^{t-1}\tilde{X}_{-i}^{t-1,\top} - X_{-i}^{t-1}X_{-i}^{t-1,\top}\right) + 2\gamma\left(X_{-(i+1)}^{t-1}X_{-i}^{t-1,\top} - \tilde{X}_{-(i+1)}^{t-1}\tilde{X}_{-i}^{t-1,\top}\right)$$

Using Lemmas 10 and 7, we conclude that:

$$\|\Delta_{t,i}\| \leq (16\gamma^2 R^2 T + 8\gamma R)\|A^{*u}\|$$

Using the recursion for $\tilde{A}_i^t$, we conclude:

$$A_{i+1}^{t-1} - \tilde{A}_{i+1}^{t-1} = (A_i^{t-1} - \tilde{A}_i^{t-1})\tilde{P}_i^t + \Delta_{t,i}$$
$$\implies \left\|A_{i+1}^{t-1} - \tilde{A}_{i+1}^{t-1}\right\| \leq \left\|A_i^{t-1} - \tilde{A}_i^{t-1}\right\|\left\|\tilde{P}_i^t\right\| + (16\gamma^2 R^2 T + 8\gamma R)\|A^{*u}\|$$
$$\implies \left\|A_{i+1}^{t-1} - \tilde{A}_{i+1}^{t-1}\right\| \leq \left\|A_i^{t-1} - \tilde{A}_i^{t-1}\right\| + (16\gamma^2 R^2 T + 8\gamma R)\|A^{*u}\| \qquad (169)$$

In the last step we have used the fact that under the event $\hat{\mathcal{D}}^{0,N-1}$, we must have $\left\|\tilde{P}_i^t\right\| \leq 1$. We conclude the statement of the lemma from Equation (169). $\qquad\square$

### N.3 Proof of Lemma 12

*Proof.* First we have

$$
\mathbb{E}\left[\left(A_j^{t-1}-A^*\right)^\top\left(A_j^{t-1}-A^*\right)\mathbb{1}\left[\mathcal{D}^{0,t-1}\right]\right] \preceq \mathbb{E}\left[\left(A_j^{t-1}-A^*\right)^\top\left(A_j^{t-1}-A^*\right)\mathbb{1}\left[\hat{\mathcal{D}}^{0,t-1}\right]\right]
$$
$$
+ 4\gamma^2(Bt)^2 R\sqrt{\mu_4}\frac{1}{T^{\alpha/2}}I
$$
$$
\preceq \mathbb{E}\left[\left(A_j^{t-1}-A^*\right)^\top\left(A_j^{t-1}-A^*\right)\mathbb{1}\left[\hat{\mathcal{D}}^{0,t-1}\right]\right]
$$
$$
+ c\gamma^2 d\sigma_{\max}(\Sigma)RT^2\frac{1}{T^{\alpha/2}}I \tag{170}
$$

Next, we have

$$
\left\|\left(A_j^{t-1}-A^*\right)^\top\left(A_j^{t-1}-A^*\right)-\left(\tilde{A}_j^{t-1}-A^*\right)^\top\left(\tilde{A}_j^{t-1}-A^*\right)\right\|
$$
$$
\leq \left\|A_j^{t-1}-\tilde{A}_j^{t-1}\right\|\left(\left\|\left(A_j^{t-1}-A^*\right)\right\|+\left\|\left(\tilde{A}_j^{t-1}-A^*\right)\right\|\right)
$$
$$
\leq \left\|A_j^{t-1}-\tilde{A}_j^{t-1}\right\|\left(2\left\|A^*\right\|+\left\|A_j^{t-1}\right\|+\left\|\tilde{A}_j^{t-1}\right\|\right) \tag{171}
$$

Thus on the event $\hat{\mathcal{D}}^{0,t-1}$, using lemma 11 and lemma 10 we get

$$
\left\|\left(A_j^{t-1}-A^*\right)^\top\left(A_j^{t-1}-A^*\right)-\left(\tilde{A}_j^{t-1}-A^*\right)^\top\left(\tilde{A}_j^{t-1}-A^*\right)\right\|
$$
$$
\leq c(\gamma^2 R^2 T^2 + \gamma RT)(\gamma RT + \|A^*\| + \|A_0\|)\|A^{*u}\| \leq c\gamma^3 R^3 T^3\|A^{*u}\| \tag{172}
$$

for some constant $c$. (We have suppressed the dependence on $A_0$ and $A^*$ since they are constants and $\gamma RT$ grows with $T$).

The proof follows by combining (170) and (172).

The proof of (17) follows similarly. □

## O Prediction error for sparse systems

In this section we consider the VAR$(A^*,\mu)$ model with sparse $A^*$ whose sparsity pattern is known. We will present a modification of SGD − RER that takes into account the sparsity pattern information. Formally, let $S_l = \{k : A_{l,k}^* \neq 0\}$ be support or sparsity pattern of row $l$ of $A^*$. Further let $s_l = |S_l|$ denote the sparsity of row $j$. We assume that $S_l$ is known for each $1 \leq l \leq d$. The claim is that the excess expected prediction loss is of order $\frac{\sum_l s_l \sigma_l^2}{T}$. We will present only a sketch of the proof highlighting the main steps. Detailed calculations follow similarly as in sections F and G.

The modification of the SGD − RER algorithm to use the sparsity pattern is as follows. Let $a_l^{*,\top}$ denote row $l$ of $A^*$. The algorithmic iterates are given by $(A_j^{t-1})$ where row $l$ is $a_{j,l}^{t-1,\top}$. Let $a_{0,l}^0 = 0 \in \mathbb{R}^d$. Let $\{e_l : 1 \leq l \leq d\}$ denote the standard basis of $\mathbb{R}^d$. Let $P_{S_l} : \mathbb{R}^d \to \mathbb{R}^d$ denote the (self adjoint) orthogonal projection operator onto the subspace spanned by $\{e_l : l \in S_l\}$. Then update for row $l$ is given by

$$
a_{j+1,l}^{t-1,\top} = \left[a_{j,l}^{t-1,\top} - 2\gamma(a_{j,l}^{t-1,\top}X_{-j}^{t-1} - \langle e_l, X_{-(j-1)}^{t-1}\rangle)X_{-j}^{t-1,\top}\right]P_{S_l} \tag{173}
$$

and $a_{0,l}^t = a_{B,l}^{t-1}$. Since each iterate above has sparsity pattern $S_l$ by construction, we can rewrite the above as

$$
a_{j+1,l}^{t-1,\top} = a_{j,l}^{t-1,\top} - 2\gamma(a_{j,l}^{t-1,\top}X_{-j}^{t-1} - \langle e_l, X_{-(j-1)}^{t-1}\rangle)\left(P_{S_l}X_{-j}^{t-1}\right)^\top \tag{174}
$$

Notice that $a_{j,l}^{t-1,\top}X_{-j}^{t-1} = a_{j,l}^{t-1,\top}P_{S_l}X_{-j}^{t-1}$ and

$$
\langle e_l, X_{-(j-1)}^{t-1}\rangle = a_l^{*,\top}X_{-j}^{t-1} + \eta_{-j,l}^{t-1}
$$

Thus

$$\left(a_{j+1,l}^{t-1} - a_l^*\right)^\top = \left(a_{j,l}^{t-1} - a_l^*\right)^\top \left(P_{S_l} - 2\gamma \left(P_{S_l} X_{-j}^{t-1}\right)\left(P_{S_l} X_{-j}^{t-1}\right)^\top\right) + 2\gamma\eta_{-j,l}^{t-1} \left(P_{S_l} X_{-j}^{t-1}\right)^\top \tag{175}$$

For a vector $v \in \mathbb{R}^d$, let $v_{S_l} \in \mathbb{R}^{s_l}$ be the vector corresponding to the support $S_l$ i.e. entries in $v_{S_l}$ correspond to the entries in $v$ whose indices are in $S_l$. So we can rewrite (175) completely in $\mathbb{R}^{s_l}$ as

$$\left(a_{j+1,l}^{t-1} - a_l^*\right)^\top_{S_l} = \left(a_{j,l}^{t-1} - a_l^*\right)^\top_{S_l} \left(I_{s_l} - 2\gamma \left(X_{-j}^{t-1}\right)_{S_l}\left(X_{-j}^{t-1}\right)^\top_{S_l}\right) + 2\gamma\eta_{-j,l}^{t-1} \left(X_{-j}^{t-1}\right)^\top_{S_l} \tag{176}$$

where $I_{s_l}$ is the identity matrix of dimension $s_l$.

Our goal is to bound the expected prediction error for this modified $\mathsf{SGD-RER}$. To that end, we will make some important observations.

(1) Since we focus on prediction error, the entire analysis can be carried out row by row. To see this, if $\hat{A}$ is any estimator, the

$$\mathcal{L}_{\mathsf{pred}}(\hat{A}; A^*, \mu) - \mathrm{Tr}(\Sigma) = \mathrm{Tr}(G(\hat{A} - A^*)^\top(\hat{A} - A)) = \sum_{l=1}^{d} \mathrm{Tr}(G(\hat{a}_l - a_l^*)(\hat{a}_l - a_l^*)^\top)$$

where $\hat{a}_l^\top$ is the row $l$ of $\hat{A}$.

(2) If $\hat{a}_l$ and $a_l^*$ have sparsity pattern $S_l$ then

$$\begin{aligned}\mathrm{Tr}(G(\hat{a}_l - a_l^*)(\hat{a}_l - a_l^*)^\top) &= \mathrm{Tr}(P_{S_l} G P_{S_l}(\hat{a}_l - a_l^*)(\hat{a}_l - a_l^*)^\top) \\ &= \mathrm{Tr}(G_{S_l}(\hat{a}_l - a_l^*)_{S_l}(\hat{a}_l - a_l^*)^\top_{S_l})\end{aligned}$$

where $G_{S_l} \in \mathbb{R}^{s_l \times s_l}$ is the submatrix of $G$ obtained by picking rows and columns corresponding to indices in $S_l$.

(3) Under the stationary measure, we have $\mathbb{E}\left[\left(P_{S_l} X_{-j}^{t-1}\right)\left(P_{S_l} X_{-j}^{t-1}\right)^\top\right] = P_{S_l} G P_{S_l}$. Thus, with high probability $\left\|P_{S_l} X_{-j}^{t-1}\right\|^2 \leq c s_l \sigma_{\max}(G) \log T$.

(4) Letting $s_0 = \max_l s_l$, we can set $R = c s_0 \sigma_{\max}(G) \log T$ and use step size $\gamma = O(1/RB)$.

(5) We can perform the same bias-variance decomposition as described in section D to obtain $a_{B,l}^{t-1,v}$ and $a_{B,l}^{t-1,b}$.

(6) From previous observations, the variance of last iterate corresponding to row $l$ turns out to be

$$\gamma\sigma_l^2(1 - o(1))I_{s_l} \preceq \mathbb{E}\left[\left(a_{B,l}^{t-1,v}\right)_{S_l}\left(a_{B,l}^{t-1,v}\right)^\top_{S_l}\right] \preceq \frac{\gamma}{1 - \gamma R}\sigma_l^2(1 + o(1))I_{s_l}$$

where $\sigma_l^2 = \Sigma_{l,l}$.

(7) Similarly, the variance of the average iterate $\mathbb{E}\left[(\hat{a}_{0,N,l}^v)(\hat{a}_{0,N,l}^v)^\top\right]$ corresponding to row $l$ can be bounded upto leading order by

$$\frac{1}{N^2}\sum_{t=1}^{N}\left[V_{t-1,l}(I_{s_l} - \mathcal{H}_{S_l})^{-1} + (I_{s_l} - \mathcal{H}_{S_l}^\top)^{-1}V_{t-1,l}\right]$$

where $V_{t-1,l} = \mathbb{E}\left[\left(a_{B,l}^{t-1,v}\right)_{S_l}\left(a_{B,l}^{t-1,v}\right)^\top_{S_l}\right]$ and (with abuse of notation) $\mathcal{H}_{S_l}$ is defined as

$$\mathcal{H}_{S_l} = \mathbb{E}\left[\prod_{j=0}^{B-1}\left(I_{s_l} - 2\gamma(\tilde{X}_{-j}^0)_{S_l}(\tilde{X}_{-j}^0)^\top_{S_l}\right)\mathbf{1}\left[\cap_{j=0}^{B-1}\left\{\left\|(\tilde{X}_{-j}^0)_{S_l}\right\|^2 \leq R\right\}\right]\right]$$

where $\tilde{X}_0^0 \sim \pi$.

(8) Now, similar to lemma 16 we can bound $\mathcal{H}_{S_l} + \mathcal{H}_{S_l}^\top$ by $2(I_{s_l} - c\gamma BG_{s_l})$ upto leading order.

(9) Thus similar to lemma 17 we obtain

$$\text{Tr}(G_{S_l}(I - \mathcal{H}_{S_l})^{-1}) \leq c\frac{s_l}{\gamma B}$$

(10) Finally as in section G.1 we can bound the variance of prediction error of row $l$ upto leading order by

$$\text{Tr}(G\mathbb{E}\left[(\hat{a}^v_{0,N,l})(\hat{a}^v_{0,N,l})^\top\right]) \lesssim \frac{\sigma_l^2 s_l}{T}$$

Thus summing over $l$ we get

$$\text{Tr}\left(G\mathbb{E}\left[(\hat{A}^v_{0,N})(\hat{A}^v_{0,N})^\top\right]\right) \lesssim \frac{\sum_l \sigma_l^2 s_l}{T}$$

(11) Bias can also be analyzed in a similar way and it will be of strictly lower order (using suitable tail-averaging).

(12) Thus the excess prediction loss is given bounded as

$$\mathbb{E}\left[\mathcal{L}_{\text{pred}}(\hat{A}_{N/2,N}; A^*, \mu)\right] - \text{Tr}(\Sigma) \lesssim \frac{\sum_l \sigma_l^2 s_l}{T}$$

So the modified $\mathsf{SGD} - \mathsf{RER}$ algorithm effectively utilizes the low dimensional structure in $A^*$.