# OpenReview forum: "Streaming Linear System Identification with Reverse Experience Replay"
_NeurIPS.cc/2021/Conference — NeurIPS 2021 Poster_

### Official Review · Reviewer_SVQN · 2021-07-15

**Rating:** 7
**Confidence:** 3

**Summary:**

The authors consider a rather special case of a linear dynamical system, where there is no hidden state and the transition matrix is "externally stable" (less than 1 in the operator norm), process noise is sub-gaussian, and the stationary distribution has bounded fourth moment. In this case, they suggest running the SGD "backwards" in small buffers, which they call "reverse experience replay". This leads to the independence of the observations, and the unbiased nature of the SGD updates.

**Ethical Concerns:**

None.

**Limitations And Societal Impact:**

None.

**Main Review:**

Originality:

This is original, as far as I can tell. The only issue could be the concurrent submission of id 1136 on the use of related algorithms for the identification of non-linear systems (under even stricter assumptions).

Quality:

The algorithm is practical and readily implementable. The analysis is quite sophisticated.

Clarity:

The paper is a joy to read.

The significance:

I imagine that this could become influential.

**Time Spent Reviewing:**

3

---

> ### Author Response · Authors · 2021-08-09
> **Response to the review**
>
> We thank the reviewer for the kind comments. We are happy to answer any subsequent queries.

---

> > ### Comment · Reviewer_SVQN · 2021-08-14
> > **Additional references**
> >
> > I would like to "chime in" on the additional references suggested by the other reviewers. Those are all important pieces of recent work, just as well as:
> >
> > MKS Faradonbeh et al: Finite time identification in unstable linear systems,  Automatica 2018.
> >
> > M Kozdoba et al.: On-Line Learning of Linear Dynamical Systems: Exponential Forgetting in Kalman Filters, AAAI 2019.
> >
> > V Kuznetsov, M Mohri: Time series prediction and online learning, Conference on Learning Theory, 2016.
> >
> > A Tsiamis et al: Sample complexity of Kalman filtering for unknown systems. Learning for Dynamics and Control, 2020

---

> > > ### Author Response · Authors · 2021-08-15
> > > **Response to additional references**
> > >
> > > Thank you for the relevant references we missed out. We will add them along with a brief comparison wherever necessary.

---

> > ### Comment · Reviewer_SVQN · 2021-08-14
> > **Additional references**
> >
> > It may also be worth adding references from within the RL community, where RER (or more often "reverse replay") is common. See e.g.
> >
> > T Haga: Recurrent network model for learning goal-directed sequences through reverse replay. Elife, 2018.
> >
> > E. Rotinov: Reverse Experience Replay, https://arxiv.org/abs/1910.08780
> >
> > MT Whelan et al: A robotic model of hippocampal reverse replay for reinforcement learning. arXiv preprint arXiv:2102.11914, 2021
> >
> > I appreciate that those are mostly unofficial publications and the authors need not have known about those.

---

### Official Review · Reviewer_EQ4W · 2021-07-16

**Rating:** 6
**Confidence:** 3

**Summary:**

The paper proposes an SGD-style algorithm for linear dynamical system identification. The authors claim that the proposed algorithm is the first of its kind to have near-optimal error rates. The key idea behind the algorithm is to run SGD in reverse on streaming segments of the data.

**Main Review:**

Online (linear) dynamical system identification is an important problem in machine learning. The paper motivates this well and clearly describes the new approach that is designed to tackle previous challenges.

What is missing in my opinion is a more thorough comparison (both theoretically and empirically) with other online algorithms that use the Sherman-Morrison-Woodbury (SMW) formula. The authors say that the application of such algorithms is limited. In what way is it really limited? Can you give a side-by-side comparison of the error rates? Also, you say that they do not apply to practically important settings like generalized non-linear dynamical systems. As far as the current paper goes, you do not show that SGD-RER can be applied to this case (you only briefly mention the work of Jain et al. (2021) on this matter). It would have also been nice to include a SMW-style online algorithm in the experiments.

The other weakness of the proposed algorithm, in my opinion, is the fact that it introduces extra parameters such as the buffer size and gap, which both need be set as a function of the horizon. Sometimes the horizon is not known. Can the new approach easily be adapted to this case?

Finally, in line 243, you claim the rate in the lower bound in Thm 3 matches that of Theorem 1. However, in Thm 1, I see 1/sqrt{T} rate, whereas Thm 3 has a 1/T rate. What am I missing here?


**Time Spent Reviewing:**

3

---

> ### Author Response · Authors · 2021-08-09
> **Response to the reviewer**
>
> We thank the reviewer for the kind comments
>
> ### Regarding comparison to SMW formula based methods:
> Methods based on the SMW formula implement the OLS solution exactly at every time step $t$. We have, therefore, compared the performance of SGD-RER with SMW in the experiments via. the OLS comparison. We will clarify this aspect in the experiments section.
>
> ### Advantages of SGD style method:
> Most machine learning problems do not have a closed form solution and hence methods like SMW formula cannot be used for streaming estimation efficiently. However, SGD style methods are widely applicable. For example, even if we obtain noisy gradients from the system, our technique should apply, but an SMW based approach wouldn’t apply anymore.
> Similarly, a recent work by (Jain et. al 2021) developed on our SGD-RER method and the techniques to develop a streaming algorithm for learning non-linear dynamical systems. We will elaborate on their results in our manuscript. We also refer to item (4) in page 7 - where we show that knowing a sparse support for the matrix A (say, via prior information through a social network type structure), then we can efficiently run SGD-RER only over the support of the matrix A, with an improved guarantee on the error.
>
> ### Regarding mismatch between Theorem 1 and 3:
> There is indeed a typo in Theorem 3 where a square root needs to be inserted in the RHS. We thank the reviewer for pointing out this important detail.

---

> > ### Comment · Reviewer_EQ4W · 2021-08-25
> > **Thanks for the response**
> >
> > Thanks for responding to my comments. I still lean towards acceptance, and will maintain my score.

---

### Official Review · Reviewer_UGDX · 2021-07-16

**Rating:** 7
**Confidence:** 4

**Summary:**

This paper considers systems identification in vectorized auto-regressive model with stochastic noise. The paper focuses on the practicality of system identification with nearly optimal computation and sample complexity instead of using offline-style methods such as OLS. The authors propose a new SI algorithm based on SGD that uses reverse experience replay to address the challenges due to the inherent correlation between samples. Performance bounds on the accuracy of estimating the transition matrix as well as prediction error are given, both of which nearly matching the lower bounds. Numerical simulations show that SGD-RER is competitive to OLS, which enjoying a smaller time complexity.

**Limitations And Societal Impact:**

Yes, the authors have discussed the limitations of their work.

**Main Review:**

Originality:
- The idea of reversing replay buffer to mitigate correlation is interesting, novel, and practical.
- It is unclear to me why the SDG approach in [25] is insufficient. The partial observable LDS considered in [25] is a more general version of the auto-regressive model considered here. Also, another work by Hazan et al. 2017 uses online projected gradient descent, which also handles correlations in LDS via a standard online convex optimization argument. Could you discuss the differences between your work and these papers?

Quality:
- Strong theoretical guarantees are provided for both estimation and prediction errors. The leading term in error bound is independent of mixing time, which improves over prior work.
- Recent results on linear system identification and prediction that use OLS completely remove dependency on mixing time. While the leading term in error bound is nearly independent of mixing time, I wonder if it is also possible to remove the dependency of SGD stepsize on mixing time as well. Could you explain the challenges involved in removing this dependency?
- The experiments section can be improved. It would be interesting to see whether the SGD-RER is robust to the choice of step size.

Clarity:
- The paper is organized well and easy to follow.

Significance:
- The paper is well-motivated and increases the practicality of linear system identification.

Additional references:
- Hazan, Elad, Karan Singh, and Cyril Zhang. "Learning linear dynamical systems via spectral filtering." Advances in Neural Information Processing Systems 30 (2017): 6702-6712.
- Tsiamis, Anastasios, and George J. Pappas. "Finite sample analysis of stochastic system identification." 2019 IEEE 58th Conference on Decision and Control (CDC). IEEE, 2019.
- Rashidiejad, Paria, Jiantao Jiao, and Stuart Russell. "SLIP: Learning to Predict in Unknown Dynamical Systems with Long-Term Memory." 34th Conference on Neural Information Processing Systems (NeurIPS 2020), Vancouver, Canada. 2020.
- Lee, Holden. "Improved rates for identification of partially observed linear dynamical systems." arXiv preprint arXiv:2011.10006 (2020).
- Tsiamis, Anastasios, and George J. Pappas. "Linear systems can be hard to learn." arXiv preprint arXiv:2104.01120 (2021).

-----
After the response:

I thank the authors for responding to my questions. The discussion on the dependency of the bounds on mixing time is interesting and would be good to include it in the paper. I am happy with the answers and stick to my score in favor of accepting this paper.

**Time Spent Reviewing:**

5

---

> ### Author Response · Authors · 2021-08-09
> **Response to the reviewer**
>
> We thank the reviewer for the kind comments.
>
> ### Comparison to [25]:
> We agree that this work has a more general setting (while being applicable only to Single Input Single Output  systems). Even though it shows that SGD can learn such systems via a single trajectory, the bounds have a dependence on the spectral radius (more specifically contain polynomial factors of the mixing time of the system) and has  sub-optimal dependence on the dimension of the hidden state $n$ (it depends on $n^5$ instead of $n^2$). We refer to Theorem 5.1 in [25] for the exact bounds. In contrast, our work has no dependence on the mixing time in the dominant error term and has the correct dependence on the dimension as shown by the lower bounds. In fact, removing the dependence on polynomial factors of mixing time is one of the primary goals of our current work.
>
>
> ### Comparison to Hazan et al. (2017) :
> This and subsequent works concern approximating arbitrary sequences with the output of a partially observed linear system with a control input - and attempts to bound the regret with respect to the best fitting linear system in hind-sight. The assumptions are very different in that the system matrix A is assumed to be positive semi definite (which is not the case in our work). They use spectral filtering methods instead of the SGD type methods used in our work. Our work concerns learning the parameters of a linear system without control input via. a single trajectory and aims at providing recovery guarantees instead of bounding regret. Therefore, the setting and goals of their work and the present work are very different.
>
> ### Regarding step size dependence on $\tau_{\mathsf{mix}}$:
> We do not believe that the step size dependence can be improved in the worst case for the following reason: the typical squared norm of the iterates grows as $O(d \tau_{\mathsf{mix}} )$ because of the nature of the VAR models. Indeed, consider the bias decay matrix at time $t$ which is of the form $(I - 2\gamma X_t X_t^{\top})$. Since we need this to be a contraction, we must have $\gamma \leq \frac{1}{|X_t|^2} \lesssim \frac{1}{d \tau_{\mathsf{mix}}}$. It would be interesting to consider a different version of the algorithm where the step size is adaptively normalized by a factor of $1+|X_t|^2$. We will perform more experiments to demonstrate the robustness with respect to the choice of step size.
>
> We thank the reviewer for providing additional references. We will cite these in our manuscript.

---

### Decision · Program_Chairs · 2021-09-27

**Decision:**

Accept (Poster)

**Comment:**

This paper considers the system identification problem in a vector autoregressive process using a single-pass SGD type algorithm with reverse experience replay. The analysis is interesting, and compared to existing work, it has improved the dependence on the mixing time and dimension. Although it has not completely resolved the dependence on the mixing time, I agree with the reviewers that it is already an interesting contribution and a concrete step towards a better understanding of system identifications in linear systems. I am happy to recommend acceptance.